# Nearest Neighbour with Bandit Feedback

**Stephen Pasteris**
The Alan Turing Institute
London UK
spasteris@turing.ac.uk

**Chris Hicks**
The Alan Turing Institute
London UK
c.hicks@turing.ac.uk

**Vasilios Mavroudis**
The Alan Turing Institute
London UK
vmavroudis@turing.ac.uk

## Abstract

In this paper we adapt the nearest neighbour rule to the contextual bandit problem. Our algorithm handles the fully adversarial setting in which no assumptions at all are made about the data-generation process. When combined with a sufficiently fast data-structure for (perhaps approximate) adaptive nearest neighbour search, such as a navigating net, our algorithm is extremely efficient - having a per trial running time polylogarithmic in both the number of trials and actions, and taking only quasi-linear space. We give generic regret bounds for our algorithm and further analyse them when applied to the stochastic bandit problem in euclidean space. We note that our algorithm can also be applied to the online classification problem.

## 1 Introduction

In this paper we adapt the classic *nearest neighbour* rule to the contextual bandit problem and develop an extremely efficient algorithm. The problem proceeds in trials, where on trial $t$: (1) a *context* $x_t$ is revealed to us, (2) we must select an *action* $a_t$, and (3) the *loss* $\ell_{t,a_t} \in [0,1]$ of action $a_t$ on trial $t$ is revealed to us. We assume that the contexts are points in a metric space and the distance between two contexts represents their similarity. A *policy* is a mapping from contexts to actions and the inductive bias of our algorithm is towards learning policies that typically map similar contexts to similar actions. Our main result has absolutely no assumptions whatsoever about the generation of the context/loss sequence and has no restriction on what policies we can compare our algorithm to.

Our algorithm requires, as a subroutine, a data-structure that performs $c$-nearest neighbour search. This data-structure must be *adaptive* - in that new contexts can be inserted into it over time. An example of such a data-structure is the *Navigating net* [17] which, given mild conditions on our metric and dataset, performs both search and insertion in polylogarithmic time. When utilising a data-structure of this speed our algorithm is extremely efficient - with a per-trial time complexity polylogarithmic in both the number of trials and actions, and requiring only quasi-linear space.

As an example we will apply our methodology to the special case of the contextual bandit problem in which the context sequence is drawn i.i.d. from a probability distribution over the $d$-dimensional hypercube. In this case, for any policy $\hat{y}$ with a finite-volume decision boundary, our algorithm achieves $\tilde{\mathcal{O}}\left(\hat{\alpha}T^{d/(d+1)}K^{1/(d+1)}\right)$ regret w.r.t. $\hat{y}$, where $\hat{\alpha}$ measures the magnitude of what is essentially the part of the decision boundary of $\hat{y}$ that lies in the support of the probability distribution.

In the course of this paper we develop some novel algorithmic techniques, including a new algorithmic framework CANPROP and efficient algorithms for searching in trees, which may find further application.

We now describe related works. The bandit problem [20] was first introduced in [28] but was originally studied in the stochastic setting in which all losses are drawn i.i.d. at random [18], [1], [3]. However, our world is very often not i.i.d. stochastic. The work of [4] introduced the seminal EXP3 algorithm which handled the case in which the losses were selected arbitrarily. This work also

37th Conference on Neural Information Processing Systems (NeurIPS 2023).

introduced the EXP4 algorithm for contextual bandits. In general this algorithm is exponential time but in some situations can be implemented in polynomial time - such as their EXP3.S algorithm, which was a bandit version of the classic FIXEDSHARE algorithm [14]. In [12] the EXP3.S setting was greatly generalised to the situation in which the contexts where vertices of a graph. They utilised the methodology of [8], [16] and [13] in order to develop extremely efficient algorithms. Although inspiring this work, these algorithms cannot be utilised in our situation as they inherently require the set of queried contexts to be known a-priori. In the stochastic case another class of contextual bandit problems are *linear bandits* [21], [5] in which the contexts are mappings from the actions into $\mathbb{R}^d$. Here the queried contexts need not be known in advance but the losses must be drawn i.i.d. from a distribution that has mean linear in the respective context. The $k$ nearest neighbour algorithm was first analysed in [6]. The work [26] utilised the $k$ nearest neighbour methodology and the works [9] and [18] to handle a stochastic contextual bandit problem. However, their setting is extremely more restricted than ours. In particular, the context/loss pairs must be drawn i.i.d. at random and the probability distribution they are sampled from must obey certain strict conditions. In addition, on each trial the contexts seen so far must be ordered in increasing distance from the current context and operations must be performed on this sequence, making their algorithm exponentially slower than ours. Our algorithm utilises the works of [22] and [7] as subroutines. It should be noted that the later work, which was based on [23], was improved on in [11] - we leave it as an open problem as to whether we can utilise their work in our algorithm.

Algorithms for contextual bandits in metric spaces have been studied in [15, 19, 24, 25, 27, 29, 30, 31, 2, 26] but as far as we know ours is the first work to give a non-trivial bound for our problem in general (with no additional assumptions). As far as we are aware our above example bound for the fully stochastic case in euclidean space (with finite decision boundary) is also novel - the works [30, 24, 26] scale as $\tilde{\mathcal{O}}(T^{(d+1)/(d+2)})$ in general (and [24, 26] also require additional assumptions). As far as we are aware we are also the first work, stochastic or otherwise, to give a regret scaling as $\tilde{\mathcal{O}}(T^{1/2})$ when the contexts are drawn from well-separated clusters (i.e. there is a positive distance between all pairs of clusters) in a finite-dimensional metric space, and the comparator policy is constant on each cluster (we do, however, conjecture that in the stochastic case [26] can obtain such a regret scaling here - but, as stated above, it is exponentially slower).

## 2    Notation

Let $\mathbb{N}$ be the set of natural numbers not including $0$. Given a natural number $m \in \mathbb{N}$ we define $[m] := \{j \in \mathbb{N} \mid j \leq m\}$. Given a predicate $p$ we define $[\![p]\!] := 1$ if $p$ is true and $[\![p]\!] := 0$ otherwise. We define $\log(\cdot)$ and $\ln(\cdot)$ to be the logarithms with base $2$ and $e$ respectively. Given sets $\mathcal{A}$ and $\mathcal{B}$ we denote by $\mathcal{B}^{\mathcal{A}}$ the set of all functions $f : \mathcal{A} \to \mathcal{B}$ and by $2^{\mathcal{A}}$ the set of all subsets of $\mathcal{A}$. We write $x \sim [0, 1]$ to mean that the value $x$ is drawn from the uniform distribution on $[0, 1]$.

All trees in this paper are considered rooted. Given a tree $\mathcal{J}$ we denote its root by $r(\mathcal{J})$, its vertex set by $\mathcal{J}$, its leaves by $\mathcal{J}^{\star}$, and its internal vertices by $\mathcal{J}^{\dagger}$. Given a vertex $v$ in a tree $\mathcal{J}$ we denote its parent by $\uparrow_{\mathcal{J}}(v)$ and the subtree of all its descendants by $\Downarrow_{\mathcal{J}}(v)$. Given an internal node $v$ in a (full) binary tree $\mathcal{J}$ we denote its left and right children by $\triangleleft_{\mathcal{J}}(v)$ and $\triangleright_{\mathcal{J}}(v)$ respectively. Internal nodes $v$ in a (full) ternary tree $\mathcal{J}$ have an additional child $\triangledown_{\mathcal{J}}(v)$ called the *centre* child. Given vertices $v$ and $v'$ in a tree $\mathcal{J}$ we denote by $\Gamma_{\mathcal{J}}(v, v')$ the *least common ancestor* of $v$ and $v'$: i.e. the vertex of maximum depth which is an ancestor of both $v$ and $v'$. We will drop the subscript $\mathcal{J}$ in all these functions when unambiguous. Given a tree $\mathcal{J}$, a *subtree* of $\mathcal{J}$ is a tree whose edge set is a subset of that of $\mathcal{J}$.

## 3    Results

### 3.1    The General Result

We consider the following game between *Learner* (us) and *Nature* (our adversary). We call this game the *similarity bandit problem*. We have $K$ *actions*. Learning proceeds in $T$ trials. A-priori Nature chooses a sequence $\langle n(t) \mid t \in [T] \setminus \{1\} \rangle$ where for all $t \in [T] \setminus \{1\}$ we have $n(t) \in [t-1]$. A-priori Nature also chooses a sequence of loss vectors $\langle \boldsymbol{\ell}_t \mid t \in [T] \rangle \subseteq [0, 1]^K$, but does not reveal them to Learner. On the $t$-th trial the following happens:

1. If $t > 1$ then Nature reveals $n(t)$ to Learner.
2. Learner chooses some action $a_t \in [K]$.
3. Nature reveals $\ell_{t,a_t}$ to Learner.

Intuitively, given a trial $t \in [T] \setminus \{1\}$, $n(t)$ is a *similar* trial to $t$. Our inductive bias is that if an action $a \in [K]$ is good for trial $t$ then it is likely that $a$ will also be good for the similar trial $n(t)$. We will measure our performance with respect to any policy $\boldsymbol{y}$, where a policy is defined as a vector in $[K]^T$. Specifically, we wish to minimise the $\boldsymbol{y}$-*regret*, which is defined as the difference between the total cumulative loss suffered by Learner and that which Learner would have suffered if it had instead chosen $a_t$ equal to $y_t$ for all trials $t$. Formally, this quantity is defined as follows:

**Definition 3.1.** *Given a policy $\boldsymbol{y} \in [K]^T$ we define the $\boldsymbol{y}$-regret of Learner as:*

$$R(\boldsymbol{y}) := \sum_{t \in [T]} \ell_{t,a_t} - \sum_{t \in [T]} \ell_{t,y_t} \,.$$

The following quantity quantifies how much a policy agrees with our inductive bias.

**Definition 3.2.** *Given a policy $\boldsymbol{y} \in [K]^T$ we define the complexity of $\boldsymbol{y}$ by:*

$$\Phi(\boldsymbol{y}) := 1 + \sum_{t \in [T] \setminus \{1\}} [\![ y_t \neq y_{n(t)} ]\!] \,.$$

We now state our main result:

**Theorem 3.3.** *Consider the similarity bandit problem described above. Our algorithm* CBNN *takes a single parameter $\rho > 0$ and, for all policies $\boldsymbol{y} \in [K]^T$ simultaneously, obtains an expected $\boldsymbol{y}$-regret bounded by:*

$$\mathbb{E}[R(\boldsymbol{y})] \in \tilde{\mathcal{O}} \left( \left( \rho + \frac{\Phi(\boldsymbol{y})}{\rho} \right) \sqrt{KT} \right)$$

*where the expectation is taken over the randomisation of the algorithm.* CBNN *needs no initialisation time and has a per-trial time complexity of:*

$$\mathcal{O}(\ln(T)^2 \ln(K)) \,.$$

## 3.2 Bandits in a Metric Space

We consider the following game between *Learner* (us) and *Nature* (our adversary). We call this game the *metric bandit problem*. We have $K$ *actions* and a metric space $(\mathcal{C}, \Delta)$ where $\mathcal{C}$ is a (possibly infinite) set of *contexts* and for all $x, x' \in \mathcal{C}$ we have that $\Delta(x, x')$ is the *distance* from $x$ to $x'$. We assume that Learner does not necessarily know $(\mathcal{C}, \Delta)$ a-priori but has access to an oracle for computing $\Delta(x, x')$ for any $x, x' \in \mathcal{C}$. Learning proceeds in $T$ trials. A-priori Nature chooses a sequence of contexts $\langle x_t \,|\, t \in [T] \rangle \subseteq \mathcal{C}$ and a sequence of loss vectors $\langle \ell_t \,|\, t \in [T] \rangle \subseteq [0, 1]^K$, but does not reveal them to Learner. On the $t$-th trial the following happens:

1. Nature reveals $x_t$ to Learner.
2. Learner chooses some action $a_t \in [K]$.
3. Nature reveals $\ell_{t,a_t}$ to Learner.

Here, our inductive bias is that if an action $a \in [K]$ is good for a context $x \in \mathcal{C}$ then it is likely also good for contexts that are near to $x$ with respect to the metric $\Delta$. Our algorithm for this problem will be based on the concept of a $c$-nearest neighbour which is defined as follows.

**Definition 3.4.** *Given some $c \geq 1$, a finite set $\mathcal{S} \subseteq \mathcal{C}$, and a context $x \in \mathcal{C}$ we have that some $\hat{x} \in \mathcal{S}$ is a $c$-nearest neighbour of $x$ in the set $\mathcal{S}$ if and only if:*

$$\Delta(x, \hat{x}) \leq c \min_{x' \in \mathcal{S}} \Delta(x, x') \,.$$

In order to utilise CBNN for this problem we need a data-structure for *adaptive nearest neighbour search*. This problem is as follows. We maintain a finite set $\mathcal{S} \subseteq \mathcal{C}$. At any point in time we must either:

- Insert a new context $x \in C$ into the set $S$ and update the data-structure.

- Given a context $x \in C$, utilise the data-structure to find a $c$-nearest neighbour of $x$ in the set $S$.

An efficient example of such a data-structure is the *navigating net* [17].

We can now reduce the metric bandit problem to the similarity bandit problem as follows. On any trial $t \in [T] \setminus \{1\}$ choose $\hat{x}_t$ to be a $c$-nearest neighbour of $x_t$ in the set $\{x_{t'} \mid t' \in [t-1]\}$. Then choose $n(t) \in [t-1]$ such that $x_{n(t)} = \hat{x}_t$.

We will utilise the following definition in order to bound the complexity of policies when $n(\cdot)$ is chosen in this way.

**Definition 3.5.** *Given a function $\hat{y} : C \to [K]$ and a set $\mathcal{X} \subseteq C$ then for all $x \in C$ define:*

$$\gamma(x, \hat{y}, \mathcal{X}) := \min\{\Delta(x, x') \mid x' \in \mathcal{X} \wedge \hat{y}(x') \neq \hat{y}(x)\} \,.$$

We can now bound the complexity of policies as follows, noting that by Theorem 3.3 this leads directly to a regret bound.

**Theorem 3.6.** *Assume we have a sequence $\langle x_t \mid t \in [T] \rangle \subseteq C$ and a function $\hat{y} : C \to [K]$. Define $\mathcal{X} := \{x_t \mid t \in [T]\}$ and for all $t \in [T]$ define $y_t := \hat{y}(x_t)$. Assume that for all $t \in [T] \setminus \{1\}$ we have that $x_{n(t)}$ is a $c$-nearest neighbour of $x_t$ in the set $\{x_{t'} \mid t' \in [t-1]\}$. Then $\Phi(\boldsymbol{y})$ is no greater than the minimum cardinality of any set $S \subseteq C$ in which for all $x' \in \mathcal{X}$ there exists $x \in S$ with $\Delta(x, x') < \gamma(x, \hat{y}, \mathcal{X})/3c$.*

A strength of our algorithm is that it can be combined with *binning* algorithms, where the contexts $x_t$ are partitioned into sets called *bins* and, for each bin, all contexts in that bin are replaced by a single context called the *centre* of the bin. The advantage of binning is that $\gamma(x, \hat{y}, \mathcal{X})$ can increase, so that by Theorem 3.6 we may have that $\Phi(\boldsymbol{y})$ decreases. However, when a context $x_t$ is binned (i.e. replaced by its bin centre) its label $\hat{y}(x_t)$ can change, increasing the final regret by $\mathcal{O}(1)$. In Section 3.3 we give an example of the utilisation of binning.

### 3.3 Stochastic Bandits in Euclidean Space

As an example we now consider the utilisation of the above algorithms for the problem of stochastic bandits in $[0, 1]^d$ for some arbitrary $d \in \mathbb{N}$ which we view as a constant in our bounds. Here we will only focus on what happens in the limit $T \to \infty$. We focus on stochastic bandits for simplicity, but the same methodology can be used to study the limiting behaviour of adversarial bandits. In this problem we have an unknown probability density $\tilde{\mu} : [0, 1]^d \times [0, 1]^K \to \mathbb{R}$. We have $T$ trials. On trial $t$ the following happens:

1. Nature draws $(z_t, \boldsymbol{\ell}_t)$ from $\tilde{\mu}$.

2. Nature reveals $z_t$ to Learner.

3. Learner chooses some action $a_t \in [K]$.

4. Nature reveals $\ell_{t,a_t}$ to Learner.

To aid us in this problem we will first quantise the contexts $z_t$ to a grid. Note that this is an example of binning. The grid is defined as follows:

**Definition 3.7.** *Given $q \in \mathbb{N}$ define $\mathcal{G}_q^d$ to be the set of vectors in $[0, 1]^d$ in which each component is an integer multiple of $1/q$.*

On each trial $t$ we will first quantise the vector $z_t$ by defining $x_t$ to be its nearest neighbour (w.r.t. the euclidean metric) in $\mathcal{G}_q^d$, where $q := (T/K)^{1/(d+1)}$. Note that this can be done in constant time per trial. As in Section 3.2 we then use CBNN to solve the problem by defining $n(t)$ such that $x_{n(t)}$ is a $c$-nearest neighbour (w.r.t. the euclidean metric) of $x_t$ in the set $\{x_{t'} \mid t' \in [t-1]\}$. We consider $c$ as a constant in our bounds.

As before, we will compare our cumulative loss to that of a policy that follows a function $\hat{y} : [0, 1]^d \to [K]$. Our regret bound will be based on the following quantities:

**Definition 3.8.** *Let $\mu$ be the marginal of $\tilde{\mu}$ with respect to its first argument and let $\Delta$ be the euclidean metric on $[0,1]^d$. For all $\epsilon > 0$ define:*

$$\mathcal{E}(\mu, \epsilon) := \{x \in [0,1]^d \,|\, \exists x' \in [0,1]^d \,:\, \mu(x') \neq 0 \wedge \Delta(x, x') \leq \epsilon\}$$

*which is the set of contexts that are within distance $\epsilon$ of the support of $\mu$. Given $\hat{y} : [0,1]^d \to [K]$ we make the following definitions. For any $\delta > 0$ define:*

$$\mathcal{M}(\hat{y}, \mu, \epsilon, \delta) := \{x \in [0,1]^d \,|\, \exists x' \in \mathcal{E}(\mu, \epsilon) \,:\, \Delta(x, x') \leq \delta \wedge \hat{y}(x) \neq \hat{y}(x')\}$$

*which is the set of contexts that are at distance no more than $\delta$ from the intersection of the decision boundary of $\hat{y}$ and $\mathcal{E}(\mu, \epsilon)$. We then define:*

$$\alpha(\hat{y}, \mu) := \lim_{\epsilon \to 0} \lim_{\delta \to 0} \frac{1}{\delta} \int_{x \in \mathcal{M}(\hat{y}, \mu, \epsilon, \delta)} 1 \qquad ; \qquad \tilde{\alpha}(\hat{y}, \mu) := \lim_{\epsilon \to 0} \lim_{\delta \to 0} \frac{1}{\delta} \int_{x \in \mathcal{M}(\hat{y}, \mu, \epsilon, \delta)} \mu(x)$$

*which are essentially the volumes of the part of the decision boundary of $\hat{y}$ that lies in the support of $\mu$, with respect to the uniform density and the density $\mu$ respectively.*

With these definitions in hand we now present our regret bound, which utilises Theorem 3.6 in its proof:

**Theorem 3.9.** *Let $q := \lceil (T/K)^{1/(d+1)} \rceil$. For all $t \in [T]$ let $x_t$ be the nearest neighbour of $z_t$ in the set $\mathcal{G}_q^d$. For all $t \in [T] \setminus \{1\}$ let $n(t)$ be such that $x_{n(t)}$ is a c-nearest neighbour of $x_t$ in the set $\{x_{t'} \,|\, t' \in [t-1]\}$. Given some $\hat{y} : [0,1]^d \to [K]$ let $y_t := \hat{y}(z_t)$ for all $t \in [T]$. Then when $\rho := q^{\frac{d-1}{2}}$ CBNN gives us:*

$$\mathbb{E}[R(\boldsymbol{y})] \in \tilde{\mathcal{O}}\left((1 + \alpha(\hat{y}, \mu) + \tilde{\alpha}(\hat{y}, \mu))T^{\frac{d}{d+1}} K^{\frac{1}{d+1}}\right)$$

*as $T \to \infty$.*

We note that varying $q$ and $\rho$ in Theorem 3.9 will allow us to trade off the values $1$, $\alpha(\hat{y}, \mu)$ and $\tilde{\alpha}(\hat{y}, \mu)$ in different ways.

## 4 The Algorithm

In this section we describe our algorithm CBNN, for solving the similarity bandit problem, and give the pseudocode for the novel subroutines. In the appendix we give a more detailed description of how CBNN works and prove our theorems.

Instead of working directly with trial numbers we create a sequence of distinct *nodes* $\langle x_t \,|\, t \in [T] \rangle$ and define, for all $t \in [T] \setminus \{1\}$, the node $n(x_t) := x_{n(t)}$. Let $\mathcal{X} := \{x_t \,|\, t \in [T]\}$. We can now represent policies as functions from $\mathcal{X}$ into $[K]$. Hence, given some $y : \mathcal{X} \to [K]$, we define the *$y$-regret* and the *complexity* of $y$ as:

$$R(y) := \sum_{t \in [T]} \ell_{t, a_t} - \sum_{t \in [T]} \ell_{t, y(x_t)} \qquad ; \qquad \Phi(y) := 1 + \sum_{x \in \mathcal{X} \setminus \{x_1\}} [\![y(x) \neq y(n(x))]\!]$$

respectively.

### 4.1 A Simple but Inefficient Algorithm

To give the reader intuition we first describe our initial idea - a simple algorithm which attains our desired regret bound but is exponentially slower - taking a per-trial time of $\tilde{\Theta}(KT)$. The algorithm is based on EXP4 [4] which we now describe. On every trial $t$ we maintain a weighting $\hat{w}_t : [K]^{\mathcal{X}} \to [0,1]$. We are free to choose the initial weighting $\hat{w}_1$ to be any probability distribution. On each trial $t$ the following happens:

1. For all $a \in [K]$ set $p_{t,a} \leftarrow \sum_{y \in [K]^{\mathcal{X}}} [\![y(x_t) = a]\!] \hat{w}_t(y)$.
2. Set $a_t \leftarrow a$ with probability proportional to $p_{t,a}$.
3. Receive $\ell_{t, a_t}$.

4. For all $a \in [K]$ set $\hat{\ell}_{t,a} \leftarrow [\![a = a_t]\!]\ell_{t,a_t}\|\boldsymbol{p}_t\|_1/p_{t,a_t}$ .

5. For all $y \in [K]^{\mathcal{X}}$ set $\hat{w}_{t+1}(y) \leftarrow \hat{w}_t(y)\exp(-\eta\hat{\ell}_{t,y(x_t)})$ .

For us we choose, for all $y : \mathcal{X} \to [K]$, an initial weight of:

$$\hat{w}_1(y) := (1/K)(T(K-1))^{-\Phi(y)}(1 - 1/T)^{(T-1-\Phi(y))} \ .$$

Of course, we don't know $\Phi(y)$ a-priori, and hence we cannot implement EXP4 explicitly (and it would take exponential time even if we did know $\Phi(y)$ a-priori). Our crucial insight is the following. For any $t \in [T]$ let $\mathcal{X}_t := \{x_{t'} \,|\, t' \in [t]\}$ and for any $y : \mathcal{X} \to [K]$ let $y^t$ be the restriction of $y$ onto $\mathcal{X}_t$. Then for any $t \in [T]$ and for any $y' : \mathcal{X}_t \to [K]$ we have:

$$\sum_{y \in [K]^{\mathcal{X}} \,:\, y^t = y'} \hat{w}_1(y) \propto \prod_{x \in \mathcal{X}_t \setminus \{x_1\}} \left( \frac{[\![y'(n(x)) \neq y'(x)]\!]}{T(K-1)} + [\![y'(n(x)) = y'(x)]\!]\left(1 - \frac{1}{T}\right) \right) \ . \quad (1)$$

Note that, for all $t \in [T]$ and $a \in [K]$, we can write $p_{t,a}$ as follows:

$$p_{t,a} = \sum_{y \in [K]^{\mathcal{X}}} [\![y(x_t) = a]\!]\hat{w}_1(y) \prod_{t' \in [t-1]} \exp(-\eta\hat{\ell}_{t,y(x_t)})$$

$$= \sum_{y' \in [K]^{\mathcal{X}_t}} [\![y'(x_t) = a]\!] \left( \prod_{t' \in [t-1]} \exp(-\eta\hat{\ell}_{t,y'(x_t)}) \right) \sum_{y \in [K]^{\mathcal{X}} \,:\, y^t = y'} \hat{w}_1(y) \ .$$

By substituting in Equation (1) we have now brought $p_{t,a}$ into a form that can be solved via *Belief propagation* [23] over the tree with vertex set $\mathcal{X}_t$ and in which, for all $t' \in [t] \setminus \{1\}$, the parent of $x_{t'}$ is $n(x_{t'})$.

It is well known that for any $y : \mathcal{X} \to [K]$, EXP4 attains a $y$-regret of at most $\ln(\hat{w}_1(y))/\eta + \eta KT/2$. By setting $\eta := \rho/\sqrt{KT}$ and noting our choice of $\hat{w}_1$ we obtain our desired regret bound.

## 4.2 Cancellation Propagation

In the remainder of this section we describe our algorithm CBNN, which is based on the same idea as the simple algorithm of Section 4.1.

In this subsection we describe a novel algorithmic framework CANPROP for designing contextual bandit algorithms with a running time logarithmic in $K$. It is inspired by EXP3 [4], specialist algorithms [8] and online decision-tree pruning algorithms [10] but is certainly not a simple combination of these works. CBNN will be an efficient implementation of an instance of CANPROP. Although in general CANPROP requires a-priori knowledge, CBNN is designed in a way that, crucially, does not need it to be known.

We assume, without loss of generality, that $K$ and $T$ are integer powers of two. CANPROP, which takes a parameter $\eta > 0$, works on a full, balanced binary tree $\mathcal{B}$ with leaves $\mathcal{B}^{\star} = [K]$. On every trial $t$ each pair $(v, \mathcal{S}) \in \mathcal{B} \times 2^{\mathcal{X}}$ has a weight $w_t(v, \mathcal{S}) \in [0, 1]$. These weights induce a function $\theta_t : \mathcal{B} \to [0, 1]$ defined by:

$$\theta_t(v) := \sum_{\mathcal{S} \in 2^{\mathcal{X}}} [\![x_t \in \mathcal{S}]\!]w_t(v, \mathcal{S}) \ .$$

On each trial $t$ a root-to-leaf path $\{v_{t,j} \,|\, j \in [\log(K)] \cup \{0\}\}$ is sampled such that $v_{t,0} := r(\mathcal{B})$ and, given $v_{t,j}$, we have that $v_{t,(j+1)}$ is sampled from $\{\triangleleft(v_{t,j}), \triangleright(v_{t,j})\}$ with probability proportional to the value of $\theta_t$ when applied to each of these vertices. The action $a_t$ is then chosen equal to $v_{t,\log(K)}$. Once the loss has been observed we climb back up the root-to-leaf path, updating the function $w_t$ to $w_{t+1}$.

CANPROP (at trial $t$) is given in Algorithm 1. We note that if $w_{t+1}(v, \mathcal{S})$ is not set in the pseudocode then it is defined to be equal to $w_t(v, \mathcal{S})$.

In Appendix B we give a general regret bound for CANPROP. For CBNN we set:

$$\eta := \rho\sqrt{\ln(K)\ln(T)/KT}$$

**Algorithm 1** CANPROP at trial $t$

1: $v_{t,0} \leftarrow r(\mathcal{B})$
2: **for** $j = 0, 1, \cdots, (\log(K) - 1)$ **do**
3:     **for** $v \in \{\lhd(v_{t,j}), \rhd(v_{t,j})\}$ **do**
4:         $\theta_t(v) \leftarrow \sum_{\mathcal{S} \in 2^{\mathcal{X}}} [\![x_t \in \mathcal{S}]\!] w_t(v, \mathcal{S})$
5:     **end for**
6:     $z_{t,j} \leftarrow \theta_t(\lhd(v_{t,j})) + \theta_t(\rhd(v_{t,j}))$
7:     **for** $v \in \{\lhd(v_{t,j}), \rhd(v_{t,j})\}$ **do**
8:         $\pi_t(v) \leftarrow \theta_t(v)/z_{t,j}$
9:     **end for**
10:   $\zeta_{t,j} \sim [0, 1]$
11:   **if** $\zeta_{t,j} \leq \pi_t(\lhd(v_{t,j}))$ **then**
12:     $v_{t,j+1} \leftarrow \lhd(v_{t,j})$
13:   **else**
14:     $v_{t,j+1} \leftarrow \rhd(v_{t,j})$
15:   **end if**
16: **end for**

17: $a_t \leftarrow v_{t,\log(K)}$
18: $\tilde{\pi}_t \leftarrow \prod_{j \in [\log(K)]} \pi_t(v_{t,j})$
19: $\psi_{t,\log(K)} \leftarrow \exp(-\eta \ell_{t,a_t}/\tilde{\pi}_t)$
20: **for** $j = \log(K), (\log(K) - 1), \cdots, 1$ **do**
21:   $\psi_{t,(j-1)} \leftarrow 1 - (1 - \psi_{t,j})\pi_t(v_{t,j})$
22:   $\psi'_{t,j} \leftarrow \psi_{t,j}/\psi_{t,j-1}$
23:   **if** $v_{t,j} = \lhd(v_{t,j-1})$ **then**
24:     $\tilde{v}_{t,j} \leftarrow \rhd(v_{t,j-1})$
25:   **else**
26:     $\tilde{v}_{t,j} \leftarrow \lhd(v_{t,j-1})$
27:   **end if**
28:   **for** $\mathcal{S} \in 2^{\mathcal{X}} : x_t \in \mathcal{S}$ **do**
29:     $w_{t+1}(v_{t,j}, \mathcal{S}) \leftarrow w_t(v_{t,j}, \mathcal{S})\psi'_{t,j}$
30:     $w_{t+1}(\tilde{v}_{t,j}, \mathcal{S}) \leftarrow w_t(\tilde{v}_{t,j}, \mathcal{S})/\psi_{t,j-1}$
31:   **end for**
32: **end for**

and for all $(v, \mathcal{S}) \in \mathcal{B} \times 2^{\mathcal{X}}$ we set:

$$w_1(v, \mathcal{S}) := \frac{1}{4} \prod_{x \in \mathcal{X} \setminus \{x_1\}} \left( \sigma(x, \mathcal{S})\frac{1}{T} + (1 - \sigma(x, \mathcal{S})) \left( 1 - \frac{1}{T} \right) \right) \tag{2}$$

where:

$$\sigma(x, \mathcal{S}) := [\![ [\![x \in \mathcal{S}]\!] \neq [\![n(x) \in \mathcal{S}]\!] ]\!].$$

This choice gives us the regret bound in Theorem 3.3. We note that CBNN will be implemented in such a way that $n$ need not be known a-priori.

### 4.3 Ternary Search Trees

As we shall see, CBNN works by storing a binary tree $\mathcal{A}(v)$ at each vertex $v \in \mathcal{B}$. In order to perform efficient operations on these trees we will utilise the rebalancing data-structure defined in [22] which here we shall call a *ternary search tree* (TST) due to the fact that it is a generalisation of the classic *binary search tree* and, as we shall show, has searching applications. However, as for binary search trees, the applications of TSTs are more than just searching: we shall also utilise them for online belief propagation.

We now define what is meant by a TST. Suppose we have a full binary tree $\mathcal{J}$. A TST of $\mathcal{J}$ is a full ternary tree $\mathcal{D}$ which satisfies the following. The vertex set of $\mathcal{D}$ is partitioned into two sets $\mathcal{D}^{\circ}$ and $\mathcal{D}^{\bullet}$ where each vertex $s \in \mathcal{D}$ is associated with a vertex $\mu(s) \in \mathcal{J}$ and every $s \in \mathcal{D}^{\bullet}$ is also associated with a vertex $\mu'(s) \in \Downarrow(\mu(s))^{\dagger}$. In addition, each internal vertex $s \in \mathcal{D}^{\dagger}$ is associated with a vertex $\xi(s) \in \mathcal{J}$. For all $u \in \mathcal{J}$ there exists an unique leaf $\Upsilon_{\mathcal{D}}(u) \in \mathcal{D}^{\star}$ in which $\mu(\Upsilon_{\mathcal{D}}(u)) = u$.

Essentially, each vertex $s \in \mathcal{D}$ corresponds to a subtree $\hat{\mathcal{J}}(s)$ of $\mathcal{J}$ where $\hat{\mathcal{J}}(r(\mathcal{D})) = \mathcal{J}$. Such a vertex $s$ is a leaf of $\mathcal{D}$ if and only if $|\hat{\mathcal{J}}(s)| = 1$. For each internal vertex $s \in \mathcal{D}^{\dagger}$ the subtree $\hat{\mathcal{J}}(s)$ is *split* at the vertex $\xi(s)$ into the subtrees $\hat{\mathcal{J}}(\lhd(s))$, $\hat{\mathcal{J}}(\nabla(s))$, and $\hat{\mathcal{J}}(\rhd(s))$ corresponding to the children of $s$. The process continues recursively.

For completeness we now describe the rules that a TST $\mathcal{D}$ of $\mathcal{J}$ must satisfy. We have that $r(\mathcal{D}) \in \mathcal{D}^{\circ}$ and $\mu(r(\mathcal{D})) := r(\mathcal{J})$. Each vertex $s \in \mathcal{D}$ represents a subtree $\hat{\mathcal{J}}(s)$ of $\mathcal{J}$. If $s \in \mathcal{D}^{\circ}$ then $\hat{\mathcal{J}}(s) := \Downarrow(\mu(s))$ and otherwise $\hat{\mathcal{J}}(s)$ is the set of all descendants of $\mu(s)$ which are not proper descendants of $\mu'(s)$. Given that $s \in \mathcal{D}^{\dagger}$ this subtree is *split* at the vertex $\xi(s)$ where if $s \in \mathcal{D}^{\bullet}$ we have that $\xi(s)$ lies on the path from $\mu(s)$ to $\mu'(s)$. The children of $s$ are then defined so that $\hat{\mathcal{J}}(\lhd(s)) = \hat{\mathcal{J}}(s) \cap \Downarrow(\lhd(\xi(s)))$ and $\hat{\mathcal{J}}(\rhd(s)) = \hat{\mathcal{J}}(s) \cap \Downarrow(\rhd(\xi(s)))$ and $\hat{\mathcal{J}}(\nabla(s)) = \hat{\mathcal{J}}(s) \setminus (\hat{\mathcal{J}}(\lhd(s)) \cup \hat{\mathcal{J}}(\rhd(s)))$. i.e. $\xi(s)$ partitions the subtree $\hat{\mathcal{J}}(s)$ into the subtrees $\hat{\mathcal{J}}(\lhd(s))$, $\hat{\mathcal{J}}(\rhd(s))$, and $\hat{\mathcal{J}}(\nabla(s))$. This process continues recursively until $|\hat{\mathcal{J}}(s)| = 1$ in which case $s$ is a leaf of $\mathcal{D}$.

For all binary trees $\mathcal{J}$ in our algorithm we shall maintain a TST $\mathcal{H}(\mathcal{J})$ of $\mathcal{J}$ with height $\mathcal{O}(\ln(|\mathcal{J}|))$. Such trees $\mathcal{J}$ are *dynamic* in that on any trial it is possible that two vertices, $u$ and $u'$, are added to the tree $\mathcal{J}$ such that $u'$ is inserted between a non-root vertex of $\mathcal{J}$ and its parent, and $u$ is designated as a child of $u'$. We define the subroutine REBALANCE($\mathcal{H}(\mathcal{J}), u$) as one which rebalances the TST $\mathcal{H}(\mathcal{J})$ after this insertion, so that the height of $\mathcal{H}(\mathcal{J})$ always remains in $\mathcal{O}(\ln(|\mathcal{J}|))$. The work of [22] describes how this subroutine can be implemented in a time of $\mathcal{O}(\ln(|\mathcal{J}|))$ and we refer the reader to this work for details (noting that they use different notation).

## 4.4 Contractions

Define the quantity $\phi_0 := 0$ and for all $j \in \mathbb{N} \cup \{0\}$ inductively define:

$$\phi_{j+1} := \left(1 - \frac{1}{T}\right)\phi_j + \frac{1}{T}(1 - \phi_j).$$

At any trial $t$ the contexts in $\{x_s \mid s \in [t]\}$ naturally form a tree by designating $n(x_s)$ as the parent of $x_s$. However, to utilise the TST data-structure we must only have binary trees. Hence, we will work with a (dynamic) full binary tree $\mathcal{Z}$ which, on trial $t$, is a *binarisation* of the above tree. The relationship between these two trees is given by a map $\gamma : \mathcal{Z}_t \to \{x_s \mid s \in [t]\}$ where $\mathcal{Z}_t$ is the tree $\mathcal{Z}$ on trial $t$. For all $x \in \{x_s \mid s \in [t]\}$ we will always have an unique leaf $\tilde{\gamma}(x) \in \mathcal{Z}_t^\star$ in which $\gamma(\tilde{\gamma}(x)) = x$. We also maintain a balanced TST $\mathcal{H}(\mathcal{Z})$ of $\mathcal{Z}$.

Algorithm 2 gives the subroutine GROW$_t$ which updates $\mathcal{Z}$ at the start of trial $t$. Note that GROW$_t$ also defines a function $d : \mathcal{Z} \to \mathbb{N}$ such that $d(u)$ is the number of times the function $n$ must be applied to $\gamma(u)$ to reach $x_1$.

---

**Algorithm 2** GROW$_t$ which works on $\mathcal{Z}$

---

1: $u \leftarrow \tilde{\gamma}(n(x_t))$
2: $u^* \leftarrow \uparrow(u)$
3: $u' \leftarrow$ NEWVERTEX
4: $u'' \leftarrow$ NEWVERTEX
5: $\gamma(u') \leftarrow n(x_t)$
6: $\gamma(u'') \leftarrow x_t$
7: $\tilde{\gamma}(x_t) \leftarrow u''$
8: **if** $u = \triangleleft(u^*)$ **then**
9: $\quad \triangleleft(u^*) \leftarrow u'$
10: **else**
11: $\quad \triangleright(u^*) \leftarrow u'$
12: **end if**
13: $\triangleleft(u') \leftarrow u''$
14: $\triangleright(u') \leftarrow u$
15: $d(u') \leftarrow d(u)$
16: $d(u'') \leftarrow d(u) + 1$
17: REBALANCE($\mathcal{H}(\mathcal{Z}), u''$)

---

A *contraction* (of $\mathcal{Z}$) is defined as a full binary tree $\mathcal{J}$ in which the following holds. (1) The vertices of $\mathcal{J}$ are a subset of those of $\mathcal{Z}$. (2) $r(\mathcal{J}) = r(\mathcal{Z})$. (3) Given a vertex $u \in \mathcal{J}$ we have $\triangleleft_{\mathcal{J}}(u) \in \Downarrow_{\mathcal{Z}}(\triangleleft_{\mathcal{Z}}(u))$ and $\triangleright_{\mathcal{J}}(u) \in \Downarrow_{\mathcal{Z}}(\triangleright_{\mathcal{Z}}(u))$. (4) Any leaf of $\mathcal{J}$ is a leaf of $\mathcal{Z}$.

CBNN will maintain, on every vertex $v \in \mathcal{B}$, a contraction $\mathcal{A}(v)$ as well as a TST $\mathcal{H}(\mathcal{A}(v))$ of $\mathcal{A}(v)$. Given $\mathcal{J}$ is one of these contractions, we also maintain, for all $i, i' \in \{0, 1\}$ and all $u \in \mathcal{J}$, a value $\tau_{i,i'}(\mathcal{J}, u) \in \mathbb{R}_+$. Technically these quantities, which depend on the above function $d$, define a *bayesian network* on $\mathcal{J}$ which is explained in Appendix C.3. For all $i \in \{0, 1\}$ and all $u \in \mathcal{J}$ we also maintain a value $\kappa_i(\mathcal{J}, u)$ initialised equal to 1.

On each of our contractions $\mathcal{J}$ we will define, on trial $t$, a subroutine INSERT$_t(\mathcal{J})$ that simply modifies $\mathcal{J}$ so that $\tilde{\gamma}(x_t)$ is added to its leaves. This subroutine is only called on certain trials $t$. Specifically, it is called on the contraction $\mathcal{A}(v)$ only when $v$ is involved in CANPROP on trial $t$. Although the effect of this subroutine is simple to describe, its polylogarithmic-time implementation is quite complex. A function that is used many times during this subroutine is $\nu : \mathcal{Z} \times \mathcal{Z} \to \{\blacktriangleleft, \blacktriangleright, \blacktriangle\}$ in which $\nu(u, u')$ is equal to $\blacktriangleleft$, $\blacktriangleright$ or $\blacktriangle$ if $u'$ is contained in $\Downarrow_{\mathcal{Z}}(\triangleleft_{\mathcal{Z}}(u))$, in $\Downarrow_{\mathcal{Z}}(\triangleright_{\mathcal{Z}}(u))$ or in neither, respectively. Algorithm 3 shows how to compute this function. Now that we have a subroutine for computing $\nu$ we can turn to the pseudocode for the subroutine INSERT$_t(\mathcal{J})$ in Algorithm 4. In the appendix we give a full description of how and why this subroutine works.

**Algorithm 3** Computing $\nu(u, u')$ for $u, u' \in \mathcal{Z}$

1: $\mathcal{E} \leftarrow \mathcal{H}(\mathcal{Z})$
2: **if** $u = u'$ **then**
3:     **return** $\blacktriangle$
4: **end if**
5: $\tilde{s} \leftarrow \Upsilon_{\mathcal{E}}(u)$
6: $\tilde{s}' \leftarrow \Upsilon_{\mathcal{E}}(u')$
7: $s^* \leftarrow \Gamma_{\mathcal{E}}(\tilde{s}, \tilde{s}')$
8: **for** $s \in \{\triangleleft(s^*), \triangledown(s^*), \triangleright(s^*)\}$ **do**
9:     **if** $\tilde{s} \in \Downarrow(s)$ **then**
10:         $\hat{s} \leftarrow s$
11:     **end if**
12:     **if** $\tilde{s}' \in \Downarrow(s)$ **then**
13:         $\hat{s}' \leftarrow s$
14:     **end if**
15: **end for**
16: **if** $\hat{s} \neq \triangledown(s^*)$ **then**
17:     **return** $\blacktriangle$
18: **end if**
19: **if** $\xi(s^*) = u \wedge \hat{s}' = \triangleleft(s^*)$ **then**
20:     **return** $\blacktriangleleft$
21: **else if** $\xi(s^*) = u \wedge \hat{s}' = \triangleright(s^*)$ **then**
22:     **return** $\blacktriangleright$
23: **end if**
24: $s \leftarrow \hat{s}$
25: **while** TRUE **do**
26:     **if** $s \in \mathcal{E}^{\circ}$ **then**
27:         **return** $\blacktriangle$
28:     **else if** $u = \xi(s) \wedge \triangleleft(s) \in \mathcal{E}^{\bullet}$ **then**
29:         **return** $\blacktriangleleft$
30:     **else if** $u = \xi(s) \wedge \triangleright(s) \in \mathcal{E}^{\bullet}$ **then**
31:         **return** $\blacktriangleright$
32:     **end if**
33:     **for** $s' \in \{\triangleleft(s), \triangledown(s), \triangleright(s)\}$ **do**
34:         **if** $\tilde{s} \in \Downarrow(s')$ **then**
35:             $s \leftarrow s'$
36:         **end if**
37:     **end for**
38: **end while**

---

**Algorithm 4** The operation $\text{INSERT}_t(\mathcal{J})$ on a contraction $\mathcal{J}$ of $\mathcal{Z}$ at trial $t$

1: $\mathcal{E} \leftarrow \mathcal{H}(\mathcal{Z})$
2: $\mathcal{D} \leftarrow \mathcal{H}(\mathcal{J})$
3: $s \leftarrow r(\mathcal{D})$
4: $u_t \leftarrow \tilde{\gamma}(x_t)$
5: **while** $s \in \mathcal{D}^{\dagger}$ **do**
6:     **if** $\nu(\xi(s), u_t) = \blacktriangleleft$ **then**
7:         $s \leftarrow \triangleleft(s)$
8:     **else if** $\nu(\xi(s), u_t) = \blacktriangleright$ **then**
9:         $s \leftarrow \triangleright(s)$
10:     **else if** $\nu(\xi(s), u_t) = \blacktriangle$ **then**
11:         $s \leftarrow \triangledown(s)$
12:     **end if**
13: **end while**
14: $\hat{u} \leftarrow \mu(s)$
15: $s \leftarrow r(\mathcal{E})$
16: **while** $s \in \mathcal{E}^{\dagger}$ **do**
17:     **if** $\nu(\xi(s), u_t) = \nu(\xi(s), \hat{u})$ **then**
18:         **if** $\nu(\xi(s), u_t) = \blacktriangleleft$ **then**
19:             $s \leftarrow \triangleleft(s)$
20:         **else if** $\nu(\xi(s), u_t) = \blacktriangleright$ **then**
21:             $s \leftarrow \triangleright(s)$
22:         **else if** $\nu(\xi(s), u_t) = \blacktriangle$ **then**
23:             $s \leftarrow \triangledown(s)$
24:         **end if**
25:     **else**
26:         $s \leftarrow \triangledown(s)$
27:     **end if**
28: **end while**
29: $u^* \leftarrow \mu(s)$
30: $u' \leftarrow \uparrow_{\mathcal{J}}(\hat{u})$
31: **if** $\hat{u} = \triangleleft_{\mathcal{J}}(u')$ **then**
32:     $\triangleleft_{\mathcal{J}}(u') \leftarrow u^*$
33: **else**
34:     $\triangleright_{\mathcal{J}}(u') \leftarrow u^*$
35: **end if**
36: **if** $\nu(u^*, \hat{u}) = \blacktriangleleft$ **then**
37:     $\triangleleft_{\mathcal{J}}(u^*) \leftarrow \hat{u}$
38:     $\triangleright_{\mathcal{J}}(u^*) \leftarrow u_t$
39: **else**
40:     $\triangleright_{\mathcal{J}}(u^*) \leftarrow \hat{u}$
41:     $\triangleleft_{\mathcal{J}}(u^*) \leftarrow u_t$
42: **end if**
43: **for** $i \in \{0, 1\}$ **do**
44:     $\kappa_i(\mathcal{J}, u^*) \leftarrow 1$
45:     $\kappa_i(\mathcal{J}, u_t) \leftarrow 1$
46: **end for**
47: **for** $u \in \{u^*, \hat{u}, u_t\}$ **do**
48:     $\delta(u) \leftarrow d(u) - d(\uparrow_{\mathcal{J}}(u))$
49: **end for**
50: **for** $(i, i') \in \{0, 1\} \times \{0, 1\}$ **do**
51:     **if** $i = i'$ **then**
52:         $\tau_{i,i'}(\mathcal{J}, u) \leftarrow 1 - \phi_{\delta(u)}$
53:     **else**
54:         $\tau_{i,i'}(\mathcal{J}, u) \leftarrow \phi_{\delta(u)}$
55:     **end if**
56: **end for**
57: $\text{REBALANCE}(\mathcal{H}(\mathcal{J}), u_t)$

## 4.5 Online Belief Propagation

In this subsection we utilise the work of [7] in order to be able to efficiently compute the function $\theta_t$ that appears in CANPROP.

Given a vertex $u$ in one of our contractions $\mathcal{J}$ we define $\mathcal{F}(\mathcal{J}, u) := \{f \in \{0,1\}^{\mathcal{J}} \mid f(u) = 1\}$ and then define:

$$\Lambda(\mathcal{J}, u) := \sum_{f \in \mathcal{F}(\mathcal{J},u)} \prod_{u' \in \mathcal{J} \setminus \{r(\mathcal{J})\}} \tau_{f(\uparrow_{\mathcal{J}}(u')), f(u')}(\mathcal{J}, u') \kappa_{f(u')}(\mathcal{J}, u').$$

As stated in the previous subsection, when a vertex $v \in \mathcal{B}$ becomes involved in CANPROP on trial $t$, CBNN will add $\tilde{\gamma}(x_t)$ to the leaves of $\mathcal{A}(v)$ via the operation INSERT$_t(\mathcal{A}(v))$. In the appendix we shall show that for each such $v$ we then have:

$$\theta_t(v) = \Lambda(\mathcal{A}(v), \tilde{\gamma}(x_t))/4.$$

We now outline how to compute this efficiently, deferring a full description for Appendix D.3. First note that for all contractions $\mathcal{J}$ and all $u \in \mathcal{J}$ we have that $\Lambda(\mathcal{J}, u)$ is of the exact form to be solved by the classic *Belief propagation* algorithm [23]. The work of [7] shows how to compute this term in logarithmic time by maintaining a data-structure based on a balanced TST of $\mathcal{J}$ - in our case the TST $\mathcal{H}(\mathcal{J})$. Whenever, for some $i \in \{0,1\}$ and $u' \in \mathcal{J}$, the value $\kappa_i(\mathcal{J}, u')$ changes, the data-structure is updated in logarithmic time. We define the subroutine EVIDENCE$(\mathcal{J}, u')$ as that which updates this data-structure after $\kappa_i(\mathcal{J}, u')$ changes. We also make sure that the data-structure is updated whenever REBALANCE$(\mathcal{H}(\mathcal{J}), \cdot)$ is called. We then define the subroutine MARGINAL$(\mathcal{J}, u)$ as that which computes $\Lambda(\mathcal{J}, u)/4$. Hence, the output of MARGINAL$(\mathcal{A}(v), \tilde{\gamma}(x_t))$ is equal to $\theta_t(v)$.

## 4.6 CBNN

Now that we have defined all our subroutines we give, in Algorithm 5, the algorithm CBNN which is an efficient implementation of CANPROP with initial weighting given in Equation (2).

---

**Algorithm 5** CBNN at trial $t$

---

1: GROW$_t$
2: $u_t \leftarrow \tilde{\gamma}(x_t)$
3: $v_{t,0} \leftarrow r(\mathcal{B})$
4: **for** $j = 0, 1, \cdots, (\log(K) - 1)$ **do**
5:     **for** $v \in \{\triangleleft(v_{t,j}), \triangleright(v_{t,j})\}$ **do**
6:         INSERT$_t(\mathcal{A}(v))$
7:         $\theta_t(v) \leftarrow$ MARGINAL$(\mathcal{A}(v), u_t)$
8:     **end for**
9:     $z_{t,j} \leftarrow \theta_t(\triangleleft(v_{t,j})) + \theta_t(\triangleright(v_{t,j}))$
10:     **for** $v \in \{\triangleleft(v_{t,j}), \triangleright(v_{t,j})\}$ **do**
11:         $\pi_t(v) \leftarrow \theta_t(v)/z_{t,j}$
12:     **end for**
13:     $\zeta_{t,j} \sim [0,1]$
14:     **if** $\zeta_{t,j} \leq \pi_t(\triangleleft(v_{t,j}))$ **then**
15:         $v_{t,j+1} \leftarrow \triangleleft(v_{t,j})$
16:     **else**
17:         $v_{t,j+1} \leftarrow \triangleright(v_{t,j})$
18:     **end if**
19: **end for**
20: $a_t \leftarrow v_{t,\log(K)}$
21: $\tilde{\pi}_t \leftarrow \prod_{j \in [\log(K)]} \pi_t(v_{t,j})$
22: $\psi_{t,\log(K)} \leftarrow \exp(-\eta \ell_{t,a_t}/\tilde{\pi}_t)$
23: **for** $j = \log(K), (\log(K) - 1), \cdots, 1$ **do**
24:     $\psi_{t,(j-1)} \leftarrow 1 - (1 - \psi_{t,j})\pi_t(v_{t,j})$
25:     **if** $v_{t,j} = \triangleleft(v_{t,j-1})$ **then**
26:         $\tilde{v}_{t,j} \leftarrow \triangleright(v_{t,j-1})$
27:     **else**
28:         $\tilde{v}_{t,j} \leftarrow \triangleleft(v_{t,j-1})$
29:     **end if**
30:     $\kappa_1(\mathcal{A}(v_{t,j}), u_t) \leftarrow \psi_{t,j}/\psi_{t,j-1}$
31:     $\kappa_1(\mathcal{A}(\tilde{v}_{t,j}), u_t) \leftarrow 1/\psi_{t,j-1}$
32:     EVIDENCE$(\mathcal{A}(v_{t,j}), u_t)$
33:     EVIDENCE$(\mathcal{A}(\tilde{v}_{t,j}), u_t)$
34: **end for**

---

## 5 Acknowledgments

We would like to thank Mark Herbster (University College London) for valuable discussions.

Research funded by the Defence Science and Technology Laboratory (Dstl) which is an executive agency of the UK Ministry of Defence providing world class expertise and delivering cutting-edge science and technology for the benefit of the nation and allies. The research supports the Autonomous Resilient Cyber Defence (ARCD) project within the Dstl Cyber Defence Enhancement programme.

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

# A  Introduction to the Appendix

We now turn to the full description and analysis of our algorithm CBNN. In Appendix B we describe our novel algorithmic framework CANPROP. In Appendix C we describe contractions and bayesian networks on them, showing how CANPROP can be implemented with them. Finally, in Appendix D we describe TSTs and how they are used to perform our required operations efficiently. In Appendix E we prove, in order, all of the theorems stated in this paper.

Recall that we created a sequence of distinct *nodes* $\langle x_t \mid t \in [T] \rangle$ and defined, for all $t \in [T] \setminus \{1\}$, the node $n(x_t) := x_{n(t)}$. Let $\mathcal{X} := \{x_t \mid t \in [T]\}$. Given some $y : \mathcal{X} \to [K]$, we defined the *y-regret* and the *complexity* of $y$ as:

$$R(y) := \sum_{t \in [T]} \ell_{t,a_t} - \sum_{t \in [T]} \ell_{t,y(x_t)} \quad ; \quad \Phi(y) := 1 + \sum_{x \in \mathcal{X} \setminus \{x_1\}} [\![ y(x) \neq y(n(x)) ]\!]$$

respectively.

# B  Cancellation Propagation

## B.1  The General CANPROP Algorithm

We now introduce a general algorithmic framework CANPROP for handling contextual bandit problems with a per-trial time logarithmic in $K$. Without loss of generality assume that $K$ is an integer power of two. Let $\mathcal{B}$ be a full, balanced binary tree whose leaves are the set of actions $[K]$. Let $\mathcal{B}' := \mathcal{B} \setminus \{r(\mathcal{B})\}$. CANPROP takes a parameter $\eta \in \mathbb{R}_+$ called the *learning rate*. On each trial $t$ CANPROP maintains a function:

$$w_t : \mathcal{B}' \times 2^{\mathcal{X}} \to [0, 1] .$$

The function $w_1$ is free to be defined how one likes, as long as it satisfies the constraint that for all internal vertices $v \in \mathcal{B}^{\dagger}$ we have:

$$\sum_{\mathcal{S} \in 2^{\mathcal{X}}} \left( w_1(\lhd(v), \mathcal{S}) + w_1(\rhd(v), \mathcal{S}) \right) = 1 .$$

We now describe how CANPROP acts on trial $t$. For all $v \in \mathcal{B}'$ we define:

$$\theta_t(v) := \sum_{\mathcal{S} \in 2^{\mathcal{X}}} [\![ x_t \in \mathcal{S} ]\!] w_t(v, \mathcal{S})$$

and for all $v \in \mathcal{B}^{\dagger}$ we define:

$$\pi_t(\lhd(v)) := \frac{\theta_t(\lhd(v))}{\theta_t(\lhd(v)) + \theta_t(\rhd(v))} \quad ; \quad \pi_t(\rhd(v)) := \frac{\theta_t(\rhd(v))}{\theta_t(\lhd(v)) + \theta_t(\rhd(v))} .$$

As we shall see CANPROP needs only compute these values for $\mathcal{O}(\ln(K))$ vertices $v$. CANPROP samples a root-to-leaf path $\{v_{t,j} \mid j \in [\log(K)] \cup \{0\}\}$ as follows. $v_{t,0}$ is defined equal to $r(\mathcal{B})$. For all $j \in [\log(K) - 1] \cup \{0\}$, once $v_{t,j}$ has been sampled we sample $v_{t,(j+1)}$ from the probability distribution defined by:

$$\mathbb{P}[v_{t,(j+1)} = v] := [\![ \uparrow(v) = v_{t,j} ]\!] \pi_t(v) \quad \forall v \in \mathcal{B}'$$

noting that $v_{t,(j+1)}$ is a child of $v_{t,j}$. We define:

$$\mathcal{P}_t := \{v_{t,j} \mid j \in [\log(K)] \cup \{0\}\} .$$

CANPROP then selects:

$$a_t := v_{t,\log(K)}$$

and then receives the loss $\ell_{t,a_t}$. The function $w_t$ is then updated to $w_{t+1}$ as follows. Firstly we define,

$$w_{t+1}(v, \mathcal{S}) := w_t(v, \mathcal{S}) \quad \forall (v, \mathcal{S}) \in \{v' \in \mathcal{B}' \mid \uparrow(v') \notin \mathcal{P}_t\} \times 2^{\mathcal{X}} .$$

We then define:

$$\psi_{t,\log(K)} := \exp\left( \frac{-\eta \ell_{t,a_t}}{\prod_{j \in [\log(K)]} \pi_t(v_{t,j})} \right) .$$

Once we have defined $\psi_{t,j}$ for some $j \in [\log(K)]$ we then define:

$$\psi_{t,(j-1)} := 1 - (1 - \psi_{t,j})\pi_t(v_{t,j})$$

$$\beta_t(v) := \frac{[\![v \in \mathcal{P}_t]\!]\psi_{t,j} + [\![v \notin \mathcal{P}_t]\!]}{\psi_{t,(j-1)}} \quad \forall v \in \{\lhd(v_{t,(j-1)}), \rhd(v_{t,(j-1)})\}$$

$$w_{t+1}(v,\mathcal{S}) := ([\![x_t \in \mathcal{S}]\!]\beta_t(v) + [\![x_t \notin \mathcal{S}]\!])w_t(v,\mathcal{S}) \quad \forall (v,\mathcal{S}) \in \{\lhd(v_{t,(j-1)}), \rhd(v_{t,(j-1)})\} \times 2^{\mathcal{X}} \,.$$

The regret bound of CANPROP is given by the following theorem.

**Theorem B.1.** *Suppose we have a function* $y : \mathcal{X} \to [K]$. *For all* $v \in \mathcal{B}$ *define:*

$$\mathcal{Q}(y,v) := \{x \in \mathcal{X} \mid y(x) \in \Downarrow(v)\} \,.$$

*Then the expected $y$-regret of* CANPROP *is bounded by:*

$$\mathbb{E}[R(y)] \leq \frac{\eta K T}{2} - \frac{1}{\eta}\sum_{v \in \mathcal{B}'}[\![\mathcal{Q}(y,v) \neq \emptyset]\!]\ln(w_1(v,\mathcal{Q}(y,v))) \,.$$

## B.2 Our Parameter Tuning

We now describe and analyse the initial weighting $w_1$ and the learning rate $\eta$ that we will use. Define $\mathcal{X}' := \mathcal{X} \setminus \{x_1\}$. For all $(x,\mathcal{S}) \in \mathcal{X}' \times 2^{\mathcal{X}}$ define:

$$\sigma(x,\mathcal{S}) := [\![[\![x \in \mathcal{S}]\!] \neq [\![n(x) \in \mathcal{S}]\!]]\!] \,.$$

For all $(v,\mathcal{S}) \in \mathcal{B}' \times 2^{\mathcal{X}}$ we define:

$$w_1(v,\mathcal{S}) := \frac{1}{4}\prod_{x \in \mathcal{X}'}\left(\sigma(x,\mathcal{S})\frac{1}{T} + (1 - \sigma(x,\mathcal{S}))\left(1 - \frac{1}{T}\right)\right) \,.$$

Given our parameter $\rho$ we choose our learning rate as:

$$\eta := \rho\sqrt{\frac{\ln(K)\ln(T)}{KT}} \,.$$

Given this initial weighting and learning rate, Theorem B.1 implies the following regret bound.

**Theorem B.2.** *Given $w_1$ and $\eta$ are defined as above, then for any $y : \mathcal{X} \to [K]$ the expected $y$-regret of* CANPROP *is bounded by:*

$$\mathbb{E}[R(y)] \in \mathcal{O}\left(\left(\rho + \frac{\Phi(y)}{\rho}\right)\sqrt{\ln(K)\ln(T)KT}\right) \,.$$

## C Implementation with Contractions

### C.1 A Sequence of Binary Trees

For any trial $t$ we have a natural tree-structure on the set $\{x_{t'} \mid t' \in [t]\}$ formed by making $n(x_{t'})$ the parent of $x_{t'}$ for all $t' \in [t]\setminus\{1\}$. However, in order to utilise the methodology of [22] we need to work with binary trees. Hence, we now inductively define a sequence of binary trees $\langle \mathcal{Z}_t \mid t \in [T] \setminus \{1\}\rangle$ where the vertices of $\mathcal{Z}_t$ are a subset of those of $\mathcal{Z}_{t+1}$. We also define a function $\gamma : \mathcal{Z}_T \to \mathcal{X}$. This function $\gamma$ has the property that for any $t \in [T]$ and for any distinct leaves $u, u' \in \mathcal{Z}_t^{\star}$ we have that $\gamma(u) \neq \gamma(u')$, and that:

$$\{\gamma(u'') \mid u'' \in \mathcal{Z}_t^{\star}\} = \{x_{t'} \mid t' \in [t]\} \,.$$

We define $\mathcal{Z}_2$ to contain three vertices $\{r(\mathcal{Z}_2), \lhd(r(\mathcal{Z}_2)), \rhd(r(\mathcal{Z}_2))\}$ where:

$$\gamma(r(\mathcal{Z}_2)) := \gamma(\lhd(r(\mathcal{Z}_2))) := x_1 \quad ; \quad \gamma(\rhd(r(\mathcal{Z}_2))) := x_2 \,.$$

Now consider a trial $t \in [T]$. We have that $\mathcal{Z}_{t+1}$ is constructed from $\mathcal{Z}_t$ via the following algorithm GROW$_{t+1}$:

1. Let $u$ be the unique leaf in $\mathcal{Z}_t^{\star}$ in which $\gamma(u) = n(x_{t+1})$ and let $u^* := \uparrow(u)$.

2. Create two new vertices $u'$ and $u''$.

3. Set $\gamma(u') \leftarrow n(x_{t+1})$ and $\gamma(u'') \leftarrow x_{t+1}$.

4. If $u = \triangleleft(u^*)$ then set $\triangleleft(u^*) \leftarrow u'$. Else set $\triangleright(u^*) \leftarrow u'$.

5. Set $\triangleleft(u') \leftarrow u''$ and $\triangleright(u') \leftarrow u$.

We also define a function $d : \mathcal{Z}_T \to \mathbb{N} \cup \{0\}$ as follows. Define $d'(x_1) := 0$ and for all $t \in [T] \setminus \{1\}$ inductively define $d'(x_t) := d'(n(x_t)) + 1$. Finally define $d(u) := d'(\gamma(u))$ for all $u \in \mathcal{Z}_T$. Since for all $t \in [T]$ we have that the vertices of $\mathcal{Z}_t$ are a subset of those of $\mathcal{Z}_T$ we have that $d$ also defines a function over $\mathcal{Z}_t$ for all $t \in [T]$.

For all $t \in [T]$ we define $u_t$ to be the unique leaf of $\mathcal{Z}_t$ for which $\gamma(u_t) = x_t$.

## C.2 Contractions

Our efficient implementation of CANPROP will have a data-structure at every vertex $v \in \mathcal{B}'$. However, to achieve polylogarithmic time per trial we can only update a polylogarithmic number of these data-structures per trial. This necessitates the use of *contractions* of our trees $\{\mathcal{Z}_t \mid t \in [T] \setminus \{1\}\}$ which are defined as follows. A *contraction* of a full binary tree $\mathcal{Q}$ is another full binary tree $\mathcal{J}$ which satisfies the following:

- The vertices of $\mathcal{J}$ are a subset of those of $\mathcal{Q}$.
- $r(\mathcal{J}) = r(\mathcal{Q})$.
- Given an internal vertex $u \in \mathcal{J}^\dagger$ we have $\triangleleft_{\mathcal{J}}(u) \in \Downarrow_{\mathcal{Q}}(\triangleleft_{\mathcal{Q}}(u))$ and $\triangleright_{\mathcal{J}}(u) \in \Downarrow_{\mathcal{Q}}(\triangleright_{\mathcal{Q}}(u))$.
- Any leaf of $\mathcal{J}$ is a leaf of $\mathcal{Q}$.

Note that any contraction of $\mathcal{Z}_t$ is also a contraction of $\mathcal{Z}_{t+1}$ and hence, by induction, a contraction of $\mathcal{Z}_{t'}$ for all $t' \geq t$. Given a trial $t$ and a contraction $\mathcal{J}$ of $\mathcal{Z}_{t-1}$ we now define the operation $\text{INSERT}_t(\mathcal{J})$ which acts on $\mathcal{J}$ by the following algorithm:

1. Let $\hat{u}$ be the unique vertex in $\mathcal{J} \setminus \{r(\mathcal{J})\}$ such that $u_t$ is in the maximal subtree of $\mathcal{Z}_t$ with $\hat{u}$ and $\uparrow_{\mathcal{J}}(\hat{u})$ as leaves.

2. Let $u^* := \Gamma_{\mathcal{Z}_t}(u_t, \hat{u})$.

3. Add the vertices $u^*$ and $u_t$ to the tree $\mathcal{J}$.

4. Let $u' := \uparrow_{\mathcal{J}}(\hat{u})$.

5. If $\hat{u} = \triangleleft_{\mathcal{J}}(u')$ then set $\triangleleft_{\mathcal{J}}(u') \leftarrow u^*$. Else set $\triangleright_{\mathcal{J}}(u') \leftarrow u^*$.

6. If $\hat{u} \in \Downarrow_{\mathcal{Z}_t}(\triangleleft_{\mathcal{Z}_t}(u^*))$ then set $\triangleleft_{\mathcal{J}}(u^*) \leftarrow \hat{u}$ and $\triangleright_{\mathcal{J}}(u^*) \leftarrow u_t$. Else set $\triangleright_{\mathcal{J}}(u^*) \leftarrow \hat{u}$ and $\triangleleft_{\mathcal{J}}(u^*) \leftarrow u_t$.

Later in this paper we will show how this operation can be done in polylogarithmic time. Note that after the operation we have that $\mathcal{J}$ is a contraction of $\mathcal{Z}_t$ and $u_t$ has been added to it's leaves. From now on when we use the term *contraction* we mean any contraction of $\mathcal{Z}_T$.

## C.3 Contraction-Based Bayesian Networks

Here we shall define a bayesian network over any contraction $\mathcal{J}$ and show how it can be utilised to compute certain quantities required by CANPROP. First define the quantity $\phi_0 := 0$ and for all $j \in \mathbb{N} \cup \{0\}$ inductively define:

$$\phi_{j+1} := \left(1 - \frac{1}{T}\right)\phi_j + \frac{1}{T}(1 - \phi_j).$$

The algorithm must compute these quantities for all $j \in [T]$. However, for all $t \in [T]$ we have that $\phi_t$ doesn't have to be computed until trial $t$ so computing these quantities is constant time per trial. Given a contraction $\mathcal{J}$, a vertex $u \in \mathcal{J} \setminus r(\mathcal{J})$ and indices $i, i' \in \{0, 1\}$ define:

$$\tau_{i,i'}(\mathcal{J}, u) := [\![i \neq i']\!]\phi_{(d(u) - d(\uparrow_{\mathcal{J}}(u)))} + [\![i = i']\!]\left(1 - \phi_{(d(u) - d(\uparrow_{\mathcal{J}}(u)))}\right)$$

which defines the transition matrix from $\uparrow_{\mathcal{J}}(u)$ to $u$ in a bayesian network over $\mathcal{J}$. We shall now show how belief propagation over such bayesian networks can be used to compute the quantities we need in CANPROP. Suppose we have a contraction $\mathcal{J}$ and a function $\lambda : \mathcal{J}^\star \to \mathbb{R}_+$. This function $\lambda$ induces a function $\lambda' : \mathcal{X} \to \mathbb{R}_+$ defined as follows. Given $x \in \mathcal{X}$, if there exists a leaf $u \in \mathcal{J}^\star$ with $\gamma(u) = x$ then $\lambda'(x) = \lambda(u)$. Otherwise $\lambda'(x) = 1$. For all $\mathcal{S} \in 2^{\mathcal{X}}$ define:

$$\tilde{w}(\mathcal{J}, \lambda, \mathcal{S}) := \left( \prod_{x \in \mathcal{S}} \lambda'(x) \right) \left( \prod_{x \in \mathcal{X}'} \left( \sigma(x, \mathcal{S}) \frac{1}{T} + (1 - \sigma(x, \mathcal{S})) \left( 1 - \frac{1}{T} \right) \right) \right).$$

For the CANPROP algorithm we will need to compute

$$\sum_{\mathcal{S} \in 2^{\mathcal{X}}} [\![\gamma(\hat{u}) \in \mathcal{S}]\!] \tilde{w}(\mathcal{J}, \lambda, \mathcal{S}) \tag{3}$$

for some leaf $\hat{u} \in \mathcal{J}^\star$ and some function $\lambda : \mathcal{J}^\star \to \mathbb{R}_+$. We shall now show how we can compute this quantity via belief propagation on the bayesian network. In particular, we shall construct a quantity $\tilde{\Lambda}(\mathcal{J}, \lambda, u)$ equal to the quantity in Equation (3). To do this, first define the function $\lambda^* : \mathcal{J} \to \mathbb{R}_+$ so that for all $u \in \mathcal{J}^\star$ we have $\lambda^*(u) = \lambda(u)$ and for all $u \in \mathcal{J}^\dagger$ we have $\lambda^*(u) = 1$. For all vertices $u \in \mathcal{J}$ and all indices $i \in \{0, 1\}$ define:

$$\tilde{\kappa}_i(\mathcal{J}, \lambda, u) := [\![i = 0]\!] + [\![i = 1]\!] \lambda^*(u).$$

For all $\hat{u} \in \mathcal{J}$ define:

$$\mathcal{F}(\mathcal{J}, \hat{u}) := \{ f \in \{0, 1\}^{\mathcal{J}} \mid f(\hat{u}) = 1 \}$$

and then define:

$$\tilde{\Lambda}(\mathcal{J}, \lambda, \hat{u}) := \sum_{f \in \mathcal{F}(\mathcal{J}, \hat{u})} \prod_{u \in \mathcal{J} \backslash r(\mathcal{J})} \tau_{f(\uparrow_{\mathcal{J}}(u)), f(u)}(\mathcal{J}, u) \tilde{\kappa}_{f(u)}(\mathcal{J}, \lambda, u).$$

The equality of this quantity and that given in Equation (3) is given by the following theorem.

**Theorem C.1.** *Given a contraction $\mathcal{J}$, a function $\lambda : \mathcal{J}^\star \to \mathbb{R}_+$ and some leaf $\hat{u} \in \mathcal{J}^\star$ we have:*

$$\tilde{\Lambda}(\mathcal{J}, \lambda, \hat{u}) = \sum_{\mathcal{S} \in 2^{\mathcal{X}}} [\![\gamma(\hat{u}) \in \mathcal{S}]\!] \tilde{w}(\mathcal{J}, \lambda, \mathcal{S}).$$

Note that $\tilde{\Lambda}(\mathcal{J}, \lambda, \hat{u})$ is of the exact form to be solved via belief propagation over $\mathcal{J}$. However, belief propagation is too slow (taking $\Theta(|\mathcal{J}|)$ time) - we will remedy this later.

### C.4  Cancelation Propagation with Contractions

We now describe how to implement CANPROP with contractions. For each $v \in \mathcal{B}'$ we maintain a contraction $\mathcal{A}(v)$ and a function $\lambda_v : \mathcal{A}(v)^\star \to \mathbb{R}_+$. We initialise with $\mathcal{A}(v)$ identical to $\mathcal{Z}_2$ and $\lambda_v(u) = 1$ for both leaves $u \in \mathcal{Z}_2^\star$. Via induction over $t$ we will have that at the start of each trial $t$ we have, for all sets $\mathcal{S} \in 2^{\mathcal{X}}$, that:

$$w_t(v, \mathcal{S}) = \tilde{w}(\mathcal{A}(v), \lambda_v, \mathcal{S})/4. \tag{4}$$

On trial $t$ we do as follows. First we update $\mathcal{Z}_{t-1}$ to $\mathcal{Z}_t$ using the algorithm GROW$_t$. We will perform the necessary modifications to our contractions as we sample the path $\mathcal{P}_t$. In particular, we first set $v_{t,0} := r(\mathcal{B})$ and then for each $j \in [\log(K) - 1] \cup \{0\}$ in turn we do as follows. For each $v \in \{\triangleleft(v_{t,j}), \triangleright(v_{t,j})\}$ run INSERT$_t(\mathcal{A}(v))$ and set $\lambda_v(u_t) \leftarrow 1$. Since $\lambda_v(u_t) = 1$ Equation (4) still holds and hence, by Theorem C.1, we have:

$$\theta_t(v) = \tilde{\Lambda}(\mathcal{A}(v), \lambda_v, u_t)/4.$$

After $\theta_t(v)$ has been computed for both $v \in \{\triangleleft(v_{t,j}), \triangleright(v_{t,j})\}$ we can now sample $v_{t,j+1}$.

Once we have selected the action $a_t$ we then update the functions $\{\lambda_v \mid \uparrow_{\mathcal{B}}(v) \in \mathcal{P}_t\}$ by setting $\lambda_v(u_t) \leftarrow \beta_t(v)$ for all $v \in \mathcal{B}'$ with $\uparrow_{\mathcal{B}}(v) \in \mathcal{P}_t$. It is clear now that Equation (4) holds inductively.

## C.5 Notational Relationship to the Main Body

We now point out how the notation in this section relates to that of the main body. In particular, we have, for all $v \in \mathcal{B}'$, all $u \in \mathcal{A}(v)$ and all $i, i' \in \{0, 1\}$, that:

- $\kappa_i(\mathcal{A}(v), u) = \tilde{\kappa}_i(\mathcal{A}(v), \lambda_v, u)$.
- $\Lambda(\mathcal{A}(v), u) = \tilde{\Lambda}(\mathcal{A}(v), \lambda_v, u)$.

We note that the function $\lambda_v$ does not explicity appear in our pseudocode since it can be inferred from $\kappa_i(\mathcal{A}(v), \cdot)$.

## D  Utilising Ternary Search Trees

There are now only two things left to do in order to achieve polylogarithmic time per trial - to make an efficient online implementation of the $\text{INSERT}_t(\cdot)$ operation and an efficient online algorithm to perform belief propagation over our contractions. In order to do this we will utilise the methodology of [22] which we now describe. However, we do not give the full details of the rebalancing technique and refer the reader to [22] for these details (noting that [22] uses different notation).

### D.1  Ternary Search Trees

In this section we will consider a full binary tree $\mathcal{J}$. A *(full) ternary tree* $\mathcal{D}$ is a rooted tree in which each internal vertex $s \in \mathcal{D}^\dagger$ has three children denoted by $\lhd(s), \triangledown(s), \rhd(s)$ and called the *left*, *centre*, and *right* children respectively. We now define what it means for a ternary tree $\mathcal{D}$ to be a ternary search tree (TST) of $\mathcal{J}$. Firstly, the vertex set of $\mathcal{D}$ is partitioned into two sets $\mathcal{D}^\circ$ and $\mathcal{D}^\bullet$. Every vertex $s \in \mathcal{D}$ is associated with a vertex $\mu(s) \in \mathcal{J}$ and every $s \in \mathcal{D}^\bullet$ is also associated with a vertex $\mu'(s) \in \Downarrow_\mathcal{J}(\mu(s))^\dagger$. The root $r(\mathcal{D})$ is contained in $\mathcal{D}^\circ$ and $\mu(r(\mathcal{D})) := r(\mathcal{J})$. Each internal vertex $s \in \mathcal{D}^\dagger$ is associated with a vertex $\xi(s) \in \mathcal{J}$. If $s \in \mathcal{D}^\circ$ then $\xi(s) \in \Downarrow(\mu(s))^\dagger$ and if $s \in \mathcal{D}^\bullet$ then $\xi(s)$ lies on the path (in $\mathcal{J}$) from $\mu(s)$ to $\uparrow(\mu'(s))$. For all $s \in \mathcal{D}^\dagger$ we have:

- $\triangledown(s) \in \mathcal{D}^\bullet$, $\mu(\triangledown(s)) := \mu(s)$ and $\mu'(\triangledown(s)) := \xi(s)$.
- $\lhd(s)$ satisfies:
    - If $s \in \mathcal{D}^\circ$ then $\lhd(s) \in \mathcal{D}^\circ$ and $\mu(\lhd(s)) := \lhd(\xi(s))$.
    - If $s \in \mathcal{D}^\bullet$ and $\mu'(s) \in \Downarrow(\rhd(\xi(s)))$ then $\lhd(s) \in \mathcal{D}^\circ$ and $\mu(\lhd(s)) := \lhd(\xi(s))$.
    - Else $\lhd(s) \in \mathcal{D}^\bullet$, $\mu(\lhd(s)) := \lhd(\xi(s))$ and $\mu'(\lhd(s)) := \mu'(s)$.
- $\rhd(s)$ satisfies:
    - If $s \in \mathcal{D}^\circ$ then $\rhd(s) \in \mathcal{D}^\circ$ and $\mu(\rhd(s)) := \rhd(\xi(s))$.
    - If $s \in \mathcal{D}^\bullet$ and $\mu'(s) \in \Downarrow(\lhd(\xi(s)))$ then $\rhd(s) \in \mathcal{D}^\circ$ and $\mu(\rhd(s)) := \rhd(\xi(s))$.
    - Else $\rhd(s) \in \mathcal{D}^\bullet$, $\mu(\rhd(s)) := \rhd(\xi(s))$ and $\mu'(\rhd(s)) := \mu'(s)$.

Finally, for each leaf $s \in \mathcal{D}^\star$ we have:

- If $s \in \mathcal{D}^\circ$ then $\mu(s)$ is a leaf of $\mathcal{J}$.
- If $s \in \mathcal{D}^\bullet$ then there exists $u \in \mathcal{J}^\dagger$ such that $\mu(s) = \mu'(s) = u$.

Intuitively, each vertex $s \in \mathcal{D}$ is associated with a subtree $\hat{\mathcal{J}}(s)$ of $\mathcal{J}$. If $s \in \mathcal{D}^\circ$ then $\hat{\mathcal{J}}(s) := \Downarrow(\mu(s))$ and if $s \in \mathcal{D}^\bullet$ then $\hat{\mathcal{J}}(s)$ is the subtree of all descendants of $\mu(s)$ which are not proper descendants of $\mu'(s)$. For every $s \in \mathcal{D}$ such that $\hat{\mathcal{J}}(s)$ contains only a single vertex, we have that $s$ is a leaf of $\mathcal{D}$. Otherwise $s$ is an internal vertex of $\mathcal{D}$ and its children are as follows. We say that $\hat{\mathcal{J}}(s)$ is *split* at the vertex $\xi(s) \in \hat{\mathcal{J}}(s)^\dagger$. If $s \in \mathcal{D}^\bullet$ we require that $\xi(s)$ is on the path in $\mathcal{J}$ from $\mu(s)$ to $\mu'(s)$. The action of splitting $\hat{\mathcal{J}}(s)$ at $\xi(s)$ partitions $\hat{\mathcal{J}}(s)$ into the subtrees $\hat{\mathcal{J}}(\lhd(s))$, $\hat{\mathcal{J}}(\triangledown(s))$ and $\hat{\mathcal{J}}(\rhd(s))$ defined as follows:

- $\hat{\mathcal{J}}(\lhd(s)) := \Downarrow(\lhd(\xi(s))) \cap \hat{\mathcal{J}}(s)$.
- $\hat{\mathcal{J}}(\rhd(s)) := \Downarrow(\rhd(\xi(s))) \cap \hat{\mathcal{J}}(s)$.

- $\hat{\mathcal{J}}(\triangledown(s)) := \hat{\mathcal{J}}(s) \setminus (\hat{\mathcal{J}}(\triangleleft(s)) \cup \hat{\mathcal{J}}(\triangleright(s)))$.

Utilising the methodology of [22] we will maintain TSTs of $\mathcal{Z}_t$ (at each trial $t$) and the trees in $\{\mathcal{A}(v) \mid v \in \mathcal{B}'\}$, each with height $\mathcal{O}(\ln(T))$. Note that these trees are dynamic, in that vertices are inserted into them over time. [22] shows how, after such an insertion, the corresponding TST can be *rebalanced*, in time $\mathcal{O}(\ln(T))$, so that its height is still in $\mathcal{O}(\ln(T))$. This rebalancing is performed via a sequence of $\mathcal{O}(\ln(T))$ *tree rotations*, which generalise the concept of tree rotations in binary search trees.

## D.2 Searching

In this section we show how we can use our TSTs to implement the operation $\text{INSERT}_t(\mathcal{J})$ on any trial $t$ and contraction $\mathcal{J}$ of $\mathcal{Z}_{t-1}$. To do this we need to perform the following two search operations:

1. Find the unique vertex $\hat{u} \in \mathcal{J} \setminus \{r(\mathcal{J})\}$ such that $u_t$ is in the maximal subtree of $\mathcal{Z}_t$ with $\hat{u}$ and $\uparrow_{\mathcal{J}}(\hat{u})$ as leaves.
2. Find $u^* := \Gamma_{\mathcal{Z}_t}(u_t, \hat{u})$.

To perform these tasks in polylogarithmic time we will utilise TSTs $\mathcal{E}$ and $\mathcal{D}$ of $\mathcal{Z}_t$ and $\mathcal{J}$ respectively. Both the searching tasks utilise a function $\nu : \mathcal{Z}_t^2 \to \{\blacktriangle, \blacktriangleleft, \blacktriangleright\}$ defined, for all $u, u' \in \mathcal{Z}_t$ as follows. If $u' \in \Downarrow_{\mathcal{Z}_t}(\triangleleft(u))$ or $u' \in \Downarrow_{\mathcal{Z}_t}(\triangleright(u))$ then $\nu(u, u') := \blacktriangleleft$ or $\nu(u, u') := \blacktriangleright$ respectively. Otherwise $\nu(u, u') := \blacktriangle$. This can be computed as follows. If $u = u'$ then $\nu(u, u') := \blacktriangle$. Otherwise let $\tilde{s}$ and $\tilde{s}'$ be the unique leaves of $\mathcal{E}$ such that $\mu(\tilde{s}) = u$ and $\mu(\tilde{s}') = u'$. Let $s^* := \Gamma_{\mathcal{E}}(\tilde{s}, \tilde{s}')$ and let $\hat{s}$ and $\hat{s}'$ be the children of $s^*$ which are ancestors of $\tilde{s}$ and $\tilde{s}'$ respectively. If $\hat{s} \neq \triangledown(s^*)$ then we have $\nu(u, u') := \blacktriangle$. If $\xi(s^*) = u$ then we have $\nu(u, u') := \blacktriangleleft$ or $\nu(u, u') := \blacktriangleright$ if $\hat{s}' = \triangleleft(s^*)$ or $\hat{s}' = \triangleright(s^*)$ respectively. If $\hat{s} = \triangledown(s^*)$ and $\xi(s^*) \neq u$ then we perform the following process. Start with $s$ equal to $\hat{s}$. At any point in the process we do as follows. If $s \in \mathcal{E}^\circ$ then the process terminates with $\nu(u, u') := \blacktriangle$. If $s \in \mathcal{E}^\bullet$ and $u = \xi(s)$ then the process terminates with $\nu(u, u') := \blacktriangleleft$ or $\nu(u, u') := \blacktriangleright$ if $\triangleleft(s) \in \mathcal{E}^\bullet$ or $\triangleright(s) \in \mathcal{E}^\bullet$ respectively. If $s \in \mathcal{E}^\bullet$ and $u \neq \xi(s)$ then we reset $s$ as equal to the child of $s$ which is an ancestor of $\tilde{s}$ and continue the process.

The vertex $\hat{u}$ can be found as follows. We construct a root-to-leaf path in $\mathcal{D}$ such that, given a vertex $s$ in the path, the next vertex in the path is $\triangleleft(s)$, $\triangleright(s)$ or $\triangledown(s)$ if $\nu(\xi(s), u_t)$ is equal to $\blacktriangleleft$, $\blacktriangleright$ or $\blacktriangle$ respectively. Given that $s'$ is the leaf of $\mathcal{D}$ that is in this path we have $\hat{u} = \mu(s')$.

The vertex $u^*$ can then be found as follows. We construct a root-to-leaf path in $\mathcal{E}$ such that, given a vertex $s$ in the path, the next vertex in the path is found as follows. If $\nu(\xi(s), u_t) = \nu(\xi(s), \hat{u})$ then given $\nu(\xi(s), u_t)$ is equal to $\blacktriangleleft$, $\blacktriangleright$ or $\blacktriangle$, the next vertex is equal to $\triangleleft(s)$, $\triangleright(s)$ or $\triangledown(s)$ respectively. Otherwise, the next vertex is $\triangledown(s)$. Given that $s'$ is the leaf of $\mathcal{E}$ that is in this path we have $u^* = \mu(s')$.

The fact that these algorithms find the correct vertices is given in the following theorem:

**Theorem D.1.** *The above algorithms are correct.*

## D.3 Belief Propagation

Here we utilise the methodology of [7] in order to efficiently compute the function $\tilde{\Lambda}$ that appears in the CANPROP implementation. i.e. given a contraction $\mathcal{J}$, a function $\lambda : \mathcal{J}^* \to \mathbb{R}_+$ and some leaf $\hat{u} \in \mathcal{J}^*$ we need to compute $\tilde{\Lambda}(\mathcal{J}, \lambda, \hat{u})$. For brevity let us define, for all $i, i' \in \{0, 1\}$, and all vertices $u \in \mathcal{J} \setminus \{r(\mathcal{J})\}$, the quantities:

$$\hat{\tau}_{i,i'}(u) := \tau_{i,i'}(\mathcal{J}, u) \quad ; \quad \hat{\kappa}_i(u) := \tilde{\kappa}_i(\mathcal{J}, \lambda, u).$$

For simplicity of presentation we will utilise a tree $\mathcal{J}'$ which is defined as identical to $\mathcal{J}$ except with a single vertex added as the parent of $r(\mathcal{J})$. For all $i, i' \in \{0, 1\}$ we define $\hat{\kappa}_i(r(\mathcal{J}')) := 1$ and $\hat{\tau}_{i,i'}(r(\mathcal{J})) = [\![i = i']\!]$. For all $u \in \mathcal{J}$ we will define $\uparrow(u) := \uparrow_{\mathcal{J}'}(u)$

We will utilise a TST $\mathcal{D}$ of $\mathcal{J}$ by maintaining *potentials* on the vertices of $\mathcal{D}$ defined as follows. First, as in Section D.1, for any vertex $s \in \mathcal{D}$ we define the subtree $\hat{\mathcal{J}}(s)$ of $\mathcal{J}$ to be equal to $\Downarrow_{\mathcal{J}}(\mu(s))$ if $s \in \mathcal{D}^\circ$ and equal to the subtree in $\mathcal{J}$ of all descendants of $\mu(s)$ which are not proper descendants of $\mu'(s)$ if $s \in \mathcal{D}^\bullet$. For all $s \in \mathcal{D}^\circ$ and $i \in \{0, 1\}$ we define:

$$\Psi_i(s) := \sum_{f \in \{0,1\}^{\hat{\mathcal{J}}(s) \cup \{\uparrow(\mu(s))\}}} [\![f(\uparrow(\mu(s))) = i]\!] \prod_{u \in \hat{\mathcal{J}}(s)} \hat{\tau}_{f(\uparrow(u)), f(u)}(u) \hat{\kappa}_{f(u)}(u)$$

and for all $s \in \mathcal{D}^\bullet$ and $i, i' \in \{0, 1\}$ we define:

$$\Omega_{i,i'}(s) := \sum_{f \in \{0,1\}^{\hat{\mathcal{J}}(s) \cup \{\uparrow(\mu(s))\}}} \llbracket f(\uparrow(\mu(s))) = i \rrbracket \llbracket f(\mu'(s)) = i' \rrbracket \prod_{u \in \hat{\mathcal{J}}(s)} \hat{\tau}_{f(\uparrow(u)), f(u)}(u) \hat{\kappa}_{f(u)}(u) .$$

We have the following recurrence relations for these potentials. Suppose we have an internal vertex $s \in \mathcal{D}^\dagger$ and $i, i' \in \{0, 1\}$. If $s \in \mathcal{D}^\circ$ we have:

$$\Psi_i(s) = \sum_{i'' \in \{0,1\}} \Omega_{i,i''}(\nabla(s)) \Psi_{i''}(\triangleleft(s)) \Psi_{i''}(\triangleright(s)) .$$

If, instead, $s \in \mathcal{D}^\bullet$ then, by letting $s' := \triangleleft(s)$, $s'' := \triangleright(s)$ if $\triangleleft(s) \in \mathcal{D}^\bullet$ and $s' := \triangleright(s)$, $s'' := \triangleleft(s)$ otherwise, we have:

$$\Omega_{i,i'}(s) = \sum_{i'' \in \{0,1\}} \Omega_{i,i''}(\nabla(s)) \Omega_{i'',i'}(s') \Psi_{i''}(s'') .$$

If, on a trial $t$, we perform the operation $\text{INSERT}_t(\mathcal{J})$ or change the value of $\lambda(u_t)$ these recurrence relations can be used (in conjunction with the tree rotations) to update the potentials in logarithmic time.

Now that we have defined our potentials we will show how to use them to compute $\tilde{\Lambda}(\mathcal{J}, \lambda, \hat{u})$ in logarithmic time. To do this we recursively define the following quantities for $i \in \{0, 1\}$. Let $\omega_i(r(\mathcal{D})) := 1$. Given an internal vertex $s \in \mathcal{D}^\circ$ we define:

$$\omega_i(\nabla(s)) := \omega_i(s) \quad ; \quad \omega_i'(\nabla(s)) := \Psi_i(\triangleleft(s)) \Psi_i(\triangleright(s))$$

$$\omega_i(\triangleleft(s)) := \Psi_i(\triangleright(s)) \sum_{i' \in \{0,1\}} \omega_{i'}(s) \Omega_{i',i}(\nabla(s)) \quad ; \quad \omega_i(\triangleright(s)) := \Psi_i(\triangleleft(s)) \sum_{i' \in \{0,1\}} \omega_{i'}(s) \Omega_{i',i}(\nabla(s)) .$$

Given an internal vertex $s \in \mathcal{D}^\bullet$ define $s' := \triangleleft(s)$, $s'' := \triangleright(s)$ if $\triangleleft(s) \in \mathcal{D}^\bullet$ and $s' := \triangleright(s)$, $s'' := \triangleleft(s)$ otherwise. Then:

$$\omega_i(\nabla(s)) := \omega_i(s) \quad ; \quad \omega_i'(\nabla(s)) := \Psi_i(s'') \sum_{i' \in \{0,1\}} \Omega_{i,i'}(s') \omega_{i'}'(s)$$

$$\omega_i(s') := \sum_{i' \in \{0,1\}} \omega_{i'}(s) \Omega_{i',i}(s) \Psi_i(s'') \quad ; \quad \omega_i'(s') := \omega_i'(s)$$

$$\omega_i(s'') := \sum_{i',i'' \in \{0,1\}} \omega_{i'}(s) \Omega_{i',i}(\nabla(s)) \omega_{i''}'(s) \Omega_{i,i''}(s') .$$

For $s \in \mathcal{D}^\circ$, $\omega_i'(s)$ is not required and hence is arbitrary. We inductively compute the values $\{\omega_i(s), \omega_i'(s) \mid i \in \{0, 1\}\}$ for all $s$ in the path from $r(\mathcal{D})$ to the unique leaf $\hat{s} \in \mathcal{D}^\star$ in which $\mu(\hat{s}) = \hat{u}$. We then have $\tilde{\Lambda}(\mathcal{J}, \lambda, \hat{u}) = \omega_1(\hat{s})$.

Since this is known methodology we do not include a proof in this paper and direct the reader to [7].

# E   Proofs

## E.1   Theorem 3.3

This theorem is proved in appendices B to D and the theorems therein.

## E.2   Theorem 3.6

For all $x \in \mathcal{C}$ define $\hat{\gamma}(x) := \gamma(x, \hat{y}, \mathcal{X})$.

Choose a set $\mathcal{S} \subseteq \mathcal{C}$ in which for all $t \in [T]$ there exists $x \in \mathcal{S}$ with $\Delta(x, x_t) < \hat{\gamma}(x)/3c$. For all trials $t$ let $\mathcal{S}_t$ be the set of all contexts $x \in \mathcal{S}$ in which there exists $s \in [t]$ with $\Delta(x, x_s) < \hat{\gamma}(x)/3c$.

Now consider a trial $t \in [T] \setminus \{1\}$ in which $\hat{y}(x_t) \neq \hat{y}(x_{n(t)})$ and choose $x \in \mathcal{S}$ with $\Delta(x, x_t) < \hat{\gamma}(x)/3c$.

Assume, for contradiction, that $x \in \mathcal{S}_{t-1}$. Then there exists $s \in [t-1]$ with $\Delta(x, x_s) < \hat{\gamma}(x)/3c$ so that by the triangle inequality we have:

$$\Delta(x_t, x_s) \leq \Delta(x, x_s) + \Delta(x, x_t) < 2\hat{\gamma}(x)/3c$$

which implies that $\Delta(x_t, x_{n(t)}) < 2\hat{\gamma}(x)/3$. By the triangle inequality we then have that:

$$\Delta(x, x_{n(t)}) \leq \Delta(x_t, x_{n(t)}) + \Delta(x, x_t) < 2\hat{\gamma}(x)/3 + \hat{\gamma}(x)/3c \leq 3\hat{\gamma}(x)/3 = \hat{\gamma}(x)$$

Since $\Delta(x, x_t) < \hat{\gamma}(x)$ we have $y(x) = y(x_t)$ and hence that $y(x) \neq y(x_{n(t)})$. But this contradicts the fact that $\Delta(x, x_{n(t)}) < \hat{\gamma}(x)$.

We have hence shown that $x \notin \mathcal{S}_{t-1}$. Since $x \in \mathcal{S}_t$ we then have that $|\mathcal{S}_t| \geq |\mathcal{S}_{t-1}|$. Since $|\mathcal{S}_1| \geq 1$ this implies that:

$$\Phi(\boldsymbol{y}) = 1 + \sum_{t \in [T] \setminus \{1\}} [\![\hat{y}(x_t) \neq \hat{y}(x_{n(t)})]\!] \leq |\mathcal{S}_T| \leq |\mathcal{S}|$$

as required.

### E.3 Theorem 3.9

Let $\epsilon$ be such that:

$$\lim_{\delta \to 0} \frac{1}{\delta} \int_{x \in \mathcal{M}(\hat{y}, \mu, 2\epsilon, \delta)} 1 \leq 2\alpha(\hat{y}, \mu) \qquad ; \qquad \lim_{\delta \to 0} \frac{1}{\delta} \int_{x \in \mathcal{M}(\hat{y}, \mu, 2\epsilon, \delta)} \mu(x) \leq 2\tilde{\alpha}(\hat{y}, \mu) \qquad (5)$$

Since we are only interested in the behaviour as $q \to \infty$ assume, without loss of generality, that $\sqrt{d}/q < \epsilon/3$. Choose $\lambda > 0$ and $\lambda' > 0$ sufficiently small for this proof to work. For all $x \in \mathcal{G}_q^d$ let $\mathcal{H}(x)$ be the set of points $x' \in [0,1]^d$ such that $x$ is the nearest neighbour (or one of them if the nearest neighbour is not unique) of $x'$ in $\mathcal{G}_q^d$. Let $\mathcal{L}$ be the set of all $x \in \mathcal{G}_q^d$ for which there exists $x', x'' \in \mathcal{H}(x) \cap \mathcal{E}(\mu, \epsilon)$ with $\hat{y}(x') \neq \hat{y}(x'')$. Note that for all $x' \in \mathcal{L}$ we have $\mathcal{H}(x') \subseteq \mathcal{M}(\hat{y}, \mu, 2\epsilon, \sqrt{d}/q)$ so:

$$\frac{q}{\sqrt{d}} \int_{x \in \mathcal{M}(\hat{y}, \mu, 2\epsilon, \sqrt{d}/q)} 1 \geq \frac{q}{\sqrt{d}} \sum_{x' \in \mathcal{L}} \int_{x \in \mathcal{H}(x')} 1 = \frac{q}{\sqrt{d}} |\mathcal{L}| q^{-d} = \frac{1}{q^{d-1}\sqrt{d}} |\mathcal{L}|$$

By considering the limit of this inequality as $q \to \infty$, and noting that $d$ is being treated as a constant, we then have, by Equation (5), that:

$$|\mathcal{L}| \in \mathcal{O}(\alpha(\hat{y}, \mu) q^{d-1}) \qquad (6)$$

Now define:

$$p := \sum_{x' \in \mathcal{L}} \int_{x \in \mathcal{H}(x')} \mu(x)$$

Since $\mathcal{H}(x') \subseteq \mathcal{M}(\hat{y}, \mu, 2\epsilon, \sqrt{d}/q)$ for all $x' \in \mathcal{L}$, we have:

$$\frac{q}{\sqrt{d}} p \leq \frac{q}{\sqrt{d}} \int_{x \in \mathcal{M}(\hat{y}, \mu, 2\epsilon, \sqrt{d}/q)} \mu(x)$$

By considering the limit of this inequality as $q \to \infty$, and noting that $d$ is being treated as a constant, we then have, by Equation (5), that:

$$p \in \mathcal{O}(\tilde{\alpha}(\hat{y}, \mu)/q) \qquad (7)$$

Let $\mathcal{D}$ be the set of all $x \in \mathcal{G}_q^d$ such that there exists $x' \in \mathcal{H}(x)$ with $\mu(x) \neq 0$. Now define $\hat{y}' : [0,1]^d \to [K]$ such that for all $x \in [0,1]^d$ we have:

- If $x \in \mathcal{D} \setminus \mathcal{L}$ then $\hat{y}'(x)$ is the unique $a \in [K]$ such that there exists $x' \in \mathcal{H}(x)$ with $\mu(x') \neq 0$ and $a = \hat{y}(x')$. Note that if there existed more than one such $a$ then we would have $x \in \mathcal{L}$ which is a contradiction.
- If $x \notin \mathcal{D} \setminus \mathcal{L}$ then $\hat{y}'(x) = \hat{y}'(\hat{x})$ where $\hat{x}$ is the nearest neighbour of $x$ in $\mathcal{D} \setminus \mathcal{L}$

Let $\mathcal{A}$ be a finite set of points in $\mathbb{R}^d$ such that for all $x \in \mathbb{R}^d$ with $\Delta(x,0) \leq 1$ there exists $x' \in \mathcal{A}$ with $\Delta(x,x') < \lambda$. Define:

$$\mathcal{A}' := \bigcup_{i \in [[\lceil \log(qd) \rceil]]} \{2^i x/q \,|\, x \in \mathcal{A}\}$$

Note that:

$$|\mathcal{A}'| \in \mathcal{O}(\ln(q)) \tag{8}$$

We now show that for all $x \in \mathbb{R}^d$ with $1/q \leq \Delta(x,0) \leq d$ there exists $x'' \in \mathcal{A}'$ with:

$$\Delta(x,x'') < 2\Delta(x,0)\lambda \tag{9}$$

To show this choose any such $x$ and let $i \in \mathbb{N}$ be such that $2^{i-1}/q \leq \Delta(x,0) < 2^i/q$. By the assumption on $x$ we have that $i \in [[\lceil \log(qd) \rceil]]$. Since $\Delta(xq2^{-i}, 0) < 1$ choose $x' \in \mathcal{A}$ such that $\Delta(xq2^{-i}, x') < \lambda$. Then we have:

$$\Delta(x, 2^i x'/q) = (2^i/q)\Delta(xq2^{-i}, x') < (2^i/q)\lambda \leq 2\Delta(x,0)\lambda$$

so, since $2^i x'/q \in \mathcal{A}'$, we have proved that Equation (9) is true.

We now let $\mathcal{S}'$ be a finite set of points in $\mathbb{R}^d$ such that for all $x \in [0,1]^d$ there exists some $x' \in \mathcal{S}'$ with $\Delta(x,x') < \lambda'\epsilon$. Now define:

$$\mathcal{S} := \mathcal{L} \cup \mathcal{S}' \cup \bigcup_{x \in \mathcal{L}} \{x' + x \,|\, x' \in \mathcal{A}'\}$$

Let $\mathcal{X} := \{x_t \,|\, t \in [T]\}$. Take any $x \in \mathcal{X}$. We now show that there exists $x^\dagger \in \mathcal{S}$ with $\Delta(x, x^\dagger) < \gamma(x^\dagger, \hat{y}', \mathcal{X})/3c$. We have three cases:

- First consider the case that $\Delta(x, x') > \epsilon/3$. Let $x'$ be the nearest neighbour of $x$ in $[0,1]^d$ with $\hat{y}'(x) \neq \hat{y}'(x')$. Choose $x^\dagger \in \mathcal{S}'$ such that $\Delta(x, x^\dagger) < \lambda'\epsilon$. Since $\Delta(x, x^\dagger) < \epsilon/3$ we must have $\hat{y}'(x^\dagger) = \hat{y}'(x)$. Let $x''$ be the nearest neighbour of $x^\dagger$ in $\mathcal{X}$ with $\hat{y}'(x'') \neq \hat{y}'(x^\dagger)$. Since $\hat{y}'(x^\dagger) = \hat{y}'(x)$ we must have that $\hat{y}'(x'') \neq \hat{y}'(x)$ so that $\Delta(x, x'') > \epsilon/3$. By the triangle inequality we then have:

$$\epsilon/3 < \Delta(x, x'') \leq \Delta(x, x^\dagger) + \Delta(x^\dagger, x'') < \lambda'\epsilon + \Delta(x^\dagger, x'') = \lambda'\epsilon + \gamma(x^\dagger, \hat{y}', \mathcal{X})$$

Hence $\gamma(x^\dagger, \hat{y}', \mathcal{X}) > (1/3 - \lambda')\epsilon$ so that:

$$\Delta(x, x^\dagger) < \lambda'\epsilon < \frac{\lambda'}{1/3 - \lambda'}\gamma(x^\dagger, \hat{y}', \mathcal{X}) < \gamma(x^\dagger, \hat{y}', \mathcal{X})/3c$$

as required.

- Now consider the case that $x \in \mathcal{L}$. In this case we trivially have the result with $x^\dagger := x$.

- Finally consider the case that $x \notin \mathcal{L}$ and $\Delta(x, x') \leq \epsilon/3$. Let $x'$ be the nearest neighbour of $x$ in $[0,1]^d$ with $\hat{y}'(x) \neq \hat{y}'(x')$. Let $\hat{x}$ and $\hat{x}'$ be the nearest neighbours of $x$ and $x'$ in $\mathcal{D} \setminus \mathcal{L}$ respectively. Note that by definition of $\hat{y}'$ we have $\hat{y}'(\hat{x}) = \hat{y}'(x)$ and $\hat{y}'(\hat{x}') = \hat{y}'(x')$. By definition of $\mathcal{D}$ we must have $x \in \mathcal{D}$, so since $x \notin \mathcal{L}$ we have $\hat{x} = x$. Noting that $\Delta(x', \hat{x}') \leq \Delta(x', \hat{x})$ we must then have that $\Delta(x', \hat{x}') \leq \Delta(x', x)$ which means, by the triangle inequality, that $\Delta(x, \hat{x}') \leq 2\Delta(x, x')$. Since $x, \hat{x}' \in \mathcal{D}$ choose $z \in \mathcal{H}(x)$ and $z' \in \mathcal{H}(\hat{x}')$ such that $\mu(z), \mu(z') \neq 0$. Since $x, \hat{x}' \in \mathcal{D} \setminus \mathcal{L}$ we have:

$$\hat{y}(z) = \hat{y}'(z) = \hat{y}'(x) \neq \hat{y}'(\hat{x}') = \hat{y}'(z') = \hat{y}(z')$$

By the triangle inequality and above we have:

$$\Delta(z, z') \leq \Delta(z, x) + \Delta(x, \hat{x}') + \Delta(\hat{x}', z') \leq 2\Delta(x, x') + \sqrt{d}/q \leq \epsilon \tag{10}$$

so since $\mu(z), \mu(z') \neq 0$ we must have that the straight line from $z$ to $z'$ is entirely contained in $\mathcal{E}(\mu, \epsilon/2)$. Since $\hat{y}(z) \neq \hat{y}(z')$ we can then choose some $z^\dagger$ in the line from $z$ to $z'$ that is on the decision boundary of $\hat{y}$ (i.e. any open set around $z^\dagger$ contains some $\tilde{z}$ with $\hat{y}(\tilde{z}) \neq \hat{y}(z^\dagger)$). Since there exists an open set around $z^\dagger$ that is entirely contained in $\mathcal{E}(\mu, \epsilon)$

we must now have, by definition of $\mathcal{L}$, that there exists $\hat{z} \in \mathcal{L}$ such that $z^\dagger \in \mathcal{H}(\hat{z})$. By the triangle inequality and Equation (10) we have:

$$\Delta(x, \hat{z}) \leq \Delta(x, z) + \Delta(z, z^\dagger) + \Delta(z^\dagger, \hat{z}) \leq \Delta(z, z^\dagger) + \sqrt{d}/q \leq \Delta(z, z') + \sqrt{d}/q$$
$$\leq 2\Delta(x, x') + 2\sqrt{d}/q \tag{11}$$

Since $x \in \mathcal{D} \setminus \mathcal{L}$ we have $\hat{y}'(\tilde{z}) = \hat{y}'(x)$ for all $\tilde{z} \in \mathcal{H}(x)$ and hence we must have $x' \notin \mathcal{H}(x)$ so that $\Delta(x, x') \geq \sqrt{d}/(2q)$. By Equation (11) this means that:

$$\Delta(x, x') \geq \Delta(x, \hat{z})/6 \tag{12}$$

By Equation (9) choose $z'' \in \mathcal{A}'$ such that:

$$\Delta(x - \hat{z}, z'') < 2\Delta(x - \hat{z}, 0)\lambda \tag{13}$$

and define $x^\dagger = \hat{z} + z''$. Since $\hat{z} \in \mathcal{L}$ we have $x^\dagger \in \mathcal{S}$ as required. Note that by equations (12) and (13) we have:

$$\Delta(x, x^\dagger) = \Delta(x - \hat{z}, z'') < 2\Delta(x, \hat{z})\lambda \leq 12\Delta(x, x')\lambda \tag{14}$$

Since $2\lambda < 1/6$ we now have, from equations (12) and (14), that $\Delta(x, x^\dagger) < \Delta(x, x')$ so that $\hat{y}'(x) = \hat{y}'(x^\dagger)$. Let $x''$ be the nearest neighbour of $x^\dagger$ in $\mathcal{X}$ with $\hat{y}'(x'') \neq \hat{y}'(x^\dagger)$. Since $\hat{y}'(x^\dagger) = \hat{y}'(x)$ we must have that $\hat{y}'(x'') \neq \hat{y}'(x)$ so that $\Delta(x, x'') \geq \Delta(x, x')$. By the triangle inequality and Equation (14) we then have:

$$\Delta(x, x') \leq \Delta(x, x'') \leq \Delta(x, x^\dagger) + \Delta(x^\dagger, x'') < 12\Delta(x, x')\lambda + \Delta(x^\dagger, x'')$$
$$= 12\Delta(x, x')\lambda + \gamma(x^\dagger, \hat{y}', \mathcal{X})$$

Hence $\gamma(x^\dagger, \hat{y}', \mathcal{X}) > (1 - 12\lambda)\Delta(x, x')$ so that by Equation (14) we have:

$$\Delta(x, x^\dagger) < 12\Delta(x, x')\lambda < \frac{12\lambda}{1 - 12\lambda}\gamma(x^\dagger, \hat{y}', \mathcal{X}) < \gamma(x^\dagger, \hat{y}', \mathcal{X})/3c$$

as required.

Let $\boldsymbol{y}' \in [K]^T$ be such that $y'_t := \hat{y}'(x_t)$ for all $t \in [T]$. We have shown that for all $x \in \mathcal{X}$ there exists $x^\dagger \in \mathcal{S}$ with $\Delta(x, x^\dagger) < \gamma(x^\dagger, \hat{y}', \mathcal{X})/3c$. By equations (6) and (8) we have that:

$$|\mathcal{S}| \in \mathcal{O}(|\mathcal{L}| \ln(q)) \subseteq \mathcal{O}(\alpha(\hat{y}, \mu)q^{d-1} \ln(q)) \subseteq \tilde{\mathcal{O}}(\alpha(\hat{y}, \mu)q^{d-1})$$

Invoking Theorem 3.6 then gives us:

$$\Phi(\boldsymbol{y}') \in \mathcal{O}(\alpha(\hat{y}, \mu)q^{d-1})$$

so by Theorem 3.3 we have:

$$\mathbb{E}[R(\boldsymbol{y}')] \in \tilde{\mathcal{O}}\left(\left(\left(\rho + \frac{\Phi(\boldsymbol{y}')}{\rho}\right)\sqrt{KT}\right) \subseteq \tilde{\mathcal{O}}\left((1 + \alpha(\hat{y}, \mu))q^{\frac{d-1}{2}}\sqrt{KT}\right)\right. \tag{15}$$

We also have:

$$R(\boldsymbol{y}) - R(\boldsymbol{y}') = \sum_{t \in [T]}(\ell_{t, y_t} - \ell_{t, y'_t}) = \sum_{t \in [T]}(\ell_{t, \hat{y}(z_t)} - \ell_{t, \hat{y}'(x_t)}) \leq \sum_{t \in [T]}\llbracket \hat{y}(z_t) \neq \hat{y}'(x_t) \rrbracket$$

But $\hat{y}(z_t) \neq \hat{y}'(x_t)$ implies that $x_t \notin \mathcal{D} \setminus \mathcal{L}$ so since $x_t \in \mathcal{D}$ we must have $x_t \in \mathcal{L}$ which happens with probability $p$ and hence, by Equation (7), we have:

$$\mathbb{E}[R(\boldsymbol{y}) - R(\boldsymbol{y}')] \leq pT \in \mathcal{O}(T\tilde{\alpha}(\hat{y}, \mu)/q) \tag{16}$$

Combining equations (15) and (16) gives us:

$$\mathbb{E}[R(\boldsymbol{y})] \in \tilde{\mathcal{O}}((1 + \alpha(\hat{y}, \mu))q^{\frac{d-1}{2}}\sqrt{KT} + T\tilde{\alpha}(\hat{y}, \mu)/q)$$

Since $q := \lceil(T/K)^{1/(d+1)}\rceil$ we have now shown that:

$$\mathbb{E}[R(\boldsymbol{y})] \in \tilde{\mathcal{O}}\left((1 + \alpha(\hat{y}, \mu) + \tilde{\alpha}(\hat{y}, \mu))T^{\frac{d}{d+1}}K^{\frac{1}{d+1}}\right)$$

as required.

## E.4 Theorem B.1

For every trial $t \in [T]$ define:

$$\Delta_t := - \sum_{v \in \mathcal{B}'} [\![\mathcal{Q}(y, v) \neq \emptyset]\!] \ln(w_t(v, \mathcal{Q}(y, v)))$$

Choose some arbitrary trial $t \in [T]$. From here until we say otherwise all probabilities and expectations (i.e. whenever we use $\mathbb{P}[\cdot]$ or $\mathbb{E}[\cdot]$) are implicitly conditional on the state of the algorithm at the start of trial $t$. Note first that we have:

$$\Delta_t - \Delta_{t+1} = \sum_{v \in \mathcal{B}'} [\![\mathcal{Q}(y, v) \neq \emptyset]\!] \ln \left( \frac{w_{t+1}(v, \mathcal{Q}(y, v))}{w_t(v, \mathcal{Q}(y, v))} \right) \tag{17}$$

For all $j \in [\log(K)] \cup \{0\}$ let $\gamma_{t,j}$ be the ancestor (in $\mathcal{B}$) of $y(x_t)$ at depth $j$. Note that for all $v \in \mathcal{X} \setminus \{\gamma_{t,j} \mid j \in [\log(K)] \cup \{0\}\}$ we have $y(x_t) \notin \Downarrow(v)$ so that $x_t \notin \mathcal{Q}(y, v)$ and hence, directly from the CANPROP algorithm, we have $w_{t+1}(v, \mathcal{Q}(y, v)) = w_t(v, \mathcal{Q}(y, v))$. By Equation (17) and the fact that $\mathcal{Q}(y, v) \neq \emptyset$ for all ancestors $v$ of $y(x_t)$ this implies that:

$$\Delta_t - \Delta_{t+1} = \sum_{j \in [\log(K)]} \ln \left( \frac{w_{t+1}(\gamma_{t,j}, \mathcal{Q}(y, \gamma_{t,j}))}{w_t(\gamma_{t,j}, \mathcal{Q}(y, \gamma_{t,j}))} \right) \tag{18}$$

For all $j \in [\log(K)]$ define:

$$\lambda_{t,j} := \ln \left( \frac{w_{t+1}(\gamma_{t,j}, \mathcal{Q}(y, \gamma_{t,j}))}{w_t(\gamma_{t,j}, \mathcal{Q}(y, \gamma_{t,j}))} \right)$$

and:

$$\epsilon_{t,j} := \mathbb{E}[\ln(\psi_{t,j}) \mid \gamma_{t,j} \in \mathcal{P}_t]$$

Now choose some arbitrary $j \in [\log(K)]$. If $\gamma_{t,(j-1)} \in \mathcal{P}_t$ then $\gamma_{t,(j-1)} = v_{t,(j-1)}$ so $\uparrow(\gamma_{t,j}) = v_{t,(j-1)}$ and hence, since $x_t \in \mathcal{Q}(y, \gamma_{t,j})$, we have $\lambda_{t,j} = \ln(\beta_t(\gamma_{t,j}))$. By definition of $\beta_t(\gamma_{t,j})$ this means that:

$$\mathbb{E}[\lambda_{t,j} \mid \gamma_{t,j} \in \mathcal{P}_t , \gamma_{t,(j-1)} \in \mathcal{P}_t] = \epsilon_{t,j} - \mathbb{E}[\ln(\psi_{t,(j-1)}) \mid \gamma_{t,j} \in \mathcal{P}_t , \gamma_{t,(j-1)} \in \mathcal{P}_t]$$

and that:

$$\mathbb{E}[\lambda_{t,j} \mid \gamma_{t,j} \notin \mathcal{P}_t , \gamma_{t,(j-1)} \in \mathcal{P}_t] = -\mathbb{E}[\ln(\psi_{t,(j-1)}) \mid \gamma_{t,j} \notin \mathcal{P}_t , \gamma_{t,(j-1)} \in \mathcal{P}_t]$$

Multiplying these two equations by $\mathbb{P}[\gamma_{t,j} \in \mathcal{P}_t \mid \gamma_{t,(j-1)} \in \mathcal{P}_t]$ and $\mathbb{P}[\gamma_{t,j} \notin \mathcal{P}_t \mid \gamma_{t,(j-1)} \in \mathcal{P}_t]$ respectively, and summing them together, then gives us:

$$\mathbb{E}[\lambda_{t,j} \mid \gamma_{t,(j-1)} \in \mathcal{P}_t] = \mathbb{P}[\gamma_{t,j} \in \mathcal{P}_t \mid \gamma_{t,(j-1)} \in \mathcal{P}_t]\epsilon_{t,j} - \mathbb{E}[\ln(\psi_{t,(j-1)} \mid \gamma_{t,(j-1)} \in \mathcal{P}_t]$$

Since $\mathbb{P}[\gamma_{t,j} \in \mathcal{P}_t \mid \gamma_{t,(j-1)} \in \mathcal{P}_t] = \pi_t(\gamma_{t,j})$ we then have:

$$\mathbb{E}[\lambda_{t,j} \mid \gamma_{t,(j-1)} \in \mathcal{P}_t] = \pi_t(\gamma_{t,j})\epsilon_{t,j} - \epsilon_{t,(j-1)} \tag{19}$$

If, on the other hand, $\gamma_{t,(j-1)} \notin \mathcal{P}_t$ then $\uparrow(\gamma_{t,j}) \notin \mathcal{P}_t$ so $\lambda_{t,j} = 0$. This means that:

$$\mathbb{E}[\lambda_{t,j}] = \mathbb{P}[\gamma_{t,(j-1)} \in \mathcal{P}_t]\mathbb{E}[\lambda_{t,j} \mid \gamma_{t,(j-1)} \in \mathcal{P}_t] \tag{20}$$

Since the probability that $\gamma_{t,(j-1)} \in \mathcal{P}_t$ is equal to $\prod_{j' \in [j-1]} \pi_t(\gamma_{t,j'})$ we then have, by combining equations (19) and (20), that:

$$\mathbb{E}[\lambda_{t,j}] = \epsilon_{t,j} \prod_{j' \in [j]} \pi_t(\gamma_{t,j'}) - \epsilon_{t,(j-1)} \prod_{j' \in [j-1]} \pi_t(\gamma_{t,j'})$$

By substituting into Equation (18) (after taking expectations) we then have that:

$$\mathbb{E}[\Delta_t - \Delta_{t+1}] = -\epsilon_{t,0} + \epsilon_{t,\log(K)} \prod_{j \in [\log(K)]} \pi_t(\gamma_{t,j})$$

$$= -\mathbb{E}[\ln(\psi_{t,0})] + \mathbb{E}[\ln(\psi_{t,\log(K)}) \mid a_t = \gamma_{t,\log(K)}] \prod_{j \in [\log(K)]} \pi_t(\gamma_{t,j}) \tag{21}$$

Note that if $a_t = \gamma_{t,\log(K)}$ then $\gamma_{t,j} = v_{t,j}$ for all $j \in [\log(K)]$. By definition of $\psi_{t,\log(K)}$ and the fact that $\gamma_{t,\log(K)} = y(x_t)$, Equation (21) then gives us:

$$\mathbb{E}[\Delta_t - \Delta_{t+1}] = -\mathbb{E}[\ln(\psi_{t,0})] - \eta \ell_{t,y(x_t)} \tag{22}$$

For all $(v, a) \in \mathcal{B} \times [K]$ define:

$$p_{t,a}(v) = \mathbb{P}[a_t = a \mid v \in \mathcal{P}_t]$$

noting that this is non-zero only when $a \in \Downarrow(v)$. Suppose we have some $v \in \mathcal{B} \setminus \{r(\mathcal{B})\}$ and some $a \in \Downarrow(v) \cap [K]$. Then, since $\mathbb{P}[a_t = a \mid v \notin \mathcal{P}_t] = 0$, we have:

$$p_{t,a}(\uparrow(v)) = \mathbb{P}[a_t = a \mid \uparrow(v) \in \mathcal{P}_t] = \mathbb{P}[a_t = a \mid v \in \mathcal{P}_t]\mathbb{P}[v \in \mathcal{P}_t \mid \uparrow(v) \in \mathcal{P}_t] = \pi_t(v)p_{t,a}(v)$$

Since $p_{t,a}(v) = 0$ whenever $a \notin \Downarrow(v)$, this implies that for all $(v, a) \in \mathcal{B}^\dagger \times [K]$ we have:

$$p_{t,a}(v) = \pi_t(\vartriangleleft(v))p_{t,a}(\vartriangleleft(v)) + \pi_t(\vartriangleright(v))p_{t,a}(\vartriangleright(v)) \tag{23}$$

For all $a \in [K]$ define:

$$\hat{\ell}_{t,a} = \frac{[\![a_t = a]\!]\ell_{t,a}}{\mathbb{P}[a_t = a]}$$

We now take the inductive hypothesis that for all $j \in [\log(K)] \cup \{0\}$ we have:

$$\psi_{t,j} = \sum_{a \in [K]} p_{t,a}(v_{t,j}) \exp(-\eta \hat{\ell}_{t,a})$$

and prove this via reverse induction (i.e. from $j = \log(K)$ to $j = 0$). Note that given $a' := a_t$ we have $\mathbb{P}[a_t = a'] = \prod_{j \in [\log(K)]} \pi_t(v_{t,j})$ and hence:

$$\psi_{t,\log(K)} = \exp(-\eta \hat{\ell}_{t,a_t})$$

so the inductive hypothesis holds for $j = \log(K)$. Now suppose that we have some $j' \in [\log(K)]$ and that the inductive hypothesis holds for $j = j'$. We shall now show that it holds also for $j = j' - 1$. Let $v'$ be the child of $v_{t,(j'-1)}$ that is not equal to $v_{t,j'}$. Note that $a_t \notin \Downarrow(v')$ and hence $\exp(-\eta \hat{\ell}_{t,a}) = 1$ for all $a \in \Downarrow(v')$ (i.e. whenever $p_{t,a}(v') \neq 0$) which implies:

$$\sum_{a \in [K]} p_{t,a}(v') \exp(-\eta \hat{\ell}_{t,a}) = 1 \tag{24}$$

For all $a \in [K]$, Equation (23) gives us:

$$p_{t,a}(v_{t,(j'-1)}) \exp(-\eta \hat{\ell}_{t,a}) = \pi_t(v')p_{t,a}(v') \exp(-\eta \hat{\ell}_{t,a}) + \pi_t(v_{t,j'})p_{t,a}(v_{t,j'}) \exp(-\eta \hat{\ell}_{t,a})$$

Substituting Equation (24) and the inductive hypothesis into this equation (when summed over all $a \in [K]$) then gives us:

$$\sum_{a \in [K]} p_{t,a}(v_{t,(j'-1)}) \exp(-\eta \hat{\ell}_{t,a}) = \pi_t(v') + \pi_t(v_{t,j'})\psi_{t,j'}$$

Since $\pi_t(v') + \pi_t(v_{t,j'}) = 1$ we have, direct from the algorithm, that $\pi_t(v') + \pi_t(v_{t,j'})\psi_{t,j'} = \psi_{t,(j'-1)}$ so the inductive hypothesis holds for $j = j' - 1$. We have hence shown that the inductive hypothesis holds for all $j \in [\log(K)] \cup \{0\}$ and in particular for $j = 0$. Since $p_{t,a}(v_{t,0}) = \mathbb{P}[a_t = a]$ we then have:

$$\psi_{t,0} = \sum_{a \in [K]} \mathbb{P}[a_t = a] \exp(-\eta \hat{\ell}_{t,a}) \tag{25}$$

Since $\exp(-z) \leq 1 - z + z^2/2$ for all $z \in \mathbb{R}_+$ we have, from Equation (25), that:

$$\psi_{t,0} \leq \sum_{a \in [K]} \mathbb{P}[a_t = a] \left(1 - \eta\hat{\ell}_{t,a} + \frac{\eta^2 \hat{\ell}_{t,a}}{2}\right) = 1 - \eta \sum_{a \in [K]} \mathbb{P}[a_t = a]\hat{\ell}_{t,a} + \frac{\eta^2}{2} \sum_{a \in [K]} \mathbb{P}[a_t = a]\hat{\ell}_{t,a}^2$$

so since $\ln(1 + z) \leq z$ for all $z \in \mathbb{R}$ we have:

$$\ln(\psi_{t,0}) \leq -\eta \sum_{a \in [K]} \mathbb{P}[a_t = a]\hat{\ell}_{t,a} + \frac{\eta^2}{2} \sum_{a \in [K]} \mathbb{P}[a_t = a]\hat{\ell}_{t,a}^2 \tag{26}$$

Noting that $\mathbb{P}[a_t = a]\hat{\ell}_{t,a} = [\![a_t = a]\!]\ell_{t,a}$ for all $a \in [K]$, we have:

$$\mathbb{E}\left[\sum_{a\in[K]} \mathbb{P}[a_t = a]\hat{\ell}_{t,a}\right] = \mathbb{E}[\ell_{t,a_t}]$$

and:

$$\mathbb{E}\left[\sum_{a\in[K]} \mathbb{P}[a_t = a]\hat{\ell}_{t,a}^2\right] = \mathbb{E}\left[\sum_{a\in[K]} \frac{[\![a_t = a]\!]\ell_{t,a}^2}{\mathbb{P}[a_t = a]}\right] = \sum_{a\in[K]} \ell_{t,a}^2 \leq K$$

Substituting these equations into Equation (26) (after taking expectations) gives us:

$$\mathbb{E}[\ln(\psi_{t,0})] \leq -\eta\mathbb{E}[\ell_{t,a_t}] + \eta^2 K/2$$

which, upon substitution into Equation (22) gives us:

$$\mathbb{E}[\Delta_t - \Delta_{t+1}] \geq \eta(\mathbb{E}[\ell_{t,a_t}] - \ell_{t,y(x_t)}) - \eta^2 K/2 \tag{27}$$

Note that this equation implies that the same equation also holds when the expectation is not implicitly conditional on the state of the algorithm at the start of trial $t$. Hence, we now drop the assumption that the expectation is conditional on the state of the algorithm at the start of trial $t$. Summing Equation (27) over all trials $t \in [T]$ and then rearranging gives us:

$$\mathbb{E}[R(y)] \leq \frac{1}{\eta}(\mathbb{E}[\Delta_1] - \mathbb{E}[\Delta_{T+1}]) + \frac{\eta KT}{2} \tag{28}$$

Now consider a trial $t$. For all $v \in \mathcal{B}^\dagger$ let:

$$V_t(v) := \sum_{\mathcal{S}\in 2^{\mathcal{X}}} [\![x_t \in \mathcal{S}]\!]w_{t+1}(\lhd(v), \mathcal{S}) + \sum_{\mathcal{S}\in 2^{\mathcal{X}}} [\![x_t \in \mathcal{S}]\!]w_{t+1}(\rhd(v), \mathcal{S})$$

Now take any $j \in [\log(K) - 1] \cup \{0\}$ and let $v := v_{t,j}$. Note that:

$$V_t(v) = \beta_t(\lhd(v))\theta_t(\lhd(v)) + \beta_t(\rhd(v))\theta_t(\rhd(v))$$

so that by definition of $\pi_t(\lhd(v))$ and $\pi_t(\rhd(v))$ we have:

$$V_t(v) = (\theta_t(\lhd(v)) + \theta_t(\rhd(v)))(\pi_t(\lhd(v))\beta_t(\lhd(v)) + \pi_t(\rhd(v))\beta_t(\rhd(v)))$$

Without loss of generality assume that $\lhd(v) \in \mathcal{P}_t$. Then the above equation implies that:

$$V_t(v) = (\theta_t(\lhd(v)) + \theta_t(\rhd(v)))\frac{\pi_t(\lhd(v))\psi_{t,j+1} + \pi_t(\rhd(v))}{\psi_{t,j}}$$

so by definition of $\psi_{t,j}$ we have:

$$V_t(v) = (\theta_t(\lhd(v)) + \theta_t(\rhd(v))) = \sum_{\mathcal{S}\in 2^{\mathcal{X}}} [\![x_t \in \mathcal{S}]\!]w_t(\lhd(v), \mathcal{S}) + \sum_{\mathcal{S}\in 2^{\mathcal{X}}} [\![x_t \in \mathcal{S}]\!]w_t(\rhd(v), \mathcal{S})$$

Note that this equation trivially holds for all $v \in \mathcal{B}^\dagger \setminus \mathcal{P}_t$ and hence holds for all $v \in \mathcal{B}^\dagger$. Since for all such $v$ and all $\mathcal{S}$ with $x_t \notin \mathcal{S}$ we have $w_{t+1}(\lhd(v), \mathcal{S}) = w_t(\lhd(v), \mathcal{S})$ and $w_{t+1}(\rhd(v), \mathcal{S}) = w_t(\rhd(v), \mathcal{S})$ we then have:

$$\sum_{\mathcal{S}\in 2^{\mathcal{X}}} w_{t+1}(\lhd(v), \mathcal{S}) + \sum_{\mathcal{S}\in 2^{\mathcal{X}}} w_{t+1}(\rhd(v), \mathcal{S}) = \sum_{\mathcal{S}\in 2^{\mathcal{X}}} w_t(\lhd(v), \mathcal{S}) + \sum_{\mathcal{S}\in 2^{\mathcal{X}}} w_t(\rhd(v), \mathcal{S})$$

so, by induction on $t$ we have, for all $t \in [T + 1]$, that:

$$\sum_{\mathcal{S}\in 2^{\mathcal{X}}} w_t(\lhd(v), \mathcal{S}) + \sum_{\mathcal{S}\in 2^{\mathcal{X}}} w_t(\rhd(v), \mathcal{S}) = 1$$

Hence, for all $v \in \mathcal{B} \setminus r(\mathcal{B})$ and $\mathcal{S} \in 2^{\mathcal{X}}$, we have $w_t(v, \mathcal{S}) \in [0, 1]$. We have now shown that $\Delta_{T+1} \geq 0$ so that Equation 28 gives us:

$$\mathbb{E}[R(y)] \leq \frac{1}{\eta}\mathbb{E}[\Delta_1] + \frac{\eta KT}{2}$$

which, by definition of $\Delta_1$, gives us the desired result.

## E.5 Theorem B.2

The fact that the weighting $w_1$ is valid is given by the following lemma:

**Lemma E.1.** *For all $v \in \mathcal{B}^\dagger$ we have:*

$$\sum_{\mathcal{S} \in 2^{\mathcal{X}}} \left(w_1(\triangleleft(v), \mathcal{S}) + w_t(\triangleright(v), \mathcal{S})\right) = 1$$

*Proof.* We will show that for all $v \in \mathcal{B}'$ we have:

$$\sum_{\mathcal{S} \in 2^{\mathcal{X}}} w_1(v, \mathcal{S}) = \frac{1}{2}$$

which directly implies the result. So take some arbitrary $v \in \mathcal{B}'$. Define, for all $t \in [T]$, the sets:

$$\mathcal{X}'_t := \{x_s \mid s \in [t]\} \setminus \{x_1\} \quad \text{and} \quad \mathcal{F}_t := \{0,1\}^{\mathcal{X}'_t \cup \{x_1\}}$$

and for all $x \in \mathcal{X}'_t$ and $f \in \mathcal{F}_t$ define the quantity:

$$\beta(x, f) := [\![f(x) \neq f(n(x))]\!] 1/T + [\![f(x) = f(n(x))]\!](1 - 1/T)$$

which is defined since $n(x) \in \mathcal{X}'_t \cup \{x_1\}$. For all $t \in [T-1]$ we have:

$$\sum_{f \in \mathcal{F}_{t+1}} \prod_{x \in \mathcal{X}'_{t+1}} \beta(x, f) = \sum_{f \in \mathcal{F}_t} \left(\prod_{x \in \mathcal{X}'_t} \beta(x, f)\right) \sum_{f(x_{t+1}) \in \{0,1\}} \beta(x_{t+1}, f) \tag{29}$$

Given any $f \in \mathcal{F}_t$ we have:

$$\sum_{f(x_{t+1}) \in \{0,1\}} \beta(x_{t+1}, f) = (1 - 1/T) + 1/T = 1$$

and hence by Equation (29) we have:

$$\sum_{f \in \mathcal{F}_{t+1}} \prod_{x \in \mathcal{X}'_{t+1}} \beta(x, f) = \sum_{f \in \mathcal{F}_t} \prod_{x \in \mathcal{X}'_t} \beta(x, f)$$

Since $\mathcal{X}'_T = \mathcal{X}'$ this implies, by induction, that:

$$\sum_{f \in \mathcal{F}_T} \prod_{x \in \mathcal{X}'} \beta(x, f) = \sum_{f \in \mathcal{F}_1} \prod_{x \in \mathcal{X}'_1} \beta(x, f) = \sum_{f \in \mathcal{F}_1} \prod_{x \in \emptyset} \beta(x, f) = \sum_{f \in \mathcal{F}_1} 1 = |\mathcal{F}_1| = 2 \tag{30}$$

Note that we have a bijection $\mathcal{G} : \mathcal{F}_T \to 2^{\mathcal{X}}$ defined by:

$$\mathcal{G}(f) := \{x \in \mathcal{X} \mid f(x) = 1\} \quad \forall f \in \mathcal{F}_T$$

and that for all $(f, x) \in \mathcal{F}_T \times \mathcal{X}'$ we have:

$$\beta(x, f) = \sigma(x, \mathcal{G}(f))/T + (1 - \sigma(x, \mathcal{G}(f)))(1 - 1/T)$$

Hence, Equation (30) shows us that:

$$\sum_{\mathcal{S} \in 2^{\mathcal{X}}} \prod_{x \in \mathcal{X}'} \left(\sigma(x, \mathcal{S})\frac{1}{T} + (1 - \sigma(x, \mathcal{S}))\left(1 - \frac{1}{T}\right)\right) = 2$$

This implies that:

$$\sum_{\mathcal{S} \in 2^{\mathcal{X}}} w_1(v, \mathcal{S}) = \frac{1}{2}$$

which implies the result. $\qquad\square$

Now that we have shown that the weighting $w_1$ is valid we can utilise Theorem B.1 to prove our regret bound. For any set $\mathcal{S} \in 2^{\mathcal{X}}$ define:

$$\phi(\mathcal{S}) := \sum_{x \in \mathcal{X}'} \sigma(x, \mathcal{S})$$

Note that for all $v \in \mathcal{B}^\dagger$ and $\mathcal{S} \in 2^{\mathcal{X}}$ we have:

$$w_1(v, \mathcal{S}) = \frac{1}{4} \left(\frac{1}{T}\right)^{\phi(\mathcal{S})} \left(1 - \frac{1}{T}\right)^{T-1-\phi(\mathcal{S})} \geq \frac{1}{4} \left(\frac{1}{T}\right)^{\phi(\mathcal{S})} \left(1 - \frac{1}{T}\right)^{T}$$

so since $T \ln(1 - 1/T) \in \mathcal{O}(1)$ we have:

$$- \ln(w_1(v, \mathcal{S})) \leq \ln(4) + \phi(\mathcal{S}) \ln(T) - T \ln(1 - 1/T) \in \mathcal{O}(\phi(\mathcal{S}) \ln(T) + 1) \tag{31}$$

As in the statement of Theorem B.1 define, for all $v \in \mathcal{B}$, the set:

$$\mathcal{Q}(y, v) := \{x \in \mathcal{X} \mid y(x) \in \Downarrow(v)\}$$

First note that the graph (with vertex set $\mathcal{X}$) formed by linking $x$ to $n(x)$ for every $x \in \mathcal{X}'$ is a tree so that $\Phi(y) \geq |\{y(x) \mid x \in \mathcal{X}\}| - 1$. So since for all $v \in \mathcal{B}'$ we have $\mathcal{Q}(y, v) \neq \emptyset$ if and only if $v$ has a descendent in $\{y(x) \mid x \in \mathcal{X}\}$ and each element of $\{y(x) \mid x \in \mathcal{X}\}$ has $\log(K)$ ancestors in $\mathcal{B}'$ we have:

$$\sum_{v \in \mathcal{B}'} \llbracket \mathcal{Q}(y, v) \neq \emptyset \rrbracket \leq \log(K) |\{y(x) \mid x \in \mathcal{X}\}| \leq \log(K)(\Phi(y) + 1) \tag{32}$$

Now suppose we have some $x \in \mathcal{X}'$. If $y(x) = y(n(x))$ then for all $v \in \mathcal{B}'$ we have $x, n(x) \in \mathcal{Q}(y, v)$ or $x, n(x) \notin \mathcal{Q}(y, v)$ and hence $\sigma(x, \mathcal{Q}(y, v)) = 0$. On the other hand, if $y(x) \neq y(n(x))$ then for any $v \in \mathcal{B}'$ with $\sigma(x, \mathcal{Q}(y, v)) = 1$ we must have that either $x \in \mathcal{Q}(y, v)$ or $n(x) \in \mathcal{Q}(y, v)$ so $v$ is an ancestor of either $x$ or $n(x)$ and hence there can be at most $2 \log(K)$ such $v$. So in any case we have:

$$\sum_{v \in \mathcal{B}'} \sigma(x, \mathcal{Q}(y, v)) \leq \llbracket y(x) \neq y(n(x)) \rrbracket 2 \log(K)$$

Hence we have:

$$\sum_{v \in \mathcal{B}'} \phi(\mathcal{Q}(y, v)) = \sum_{x \in \mathcal{X}'} \sum_{v \in \mathcal{B}'} \sigma(x, \mathcal{Q}(y, v)) \leq 2 \log(K) \Phi(y) \tag{33}$$

Equation (31) gives us:

$$- \sum_{v \in \mathcal{B}'} \llbracket \mathcal{Q}(y, v) \neq \emptyset \rrbracket \ln(w_1(v, \mathcal{Q}(y, v))) \in \mathcal{O}\left( \ln(T) \sum_{v \in \mathcal{B}'} \phi(\mathcal{Q}(y, v)) + \sum_{v \in \mathcal{B}'} \llbracket \mathcal{Q}(y, v) \neq \emptyset \rrbracket \right)$$

Substituting in equations (32) and (33) then gives us:

$$- \sum_{v \in \mathcal{B}'} \llbracket \mathcal{Q}(y, v) \neq \emptyset \rrbracket \ln(w_1(v, \mathcal{Q}(y, v))) \in \mathcal{O}(\ln(K) \ln(T) \Phi(y))$$

so by Theorem B.1 we have:

$$\mathbb{E}[R(y)] \in \mathcal{O}\left( \frac{\eta K T}{2} + \frac{\ln(K) \ln(T) \Phi(y)}{\eta} \right)$$

Since $\eta = \rho \sqrt{\ln(K) \ln(T) / KT}$ we obtain the result.

### E.6 Theorem C.1

Recall our sequence of trees $\langle \mathcal{Z}_t \mid t \in [T] \setminus \{1\} \rangle$ noting that each of these trees is a contraction so that $\tau_{i,i'}(\mathcal{Z}_t, \cdot)$ is defined for all $i, i' \in \{0, 1\}$. Let $\epsilon := 1/T$. Define $\lambda' : \mathcal{X} \to \mathbb{R}_+$ as follows. Given $x \in \mathcal{X}$, if there exists a leaf $u \in \mathcal{J}^\star$ with $\gamma(u) = x$ then $\lambda'(x) = \lambda(u)$. Otherwise $\lambda'(x) = 1$. Given $t \in [T]$ define $\hat{\lambda}_t : \mathcal{Z}_t \to \mathbb{R}_+$ such that for all $u \in \mathcal{Z}_t$ we have that $\hat{\lambda}_t(u) := \lambda'(\gamma(u))$ if $u$ is a leaf of $\mathcal{Z}_t$ and $\hat{\lambda}_t(u) := 1$ otherwise. For all $t \in [T]$ and $f : \{x_{t'} \mid t' \in [t]\} \to \{0, 1\}$ define:

$$\mathcal{N}(f) := \{f' \in \{0, 1\}^{\mathcal{Z}_t} \mid \forall u \in \mathcal{Z}_t^\star, f'(u) = f(\gamma(u))\}$$

and:

$$\hat{w}(f) := \left( \prod_{t' \in [t]: f(x_{t'})=1} \lambda'(x_t) \right) \prod_{t' \in [t] \setminus \{1\}} (\llbracket f(x_t) \neq f(n(x_t)) \rrbracket \epsilon + \llbracket f(x_t) = f(n(x_t)) \rrbracket (1-\epsilon))$$

and:

$$\hat{\nu}(f) := \sum_{f' \in \mathcal{N}(f)} \prod_{u \in \mathcal{Z}_t \setminus \{r(\mathcal{Z}_t)\}} \tau_{f'(\uparrow_{\mathcal{Z}_t}(u)), f'(u)}(\mathcal{Z}_t, u) \tilde{\kappa}_{f'(u)}(\mathcal{Z}_t, \hat{\lambda}_t, u)$$

We now have the following lemma:

**Lemma E.2.** *For all $t \in [T]$ and $f : \{x_{t'} \mid t' \in [t]\} \rightarrow \{0,1\}$ we have:*

$$\hat{w}(f) = \hat{\nu}(f)$$

*Proof.* We prove by induction on $t$. Suppose the result holds for $t = s$ (for some $s \geq 2$). We now show that it holds for $t = s+1$ as well. Let $f^*$ be the restriction of $f$ onto the set $\{x_{t'} \mid t' \in [s]\}$. Let $u^*$ and $u'$ be the unique leaves in $\mathcal{Z}^*_{s+1}$ of which $\gamma(u') = n(x_{s+1})$ and $\gamma(u^*) = x_{s+1}$. By the construction of $\mathcal{Z}_{s+1}$ these vertices are siblings. Let $u''$ be the parent (in $\mathcal{Z}_{s+1}$) of both $u^*$ and $u'$. First note that:

$$\llbracket f(x_{s+1}) = 0 \rrbracket + \llbracket f(x_{s+1}) = 1 \rrbracket \lambda'(x_{s+1}) = \tilde{\kappa}_{f(x_{s+1})}(\mathcal{Z}_{s+1}, \hat{\lambda}_{s+1}, u^*) \tag{34}$$

Since, by the construction of $\mathcal{Z}_{s+1}$, we have $\gamma(\uparrow_{\mathcal{Z}_{s+1}}(u^*)) = \gamma(u'') = n(x_{s+1})$ we also have that $d(\uparrow_{\mathcal{Z}_{s+1}}(u^*)) = d(u^*) - 1$ so that, since $\phi_1 = \epsilon$, we have:

$$\llbracket f(x_{s+1}) \neq f(n(x_{s+1})) \rrbracket \epsilon + \llbracket f(x_{s+1}) = f(n(x_{s+1})) \rrbracket (1-\epsilon) = \tau_{f(n(x_{s+1})), f(x_{s+1})}(\mathcal{Z}_{s+1}, u^*) \tag{35}$$

Equations (34) and (35) give us:

$$\hat{w}(f) = \hat{w}(f^*) \tau_{f(n(x_{s+1})), f(x_{s+1})}(\mathcal{Z}_{s+1}, u^*) \tilde{\kappa}_{f(x_{s+1})}(\mathcal{J}, \hat{\lambda}_{s+1}, u^*) \tag{36}$$

Now suppose we have some $f' \in \mathcal{N}(f)$. We have $\gamma(u'') = \gamma(u')$ and hence $d(\uparrow_{\mathcal{Z}_{s+1}}(u')) = d(u'') = d(u')$ so since $f'(u') = f(n(x_{s+1}))$ and $\phi_0 = 0$ we have:

$$\tau_{f'(\uparrow_{\mathcal{Z}_{s+1}}(u')), f'(u')}(\mathcal{Z}_{s+1}, u') = \tau_{f'(u''), f'(u')}(\mathcal{Z}_{s+1}, u') = \llbracket f'(u'') = f(n(x_{s+1})) \rrbracket \tag{37}$$

Since, by the construction of $\mathcal{Z}_{s+1}$, we have $\uparrow_{\mathcal{Z}_{s+1}}(u'') = \uparrow_{\mathcal{Z}_s}(u')$ and (as above) we have $d(u'') = d(u')$, we also have:

$$\tau_{f'(\uparrow_{\mathcal{Z}_{s+1}}(u'')), f'(u'')}(\mathcal{Z}_{s+1}, u'') = \tau_{f'(\uparrow_{\mathcal{Z}_s}(u')), f'(u'')}(\mathcal{Z}_s, u') \tag{38}$$

Since $f'(u^*) = f(x_{s+1})$ and $\uparrow_{\mathcal{Z}_{s+1}}(u^*) = u''$ we have:

$$\tau_{f'(\uparrow_{\mathcal{Z}_{s+1}}(u^*)), f'(u^*)}(\mathcal{Z}_{s+1}, u^*) = \tau_{f'(u''), f(x_{s+1})}(\mathcal{Z}_{s+1}, u^*) \tag{39}$$

Now let:

$$\zeta^* := \tau_{f(n(x_{s+1})), f(x_{s+1})}(\mathcal{Z}_{s+1}, u^*) \quad ; \quad \zeta' := \tau_{f'(\uparrow_{\mathcal{Z}_s}(u')), f(n(x_{s+1}))}(\mathcal{Z}_s, u')$$

Define:

$$g(f') := \prod_{u \in \mathcal{Z}_s \setminus \{r(\mathcal{Z}_s)\}} \tau_{f'(\uparrow_{\mathcal{Z}_s}(u)), f'(u)}(\mathcal{Z}_s, u)$$

and:

$$g'(f') := \prod_{u \in \mathcal{Z}_{s+1} \setminus \{r(\mathcal{Z}_{s+1})\}} \tau_{f'(\uparrow_{\mathcal{Z}_{s+1}}(u)), f'(u)}(\mathcal{Z}_{s+1}, u)$$

Combining equations (37), (38) and (39) gives us:

$$\prod_{u \in \{u^*, u', u''\}} \tau_{f'(\uparrow_{\mathcal{Z}_{s+1}}(u)), f'(u)}(\mathcal{Z}_{s+1}, u) = \llbracket f'(u'') = f(n(x_{s+1})) \rrbracket \zeta^* \zeta' \tag{40}$$

For all $u \in \mathcal{Z}_{s+1} \setminus \{u^*, u', u''\}$ we have $\uparrow_{\mathcal{Z}_{s+1}}(u) = \uparrow_{\mathcal{Z}_s}(u)$ so that:

$$\tau_{f'(\uparrow_{\mathcal{Z}_{s+1}}(u)), f'(u)}(\mathcal{Z}_{s+1}, u) = \tau_{f'(\uparrow_{\mathcal{Z}_s}(u)), f'(u)}(\mathcal{Z}_s, u)$$

and hence, since $f(n(x_{s+1})) = f'(u')$, we have:

$$g'(f') = \frac{g(f')}{\zeta'} \prod_{u \in \{u^*, u', u''\}} \tau_{f'(\uparrow_{\mathcal{Z}_{s+1}}(u)), f'(u)}(\mathcal{Z}_{s+1}, u)$$

Substituting in Equation (40) gives us:

$$g'(f') = g(f')[\![f'(u'') = f(n(x_{s+1}))]\!]\zeta^* \tag{41}$$

We have $\tilde{\kappa}_{f'(u'')}(\mathcal{Z}_{s+1}, \hat{\lambda}_{s+1}, u'') = 1$ and for all $u \in \mathcal{Z}_s$ we have $\tilde{\kappa}_{f'(u)}(\mathcal{Z}_{s+1}, \hat{\lambda}_{s+1}, u) = \tilde{\kappa}_{f'(u)}(\mathcal{Z}_s, \hat{\lambda}_s, u)$. Substituting into Equation (41) gives us:

$$g'(f') \prod_{u \in \mathcal{Z}_{s+1}} \tilde{\kappa}_{f'(u)}(\mathcal{Z}_{s+1}, \hat{\lambda}_{s+1}, u)$$

$$= [\![f'(u'') = f(n(x_{s+1}))]\!]\tilde{\kappa}_{f'(u^*)}(\mathcal{Z}_{s+1}, \hat{\lambda}_{s+1}, u^*)\zeta^* g(f') \prod_{u \in \mathcal{Z}_s} \tilde{\kappa}_{f'(u)}(\mathcal{Z}_s, \hat{\lambda}_s, u)$$

Summing over all $f' \in \mathcal{N}(f)$ and noting that $f'(u^*) = f(x_{s+1})$ and that:

$$\tilde{\kappa}_{f'(r(\mathcal{Z}_{s+1}))}(\mathcal{Z}_{s+1}, \hat{\lambda}_{s+1}, r(\mathcal{Z}_{s+1})) = 1 = \tilde{\kappa}_{f'(r(\mathcal{Z}_s))}(\mathcal{Z}_s, \hat{\lambda}_s, r(\mathcal{Z}_s))$$

gives us:

$$\hat{\nu}(f) = \tilde{\kappa}_{f(x_{s+1})}(\mathcal{Z}_{s+1}, \hat{\lambda}_{s+1}, u^*)\zeta^* \hat{\nu}(f^*)$$

By the inductive hypothesis we then have:

$$\hat{\nu}(f) = \tilde{\kappa}_{f(x_{s+1})}(\mathcal{Z}_{s+1}, \hat{\lambda}_{s+1}, u^*)\zeta^* \hat{w}(f^*)$$

which, by Equation (36), is equal to $\hat{w}(f)$. We have hence shown that if the inductive hypothesis holds for $t = s$ then it holds for $t = s + 1$ also. An identical argument shows that the inductive hypothesis holds for $t = 2$. We have hence shown that the inductive hypothesis holds for all $t \in [T] \setminus \{1\}$. $\square$

We now define a bijection $\mathcal{G} : \{0, 1\}^{\mathcal{X}} \to 2^{\mathcal{X}}$ by:

$$\mathcal{G}(f) := \{x \in \mathcal{X} \mid f(x) = 1\} \quad \forall f \in \{0, 1\}^{\mathcal{X}}$$

Note that for all $f : \mathcal{X} \to \{0, 1\}$ and all $x \in \mathcal{X} \setminus \{x_1\}$ we have:

$$\sigma(x, \mathcal{G}(f))\epsilon + (1 - \sigma(x, \mathcal{G}(f)))(1 - \epsilon) = [\![f(x) \neq f(n(x))]\!]\epsilon + [\![f(x) = f(n(x))]\!](1 - \epsilon)$$

and:

$$\prod_{x \in \mathcal{G}(f)} \lambda'(x) = \prod_{t' \in [T] : f(x_{t'}) = 1} \lambda'(x_t)$$

so that:

$$\tilde{w}(\mathcal{J}, \lambda, \mathcal{G}(f)) = \hat{w}(f)$$

and hence, by Lemma E.2, we have:

$$\tilde{w}(\mathcal{J}, \lambda, \mathcal{G}(f)) = \hat{\nu}(f)$$

so that:

$$\sum_{\mathcal{S} \in 2^{\mathcal{X}}} [\![\gamma(\hat{u}) \in \mathcal{S}]\!]\tilde{w}(\lambda, \epsilon, \mathcal{S}) = \sum_{f \in \{0, 1\}^{\mathcal{X}}} [\![f(\gamma(\hat{u})) = 1]\!]\hat{\nu}(f) \tag{42}$$

Since:

$$\bigcup \{\mathcal{N}(f) \mid f \in \{0, 1\}^{\mathcal{X}}, f(\gamma(\hat{u})) = 1\} = \{f' \in \{0, 1\}^{\mathcal{Z}_T} \mid f'(\hat{u}) = 1\}$$

and all sets in this union are disjoint, the right hand side of Equation (42) is equal to:

$$\sum_{f' \in \{0, 1\}^{\mathcal{Z}_T}} [\![f'(\hat{u}) = 1]\!] \prod_{u \in \mathcal{Z}_T \setminus \{r(\mathcal{Z}_T)\}} \tau_{f'(\uparrow_{\mathcal{Z}_T}(u)), f'(u)}(\mathcal{Z}_T, u)\tilde{\kappa}_{f'(u)}(\mathcal{Z}_T, \hat{\lambda}_T, u) \tag{43}$$

Given a vertex $u \in \mathcal{Z}_T \setminus \{r(\mathcal{Z}_T)\}$ define:

$$\mathcal{H}(u) := \Downarrow_{\mathcal{Z}_T}(u) \cup \{\uparrow_{\mathcal{Z}_T}(u)\}$$

and for all $f : \mathcal{H}(u) \to \{0, 1\}$ define:

$$\hat{\zeta}(u, f) := \prod_{u' \in \Downarrow_{\mathcal{Z}_T}(u)} \tau_{f(\uparrow_{\mathcal{Z}_T}(u')), f(u')}(\mathcal{Z}_T, u')$$

**Lemma E.3.** *Given a vertex $u' \in \mathcal{Z}_T \setminus \{r(\mathcal{Z}_T)\}$ and an index $i \in \{0,1\}$ we have:*

$$\sum_{f \in \{0,1\}^{\mathcal{H}(u')}} \llbracket f(\uparrow_{\mathcal{Z}_T}(u')) = i \rrbracket \hat{\zeta}(u', f) = 1$$

*Proof.* We prove by induction on the height of $\Downarrow_{\mathcal{Z}_T}(u')$. If this height is equal to zero then $\mathcal{H}(u') = \{u', \uparrow_{\mathcal{Z}_T}(u')\}$ and for all $f : \mathcal{H}(u) \to \{0,1\}$ we have:

$$\hat{\zeta}(u', f) = \tau_{f(\uparrow_{\mathcal{Z}_T}(u')), f(u')}(\mathcal{Z}_T, u')$$

Since:

$$\tau_{i,0}(\mathcal{Z}_T, u') + \tau_{i,1}(\mathcal{Z}_T, u') = 1 \qquad (44)$$

we immediately have the result for the case that the height of $\Downarrow_{\mathcal{Z}_T}(u')$ is zero. Now suppose that the result holds whenever the height of $\Downarrow_{\mathcal{Z}_T}(u')$ is equal to $j$ (for some $j \in \mathbb{N}$). We will now show that it holds whenever the height of $\Downarrow_{\mathcal{Z}_T}(u')$ is equal to $j+1$ which will prove that the result holds always. By the inductive hypothesis we have, for all $i' \in \{0,1\}$, that:

$$\sum_{f \in \{0,1\}^{\mathcal{H}(\lhd(u'))}} \llbracket f(u') = i' \rrbracket \hat{\zeta}(\lhd(u'), f) = 1$$

and

$$\sum_{f \in \{0,1\}^{\mathcal{H}(\rhd(u'))}} \llbracket f(u') = i' \rrbracket \hat{\zeta}(\rhd(u'), f) = 1$$

so:

$$\sum_{f \in \{0,1\}^{\mathcal{H}(u')}} \llbracket f(\uparrow_{\mathcal{Z}_T}(u')) = i \rrbracket \llbracket f(u') = i' \rrbracket \hat{\zeta}(\lhd(u'), f) \hat{\zeta}(\rhd(u'), f) = 1$$

and hence:

$$\sum_{f \in \{0,1\}^{\mathcal{H}(u')}} \llbracket f(\uparrow_{\mathcal{Z}_T}(u')) = i \rrbracket \llbracket f(u') = i' \rrbracket \hat{\zeta}(u', f) = \tau_{i,i'}(\mathcal{Z}_T, u)$$

Summing over $i' \in \{0,1\}$ and noting Equation (44) then shows us the result holds for this case and hence, by induction, holds always. $\square$

Given $u', u'' \in \mathcal{Z}_T$ with $u'' \in \Downarrow_{\mathcal{Z}_T}(u')$ we define $\hat{\mathcal{H}}(u', u'')$ to be the maximal subtree of $\mathcal{Z}_T$ which has $u'$ and $u''$ as leaves. Given, in addition, $f : \hat{\mathcal{H}}(u', u'') \to \{0,1\}$ we define:

$$\tilde{\zeta}(u', u'', f) := \prod_{u \in \hat{\mathcal{H}}(u', u'') \setminus \{u'\}} \tau_{f(\uparrow_{\mathcal{Z}_T}(u)), f(u)}(\mathcal{Z}_T, u)$$

and:

$$\delta(u', u'') := d(u'') - d(u')$$

We now have the following lemma.

**Lemma E.4.** *Given $u', u'' \in \mathcal{Z}_T$ with $u'' \in \Downarrow_{\mathcal{Z}_t}(u') \setminus \{u'\}$ and indices $i', i'' \in \{0,1\}$ we have that:*

$$\sum_{f \in \{0,1\}^{\hat{\mathcal{H}}(u', u'')}} \llbracket f(u') = i' \rrbracket \llbracket f(u'') = i'' \rrbracket \tilde{\zeta}(u', u'', f)$$

*is equal to*

$$\llbracket i' \neq i'' \rrbracket \phi_{\delta(u', u'')} + \llbracket i' = i'' \rrbracket (1 - \phi_{\delta(u', u'')})$$

*Proof.* We prove by induction on the distance from $u'$ to $u''$ in $\mathcal{Z}_T$. If this distance is one then we have $u' = \uparrow_{\mathcal{Z}_T}(u'')$ and $\hat{\mathcal{H}}(u', u'') = \{u', u''\}$ so we have:

$$\sum_{f \in \{0,1\}^{\hat{\mathcal{H}}(u', u'')}} \llbracket f(u') = i' \rrbracket \llbracket f(u'') = i'' \rrbracket \tilde{\zeta}(u', u'', f) = \tau_{i', i''}(\mathcal{Z}_T, u'')$$

which immediately implies that the inductive hypothesis holds in this case. Now suppose that the inductive hypothesis holds whenever the distance from $u'$ to $u''$ is $j$. We now consider the case

that the distance from $u'$ to $u''$ is $j + 1$. Let $u^*$ be the child of $u'$ that lies in $\hat{\mathcal{H}}(u', u'')$. Without loss of generality assume that $u''$ is a descendant of $\lhd(u^*)$. Now choose any $i^* \in \{0, 1\}$. Given $f : \hat{\mathcal{H}}(u', u'') \to \{0, 1\}$ let:

$$h(i^*, f) = [\![f(u') = i']\!][\![f(u'') = i'']\!][\![f(u^*) = i^*]\!]$$

and let $f'$ and $f''$ be the restriction of $f$ onto the sets $\hat{\mathcal{H}}(u^*, u'')$ and $\mathcal{H}(\rhd(u^*))$ respectively. Note that

$$\tilde{\zeta}(u', u'', f) = \tau_{f(u'), f(u^*)}(\mathcal{Z}_T, u^*)\tilde{\zeta}(u^*, u'', f')\hat{\zeta}(\rhd(u^*), f'')$$

By Lemma E.3 and the inductive hypothesis we then have that the quantity:

$$\sum_{f \in \{0,1\}^{\hat{\mathcal{H}}(u', u'')}} h(i^*, f)\tilde{\zeta}(u', u'', f)$$

is equal to the quantity:

$$\tau_{i', i^*}(\mathcal{Z}_T, u^*)([\![i^* \neq i'']\!]\phi_{\delta(u^*, u'')} + [\![i^* = i'']\!](1 - \phi_{\delta(u^*, u'')}))$$

Summing over $i^* \in \{0, 1\}$ gives us the result. We have hence proved the result in general. $\qquad\square$

Suppose we have some $f : \mathcal{J} \to \{0, 1\}$. Let:

$$\hat{\mathcal{F}}(f) := \{f' \in \{0, 1\}^{\mathcal{Z}_T} \mid \forall u \in \mathcal{J}, \, f'(u) = f(u)\}$$

Given $u \in \mathcal{J}$ we have that:

$$[\![f(\uparrow_{\mathcal{J}}(u)) \neq f(u)]\!]\phi_{\delta(\uparrow_{\mathcal{J}}(u), u)} + [\![f(\uparrow_{\mathcal{J}}(u)) = f(u)]\!](1 - \phi_{\delta(\uparrow_{\mathcal{J}}(u), u)})$$

is equal to $\tau_{f(\uparrow_{\mathcal{J}}(u)), f(u)}(\mathcal{J}, u)$ and hence Lemma E.4 implies that:

$$\sum_{f' \in \hat{\mathcal{H}}(\uparrow_{\mathcal{J}}(u), u)} [\![f'(\uparrow_{\mathcal{J}}(u)) = f(\uparrow_{\mathcal{J}}(u))]\!][\![f'(u) = f(u)]\!]\tilde{\zeta}(\uparrow_{\mathcal{J}}(u), u, f') = \tau_{f(\uparrow_{\mathcal{J}}(u)), f(u)}(\mathcal{J}, u)$$

so since, by the definition of a contraction, the edge sets of the subtrees in $\{\hat{\mathcal{H}}(\uparrow_{\mathcal{J}}(u), u) \mid u \in \mathcal{J} \setminus \{r(\mathcal{J})\}\}$ partition the edge set of $\mathcal{Z}_T$ we have, by definition of $\tilde{\zeta}$, that:

$$\sum_{f' \in \hat{\mathcal{F}}(f)} \prod_{u \in \mathcal{Z}_T \setminus \{r(\mathcal{Z}_T)\}} \tau_{f'(\uparrow_{\mathcal{Z}_T}(u)), f'(u)}(\mathcal{Z}_T, u) = \prod_{u \in \mathcal{J} \setminus \{r(\mathcal{J})\}} \tau_{f(\uparrow_{\mathcal{J}}(u)), f(u)}(\mathcal{J}, u)$$

Since for all $f' \in \hat{\mathcal{F}}(f)$ and for all $u \in \mathcal{Z}_T \setminus \mathcal{J}$ we have $\tilde{\kappa}_{f'(u)}(\mathcal{Z}_T, \hat{\lambda}_T, u) = 1$ we have now shown that the quantity:

$$\sum_{f' \in \hat{\mathcal{F}}(f)} \prod_{u \in \mathcal{Z}_T \setminus \{r(\mathcal{Z}_T)\}} \tau_{f'(\uparrow_{\mathcal{Z}_T}(u)), f'(u)}(\mathcal{Z}_T, u)\tilde{\kappa}_{f'(u)}(\mathcal{Z}_T, \hat{\lambda}_T, u)$$

is equal to the quantity:

$$\prod_{u \in \mathcal{J} \setminus \{r(\mathcal{J})\}} \tau_{f(\uparrow_{\mathcal{J}}(u)), f(u)}(\mathcal{J}, u)\tilde{\kappa}_{f(u)}(\mathcal{Z}_T, \hat{\lambda}_T, u)$$

Summing over all $f \in \mathcal{F}(\mathcal{J}, \hat{u})$ and noting Equations (42) and (43) gives us the result.

## E.7 Theorem D.1

**Lemma E.5.** *Given $u, u' \in \mathcal{Z}_t$ the algorithm for computing $\nu(u, u')$ is correct.*

*Proof.* If $u = u'$ then the proof is trivial. Otherwise we consider the following cases:

- Consider first the case that $\hat{s} \neq \triangledown(s^*)$. Without loss of generality assume $\hat{s} = \lhd(s^*)$. Then we have $u \in \Downarrow(\lhd(\xi(s^*)))$ and since $\hat{s}' \neq \lhd(s^*)$ we have $u' \notin \Downarrow(\lhd(\xi(s^*)))$. Hence $u' \notin \Downarrow(u)$ so $\nu(u, u') = \blacktriangle$ as required.

- If $u = \xi(s^*)$ then $\hat{s} = \triangledown(s^*)$ so either $\hat{s}' = \triangleleft(s^*)$ or $\hat{s}' = \triangleright(s^*)$. In the former case we have $u' \in \Downarrow(\triangleleft(\xi(s^*))) = \Downarrow(\triangleleft(u))$ so that $\nu(u, u') = \blacktriangleleft$ and similarly in the later case we have $\nu(u, u') = \blacktriangleright$ as required.

- If $\hat{s} = \triangledown(s^*)$ and $u \neq \xi(s^*)$ then we invoke the process. Consider the vertex $s$ at any stage in the process. By induction we have that if $s \in \mathcal{E}^\bullet$ then $u' \in \Downarrow(\mu'(s))$. This is because if $s \in \mathcal{E}^\bullet$ then $\mu'(s)$ is an ancestor of $\mu'(\uparrow_\mathcal{E}(s))$. This further implies that when $s \neq \hat{s}$ we have $u' \in \Downarrow(\mu'(\uparrow_\mathcal{E}(s)))$. Now suppose that $s \in \mathcal{E}^\circ$ and without loss of generality assume $s = \triangleleft(\uparrow_\mathcal{E}(s))$. Then $u \in \Downarrow(\triangleleft(\xi(\uparrow_\mathcal{E}(s))))$ and $\mu'(\uparrow_\mathcal{E}(s)) \in \Downarrow(\triangleright(\xi(\uparrow_\mathcal{E}(s))))$ so that, since $u' \in \Downarrow(\mu'(\uparrow_\mathcal{E}(s)))$, we have $u' \notin \Downarrow(u)$ and hence $\nu(u, u') = \blacktriangle$ as required. Suppose now that $s \in \mathcal{E}^\bullet$ and that $u = \xi(s)$. If $\triangleleft(s) \in \mathcal{E}^\bullet$ then we have $\mu'(s) \in \Downarrow(\triangleleft(\xi(s))) = \Downarrow(\triangleleft(u))$ so that, by above, $u' \in \Downarrow(\triangleleft(u))$ and hence $\nu(u, u') = \blacktriangleleft$ as required. Similarly, if $\triangleright(s) \in \mathcal{E}^\bullet$ then $\nu(u, u') = \blacktriangleright$ as required. This completes the proof.

$\square$

**Lemma E.6.** *The algorithm correctly finds $\hat{u}$.*

*Proof.* By induction on the depth of $s$ we have, for all vertices $s$ in the constructed path, that:

- If $s \in \mathcal{D}^\circ$ then $u_t$ lies in the maximal subtree of $\mathcal{Z}_t$ containing $\mu(s)$ and having $\uparrow_\mathcal{J}(\mu(s))$ as a leaf .

- If $s \in \mathcal{D}^\bullet$ then $u_t$ lies in the maximal subtree of $\mathcal{Z}_t$ with $\uparrow_\mathcal{J}(\mu(s))$ and $\mu'(s)$ as leaves.

Let $s'$ be the unique leaf of $\mathcal{D}$ that is on the constructed path. If $s' \in \mathcal{D}^\circ$ then $\mu(s')$ is a leaf of $\mathcal{J}$ and hence also a leaf of $\mathcal{Z}_t$. So by above we have that $u_t$ lies in the maximal subtree of $\mathcal{Z}_t$ with $\uparrow_\mathcal{J}(\mu(s'))$ and $\mu(s')$ as leaves. If, on the other hand, $s' \in \mathcal{D}^\bullet$ then since $s'$ is a leaf of $\mathcal{D}$ we have that $\mu(s') = \mu'(s')$ and hence, by above, we have that $u_t$ lies in the maximal subtree of $\mathcal{Z}_t$ with $\uparrow_\mathcal{J}(\mu(s'))$ and $\mu(s')$ as leaves. In either case we have $\hat{u} = \mu(s')$ as required. $\square$

**Lemma E.7.** *The algorithm correctly finds $u^*$.*

*Proof.* By induction on the depth of $s$ we have, for all vertices $s$ in the constructed path, that:

- If $s \in \mathcal{E}^\circ$ then $\Gamma_{\mathcal{Z}_t}(u_t, \hat{u})$ lies in $\Downarrow_{\mathcal{Z}_t}(\mu(s))$.

- If $s \in \mathcal{E}^\bullet$ then $\Gamma_{\mathcal{Z}_t}(u_t, \hat{u})$ lies in the maximal subtree of $\mathcal{Z}_t$ with $\mu(s)$ and $\mu'(s)$ as leaves.

Let $s'$ be the unique leaf of $\mathcal{E}$ that is on the constructed path. If $s' \in \mathcal{E}^\circ$ then $\mu(s')$ is a leaf of $\mathcal{Z}_t$ and hence, by above, $\Gamma_{\mathcal{Z}_t}(u_t, \hat{u}) = \mu(s')$ as required. If $s \in \mathcal{E}^\bullet$ then $\mu(s) = \mu'(s)$ and hence, by above, $\Gamma_{\mathcal{Z}_t}(u_t, \hat{u}) = \mu(s')$ as required. $\square$

