# Nearest Neighbour with Bandit Feedback

## Abstract

In this paper we adapt the nearest neighbour rule to the contextual bandit problem. Our algorithm handles the fully adversarial setting in which no assumptions at all are made about the data-generation process. When combined with a sufficiently fast data-structure for (perhaps approximate) adaptive nearest neighbour search, such as a navigating net, our algorithm is extremely efficient - having a per trial running time polylogarithmic in both the number of trials and actions, and taking only quasi-linear space. We give generic regret bounds for our algorithm and further analyse them in a semi-stochastic setting. A side result of this paper is that, when applied to the online classification problem with stochastic labels, our algorithm can have sublinear regret whilst only finding a single nearest neighbour per trial - in stark contrast to the k-nearest neighbours algorithm.

## 1 Introduction

In this paper we adapt the classic *nearest neighbour* rule to the contextual bandit problem and develop an extremely efficient algorithm. The problem proceeds in trials, where on trial $t$: (1) a *context* $x_t$ is revealed to us, (2) we must select an *action* $a_t$, and (3) the *loss* $\ell_{t,a_t} \in [0,1]$ of action $a_t$ on trial $t$ is revealed to us. We assume that the contexts are points in a metric space and the distance between two contexts represents their similarity. A *policy* is a mapping from contexts to actions and the inductive bias of our algorithm is towards learning policies that typically map similar contexts to similar actions. Our main result has absolutely no assumptions whatsoever about the generation of the context/loss sequence and has no restriction on what policies we can compare our algorithm to.

Our algorithm requires, as a subroutine, a data-structure that performs $c$-nearest neighbour search. This data-structure must be *adaptive* - in that new contexts can be inserted into it over time. An example of such a data-structure is the *Navigating net* [15] which, given mild conditions on our metric and dataset, performs both search and insertion in polylogarithmic time. When utilising a data-structure of this speed our algorithm is extremely efficient - with a per-trial time complexity polylogarithmic in both the number of trials and actions, and requiring only quasi-linear space.

As an example we will further analyse the special case of the contextual bandit problem in which the context sequence is drawn i.i.d. from a probability distribution over the $d$-dimensional hypercube, whilst the loss vectors can still be arbitrary. In this case, for any policy $y$ with a finite-volume decision boundary, our algorithm achieves sub-linear regret w.r.t. $y$ without the need to know any parameters.

A side result of this paper is that, when applied to the online classification task with stochastic labels, our algorithm can achieve sublinear regret whilst only finding *one* nearest neighbour per trial: in stark contrast to the $k$-nearest neighbour algorithm. Our algorithm can hence be viewed also as a potentially faster alternative to $k$-nearest neighbours when faced with the online classification problem.

In the course of this paper we develop some novel algorithmic techniques, including a new algorithmic framework CANPROP and efficient algorithms for searching in trees, which may find further application.

We now describe related works. The bandit problem [17] was first introduced in [22] but was originally studied in the stochastic setting in which all losses are drawn i.i.d. at random [16], [1], [2]. However, our world is very often not i.i.d. stochastic. The work of [3] introduced the seminal EXP3 algorithm which handled the case in which the losses were selected arbitrarily. This work also introduced the EXP4 algorithm for contextual bandits. In general this algorithm is exponential time but in some situations can be implemented in polynomial time - such as their EXP3.S algorithm, which was a bandit version of the classic FIXEDSHARE algorithm [13]. In [11] the EXP3.S setting was greatly generalised to the situation in which the contexts where vertices of a graph. They utilised the methodology of [7], [14] and [12] in order to develop extremely efficient algorithms. Although inspiring this work, these algorithms cannot be utilised in our situation as they inherently require the set of queried contexts to be known a-priori. In the stochastic case another class of contextual bandit problems are *linear bandits* [18], [4] in which the contexts are mappings from the actions into $\mathbb{R}^d$. Here the queried contexts need not be known in advance but the losses must be drawn i.i.d. from a distribution that has mean linear in the respective context. The $k$ nearest neighbour algorithm was first analysed in [5]. The work [21] utilised the $k$ nearest neighbour methodology and the works [8] and [16] to handle a stochastic contextual bandit problem. However, their setting is extremely more restricted than ours. In particular, the context/loss pairs must be drawn i.i.d. at random and the probability distribution they are sampled from must obey certain strict conditions. In addition, on each trial the contexts seen so far must be ordered in increasing distance from the current context and operations must be performed on this sequence, making their algorithm exponentially slower than ours. Our algorithm utilises the works of [19] and [6] as subroutines. It should be noted that the later work, which was based on [20], was improved on in [10] - we leave it as an open problem as to whether we can utilise their work in our algorithm.

# 2 Notation

Let $\mathbb{N}$ be the set of natural numbers not including $0$. Given a natural number $m \in \mathbb{N}$ we define $[m] := \{j \in \mathbb{N} \mid j \leq m\}$. Given a predicate $p$ we define $[\![p]\!] := 1$ if $p$ is true and $[\![p]\!] := 0$ otherwise. We define $\log(\cdot)$ and $\ln(\cdot)$ to be the logarithms with base $2$ and $e$ respectively. Given sets $\mathcal{A}$ and $\mathcal{B}$ we denote by $\mathcal{B}^{\mathcal{A}}$ the set of all functions $f : \mathcal{A} \to \mathcal{B}$ and by $2^{\mathcal{A}}$ the set of all subsets of $\mathcal{A}$.

All trees in this paper are considered rooted. Given a tree $\mathcal{J}$ we denote its root by $r(\mathcal{J})$, its vertex set by $\mathcal{J}$, its leaves by $\mathcal{J}^\star$, and its internal vertices by $\mathcal{J}^\dagger$. Given a vertex $v$ in a tree $\mathcal{J}$ we denote its parent by $\uparrow_{\mathcal{J}}(v)$ and the subtree of all its descendants by $\Downarrow_{\mathcal{J}}(v)$. Given an internal node $v$ in a (full) binary tree $\mathcal{J}$ we denote its left and right children by $\triangleleft_{\mathcal{J}}(v)$ and $\triangleright_{\mathcal{J}}(v)$ respectively. Internal nodes $v$ in a (full) ternary tree $\mathcal{J}$ have an additional child $\triangledown_{\mathcal{J}}(v)$ called the *centre* child. Given vertices $v$ and $v'$ in a tree $\mathcal{J}$ we denote by $\Gamma_{\mathcal{J}}(v, v')$ the *least common ancestor* of $v$ and $v'$: i.e. the vertex of maximum depth which is an ancestor of both $v$ and $v'$. We will drop the subscript $\mathcal{J}$ in all these functions when unambiguous. Given a tree $\mathcal{J}$, a *subtree* of $\mathcal{J}$ is a tree whose edge set is a subset of that of $\mathcal{J}$.

Given a probability distribution $\mu$ we write $x \sim \mu$ to signify that $x$ is a random element drawn from $\mu$. Given, in addition, some $m \in \mathbb{N}$, we define $\mu^m$ to be the probability distribution over sets formed by drawing $m$ elements i.i.d. with replacment from distribution $\mu$. With a slight overloading of notation we denote the uniform distribution over $[0, 1]$ also by $[0, 1]$.

# 3 Problem and Result

## 3.1 The Contextual Bandit Problem

We consider the following game between *Learner* (us) and *Nature* (our adversary). We have $K$ *actions* and a metric space $(\mathcal{C}, \Delta)$ where $\mathcal{C}$ is a (possibly infinite) set of *contexts* and for all $x, x' \in \mathcal{C}$ we have that $\Delta(x, x')$ is the *distance* from $x$ to $x'$. Learning proceeds in $T$ trials. A-priori Nature chooses a sequence of contexts $\mathcal{X} = \{x_t \mid t \in [T]\} \subseteq \mathcal{C}$ and a sequence of loss vectors $\{\boldsymbol{\ell}_t \mid t \in [T]\} \subseteq [0, 1]^K$, but does not reveal them to Learner. On the $t$-th trial the following happens:

    1. Nature reveals $x_t$ to Learner.

    2. Learner chooses some action $a_t \in [K]$.

    3. Nature reveals $\ell_{t,a_t}$ to Learner.

A *policy* is a function from $\mathcal{C}$ into $[K]$. i.e. a policy associates each possible context with an action. Given a policy $y : \mathcal{C} \to [K]$ we define the *y-regret* of Learner as:

$$R(y) := \sum_{t \in [T]} \ell_{t,a_t} - \sum_{t \in [T]} \ell_{t,y(x_t)}$$

which is the difference between the total cumulative loss suffered by Learner and that which Learner would have suffered if it had instead chosen $a_t$ equal to $y(x_t)$ for all trials $t$.

## 3.2  The ($k$) Nearest Neighbour Classifier

We now digress from the contextual bandit problem in order to study the nearest neighbour methodology. The *nearest neighbour classifier* is a method to solve the following supervised learning problem. We assume that there exists an unknown function $y : \mathcal{C} \to [K]$. We are given a finite set $\mathcal{S} \subseteq \mathcal{C}$ along with the restriction of $y$ onto $\mathcal{S}$. We are then asked to predict the value of $y(x)$ for some given $x \in \mathcal{C}$. The nearest neighbour classifier works by first finding the nearest neighbour $\hat{x}$ of $x$, defined as:

$$\hat{x} := \operatorname{argmin}_{x' \in \mathcal{S}} \Delta(x, x'),$$

and then choosing $y(\hat{x})$ as its prediction of $y(x)$. In many important metric spaces the time taken to find such a nearest neighbour is in $\Theta(|\mathcal{S}|)$. This fact has lead to the idea of instead using $y(\tilde{x})$ as our prediction, where $\tilde{x} \in \mathcal{S}$ is an arbitrary *c-nearest neighbour* which is defined as satisfying:

$$\Delta(x, \tilde{x}) \leq c \min_{x' \in \mathcal{S}} \Delta(x, x').$$

By utilising special data-structures the time complexity of finding, for any fixed $c > 1$, such a $c$-nearest neighbour is, for many metric spaces, only polylogarithmic in $|\mathcal{S}|$.

Given a probability distribution $\mu$ over $\mathcal{C}$, some $c \geq 1$ and some $m \in \mathbb{N}$ we define the *generalisation error* as:

$$g_m(\mu, y, c) := \mathbb{P}_{\mathcal{S} \sim \mu^m, x \sim \mu} \left[ \exists z \in \mathcal{S} : \left( \Delta(x, z) \leq c \min_{x' \in \mathcal{S}} \Delta(x, x') \right) \wedge y(z) \neq y(x) \right]$$

which is the probability that it is possible for the nearest neighbour classifier to make a mistake on a context drawn from $\mu$ when $\mathcal{S}$ is formed by drawing $m$ contexts i.i.d. from $\mu$.

We will now bound this quantity when in euclidean space. We first make the following definitions. For any $\delta > 0$ define the $\delta$-*margin* of $y$ by:

$$\mathcal{M}(y, \delta) := \{x \in \mathcal{C} \mid \exists x' \in \mathcal{C} : \Delta(x, x') \leq \delta \wedge y(x) \neq y(x')\} \tag{1}$$

which is the set of contexts that are at distance no more than $\delta$ from the *decision boundary* of $y$. The *volume* (w.r.t. $\mu$) of the decision boundary is then given by:

$$\alpha(y, \mu) := \lim_{\delta \to 0} \frac{\mu(\mathcal{M}(y, \delta))}{\delta}. \tag{2}$$

When in euclidean space the following theorem bounds the generalisation error:

**Theorem 3.1.** *Given* $\mathcal{C} := [0,1]^d$ *and* $\Delta$ *is the euclidean metric then for any* $y : \mathcal{C} \to [K]$, $c \geq 1$, $\epsilon > 0$, *and probability distribution* $\mu$ *such that the probability density of* $\mu$ *is always at least* $\epsilon$, *we have:*

$$g_m(\mu, y, c) \in \tilde{\mathcal{O}}\left( c\,\alpha(y, \mu)\,(\epsilon m)^{-1/d} \right).$$

So far we have only considered deterministic functions $y : \mathcal{C} \to [K]$ with decision boundaries of finite volume. But what happens if instead we have that $y(x)$ is drawn from some probability distribution dependent on $x$ (which is itself drawn from $\mu$). Here, the *Bayes optimal classifier* is defined as that which always predicts $y^*(x) := \operatorname{argmax}_{a \in [K]} \mathbb{P}[y(x) = a | x]$ as the label of $x$. In general, even if $g_m(\mu, y^*, c) \in o(1)$, the probability of making a mistake with the nearest neighbour classifier does not approach that of the Bayes optimal classifier as $m \to \infty$. In order to converge optimally, the $k$-nearest neighbour classifier was introduced. In this algorithm, when given a context $x \in \mathcal{C}$, the $k$ nearest neighbours to $x$ are found and the predicted value of $y(x)$ is decided by majority vote. In order to converge optimally we need that $k \to \infty$ as $m \to \infty$.

A remarkable side-result of this paper is that, given $g_m(\mu, y^*, c) \in \mathcal{O}(m^{-p})$ for some $p > 0$, our algorithm can be applied to learning this situation online whilst only finding *one* nearest neighbour per trial. Since the additional overhead of our algorithm is so small it can be significantly faster than $k$-nearest neighbours. We strongly suspect that we don't need the condition on $g_m(\mu, y^*, c)$ if we are working in a bounded subset of euclidean space and $\mathbb{P}[y(x) = a | x]$ is Lipschitz.

### 3.3 Adaptive Nearest Neighbour Search

Our algorithm will require a data-structure for performing *adaptive nearest neighbour search*. This problem is as follows. We maintain a finite set $\mathcal{S} \subseteq \mathcal{C}$. At any point in time we must either (1) insert a new context into the set $\mathcal{S}$ and update the data-structure, or (2) given a context, utilise the data-structure to find a $c$-nearest neighbour in the set $\mathcal{S}$.

We are especially interested in data-structures that can do both operations in a time polylogarithmic in $|\mathcal{S}|$. An example of such a data-structure is the *navigating net* [15] which has time complexity $\tilde{\mathcal{O}}(\ln(|\mathcal{S}|))$ given that $c > 1$, the set $|\mathcal{S}|$ is of bounded doubling dimension (w.r.t. $\Delta$) and has aspect ratio (ratio between the largest and smallest distances between contexts in $\mathcal{S}$) polynomial in $|\mathcal{S}|$, as is the case in many applications and can be enforced by quantisation when working in a bounded subset of euclidean space. We note that the $\tilde{\mathcal{O}}$ hides a constant factor that is exponential in the doubling dimension of $\mathcal{S}$. In high-dimensional applications our dataset will often lie on a low-dimensional manifold so this factor should be small.

### 3.4 Our Results

We now turn back to the contextual bandit problem and give our main results.

**Theorem 3.2.** *Consider the contextual bandit problem defined in Section 3.1. Suppose that for all trials $t > 1$ we are given, in addition to $x_t$, a context $n(x_t)$ which satisfies:*

$$n(x_t) \in \{x_s \mid s \in [t-1]\}.$$

*Our algorithm* CBNN *takes a single parameter $\rho > 0$ and, for all policies $y : \mathcal{C} \to [K]$ simultaneously, obtains an expected $y$-regret bounded by:*

$$\mathbb{E}[R(y)] \in \tilde{\mathcal{O}}\left(\left(\rho + \frac{\Phi(y)}{\rho}\right)\sqrt{KT}\right)$$

*where:*

$$\Phi(y) := 1 + \sum_{t \in [T] \setminus \{1\}} [\![y(x_t) \neq y(n(x_t))]\!]$$

*and the expectation is taken over the randomisation of the algorithm.* CBNN *needs no initialisation time and has a per-trial time complexity of:*

$$\mathcal{O}(\ln(T)^2 \ln(K)).$$

We note that, although $n$ can be any valid function, we are particularly interested in the case that $n(x_t)$ is a $c$-nearest neighbour of $x_t$. i.e. that we have:

$$\Delta(x_t, n(x_t)) \leq c \min_{s \in [t-1]} \Delta(x_t, x_s). \tag{3}$$

In this case finding $n(x_t)$ typically requires only $\tilde{\mathcal{O}}(\ln(T))$ time per trial when using a navigating net or other fast data-structure for adaptive nearest neighbour search, as explained in Section 3.3. Furthermore, the quantity $\Phi(y)$ can now be bounded in a way that is dependent only on the set of queried contexts $\mathcal{X}$ and not their order. This bound is given in the following theorem.

**Theorem 3.3.** *Suppose we have a policy $y : \mathcal{C} \to [K]$. For any context $x \in \mathcal{C}$ we define $\gamma(x, y) := \max\{\delta \geq 0 \mid x \notin \mathcal{M}(y, \delta)\}$ which is the distance of $x$ from the decision boundary of $y$. Then when Equation (3) is satisfied we have that $\Phi(y)$ is no greater than the minimum cardinality of any set $\mathcal{S} \subseteq \mathcal{C}$ in which for all $t \in [T]$ there exists $x \in \mathcal{S}$ with $\Delta(x, x_t) < \gamma(x, y)/3c$.*

A direct corollary of this theorem is that for all $\delta > 0$ we have that:

$$\Phi(y) \leq N_\delta(\mathcal{X}) + |\mathcal{X} \cap \mathcal{M}(4c\delta, y)|$$

where $N_\delta(\mathcal{X})$ is the (external) covering number of $\mathcal{X}$ with radius $\delta$, and $|\mathcal{X} \cap \mathcal{M}(4c\delta, y)|$ is the number of contexts in $\mathcal{X}$ lying within distance $4c\delta$ of the decision boundary.

It will be common in applications that the set $\mathcal{X}$ of observed contexts will come from a finite set of seperated clusters and there will be a good policy $y$ which, on any such cluster, is constant on that cluster. Theorem 3.3 then implies that, as long as the inter-cluster distances are positive and the

clusters have finite covering numbers (which is guaranteed in many metric spaces), then $\Phi(y)$ will be bounded independent of $T$ and hence, by Theorem 3.2, the $y$-regret of CBNN will scale like $\tilde{\mathcal{O}}(\sqrt{T})$, whatever the value of $\rho$.

However, it will not always be the case that the dataset splits into such clusters. We shall investigate what happens when this is not the case by restricting ourselves to the situation in which the contexts $\{x_t \mid t \in [T]\}$ are drawn i.i.d. from a probability distribution $\mu$. Here we have, by linearity of expectation, that:

$$\mathbb{E}[\Phi(y)] \le 1 + \sum_{t \in [T]} g_t(\mu, y, c).$$

When in euclidean space, theorems 3.1 and 3.2 then lead to the following theorem:

**Theorem 3.4.** *Consider the contextual bandit problem defined in Section 3.1. Suppose that $\mathcal{C} = [0,1]^d$, $\Delta$ is the euclidean metric, and the contexts are drawn i.i.d. at random from a probability distribution $\mu$ with density always at least $\epsilon > 0$. Note that the loss vectors can be arbitrary. Set $\rho$ equal to $T^{-(d-1)/d}c^{-1/2}$. Then when Equation (3) is satisfied we have, for all policies $y : \mathcal{C} \to [K]$ simultaneously, that the $y$-regret of* CBNN *is bounded by:*

$$\mathbb{E}[R(y)] \in \tilde{\mathcal{O}}\left((\epsilon^{-1/d}\alpha(y,\mu) + 1)c^{1/2}K^{1/2}T^{(2d-1)/(2d)}\right)$$

*where $\alpha(y,\mu)$ is the volume (w.r.t. $\mu$) of the decision boundary of $y$ as defined in equations (1) and (2). The existence of such an $\epsilon$ can be relaxed (with an effect on the bound) but we assume it for simplicity.*

Note that, given the decision boundary of $y$ is of finite volume, the expected regret is guaranteed to be sub-linear in $T$. This implies that the per-trial performance of CBNN approaches that of always selecting $a_t = y(x_t)$. We note that if $T$ is unknown or infinite (i.e. learning never stops) then a simple doubling trick can be used to make the algorithm parameter-free (with no knowledge of $\mu$). The fact that, in this non-separated case, the regret scales like $\tilde{\mathcal{O}}(T^{(2d-1)/(2d)})$ is a facet of the well known *curse of dimensionality* and is the price we pay for being able to learn from such a vast class of policies. We note that in many high-dimensional applications the set $\mathcal{X}$ will lie on a low-dimensional manifold so that the curse of dimensionality will be significantly reduced.

# 4 The Algorithm

In this section we describe our algorithm CBNN and give the pseudocode for the novel subroutines. In appendices C to E we give a more detailed description of how CBNN works, and prove that it obtains its bounds.

To give the reader intuition, in Appendix B we describe our initial idea - an algorithm, based on EXP4 [3] and *Belief propagation* [20], which attains our regret bound but is exponentially slower - taking a per-trial time of $\tilde{\Theta}(KT)$.

## 4.1 Cancellation Propagation

In this subsection we describe a novel algorithmic framework CANPROP for designing contextual bandit algorithms with a running time logarithmic in $K$. It is inspired by EXP3 [3], specialist algorithms [7] and online decision-tree pruning algorithms [9] but is certainly not a simple combination of these works. CBNN will be an efficient implementation of an instance of CANPROP. Although in general CANPROP requires a-priori knowledge of the set $\mathcal{X} := \{x_t \mid t \in [T]\}$, CBNN is designed in a way that, crucially, does not need this set to be known.

We assume, without loss of generality, that $K$ and $T$ are integer powers of two. CANPROP, which takes a parameter $\eta > 0$, works on a full, balanced binary tree $\mathcal{B}$ with leaves $\mathcal{B}^\star = [K]$. On every trial $t$ each pair $(v, \mathcal{S}) \in \mathcal{B} \times 2^{\mathcal{X}}$ has a weight $w_t(v, \mathcal{S}) \in [0,1]$. These weights induce a function $\theta_t : \mathcal{B} \to [0,1]$ defined by:

$$\theta_t(v) := \sum_{\mathcal{S} \in 2^{\mathcal{X}}} [\![x_t \in \mathcal{S}]\!] w_t(v, \mathcal{S}).$$

On each trial $t$ a root-to-leaf path $\{z_{t,j} \mid j \in [\log(K)] \cup \{0\}\}$ is sampled such that, given $z_{t,j}$, we have that $z_{t,(j+1)}$ is sampled from $\{\triangleleft(z_{t,j}), \triangleright(z_{

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

}}(\triangleleft(s)) = \hat{\mathcal{J}}(s) \cap \Downarrow(\triangleleft(\xi(s)))$ and $\hat{\mathcal{J}}(\triangleright(s)) = \hat{\mathcal{J}}(s) \cap \Downarrow(\triangleright(\xi(s)))$ and $\hat{\mathcal{J}}(\triangledown(s)) = \hat{\mathcal{J}}(s) \setminus (\hat{\mathcal{J}}(\triangleleft(s)) \cup \hat{\mathcal{J}}(\triangleright(s)))$. i.e. $\xi(s)$ partitions the subtree $\hat{\mathcal{J}}(s)$ into the subtrees $\hat{\mathcal{J}}(\triangleleft(s))$, $\hat{\mathcal{J}}(\triangleright(s))$, and $\hat{\mathcal{J}}(\triangledown(s))$. This process continues recursively until $|\hat{\mathcal{J}}(s)| = 1$ in which case $s$ is a leaf of $\mathcal{D}$.

240 For all binary trees $\mathcal{J}$ in our algorithm we shall maintain a TST $\mathcal{H}(\mathcal{J})$ of $\mathcal{J}$ with height $\mathcal{O}(\ln(|\mathcal{J}|))$.
241 Such trees $\mathcal{J}$ are *dynamic* in that on any trial it is possible that two vertices, $u$ and $u'$, are added to
242 the tree $\mathcal{J}$ such that $u'$ is inserted between a non-root vertex of $\mathcal{J}$ and its parent, and $u$ is designated
243 as a child of $u'$. We define the subroutine REBALANCE$(\mathcal{H}(\mathcal{J}), u)$ as one which rebalances the TST
244 $\mathcal{H}(\mathcal{J})$ after this insertion, so that the height of $\mathcal{H}(\mathcal{J})$ always remains in $\mathcal{O}(\ln(|\mathcal{J}|))$. The work of
245 [19] describes how this subroutine can be implemented in a time of $\mathcal{O}(\ln(|\mathcal{J}|))$ and we refer the
246 reader to this work for details (noting that they use different notation).

## 4.3 Contractions

248 At any trial $t$ the contexts in $\{x_s \mid s \in [t]\}$ naturally form a tree by designating $n(x_s)$ as the parent
249 of $x_s$. However, to utilise the TST data-structure we must only have binary trees. Hence, we will
250 work with a (dynamic) full binary tree $\mathcal{Z}$ which, on trial $t$, is a *binarisation* of the above tree. The
251 relationship between these two trees is given by a map $\gamma : \mathcal{Z}_t \to \{x_s \mid s \in [t]\}$ where $\mathcal{Z}_t$ is the tree
252 $\mathcal{Z}$ on trial $t$. For all $x \in \{x_s \mid s \in [t]\}$ we will always have an unique leaf $\tilde{\gamma}(x) \in \mathcal{Z}_t^\star$ in which
253 $\gamma(\tilde{\gamma}(x)) = x$. We also maintain a balanced TST $\mathcal{H}(\mathcal{Z})$ of $\mathcal{Z}$.

254 Algorithm 2 gives the subroutine GROW$_t$ which updates $\mathcal{Z}$ at the start of trial $t$. Note that GROW$_t$
255 also defines a function $d : \mathcal{Z} \to \mathbb{N}$ such that $d(u)$ is the number of times the function $n$ must be
256 applied to $\gamma(u)$ to reach $x_1$.

---

**Algorithm 2** GROW$_t$ which works on $\mathcal{Z}$

---

1: $u \leftarrow \tilde{\gamma}(n(x_t))$     10: **else**
2: $u^* \leftarrow \uparrow(u)$     11:     $\triangleright(u^*) \leftarrow u'$
3: $u' \leftarrow$ NEWVERTEX     12: **end if**
4: $u'' \leftarrow$ NEWVERTEX     13: $\triangleleft(u') \leftarrow u''$
5: $\gamma(u') \leftarrow n(x_t)$     14: $\triangleright(u') \leftarrow u$
6: $\gamma(u'') \leftarrow x_t$     15: $d(u') \leftarrow d(u)$
7: $\tilde{\gamma}(x_t) \leftarrow u''$     16: $d(u'') \leftarrow d(u) + 1$
8: **if** $u = \triangleleft(u^*)$ **then**     17: REBALANCE$(\mathcal{H}(\mathcal{Z}), u'')$

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

52:          **else**
53:              $\tau_{i,i'}(\mathcal{J}, u, j) \leftarrow \phi_{\delta(u)}(2^j/T)$
54:          **end if**
55:      **end for**
56: **end for**
57: $\text{REBALANCE}(\mathcal{H}(\mathcal{J}), u_t)$

Given a vertex $u$ in one of our contractions $\mathcal{J}$ we define $\mathcal{F}(\mathcal{J}, u) := \{f \in \{0,1\}^{\mathcal{J}} \mid f(u) = 1\}$ and then for all $j \in [\log(T)]$ define:

$$\Lambda(\mathcal{J}, u, j) := \prod_{f \in \mathcal{F}(\mathcal{J}, u)} \prod_{u' \in \mathcal{J} \setminus \{r(\mathcal{J})\}} \tau_{f(\uparrow_{\mathcal{J}}(u')), f(u')}(\mathcal{J}, u', j) \kappa_{f(u')}(\mathcal{J}, u') \,.$$

As stated in the previous subsection, when a vertex $v \in \mathcal{B}$ becomes involved in CANPROP on trial $t$, CBNN will add $\tilde{\gamma}(x_t)$ to the leaves of $\mathcal{A}(v)$ via the operation INSERT$_t(\mathcal{A}(v))$. In the appendix we shall show that for each such $v$ we then have:

$$\theta_t(v) = \frac{1}{4\log(T)} \sum_{j \in [\log(T)]} \Lambda(\mathcal{A}(v), \tilde{\gamma}(x_t), j) \,.$$

We now outline how to compute this efficiently, deferring a full description for Appendix E.3. First note that for all contractions $\mathcal{J}$ and all $(j, u) \in [\log(T)] \times \mathcal{J}$ we have that $\Lambda(\mathcal{J}, u, j)$ is of the exact form to be solved by the classic *Belief propagation* algorithm [20]. The work of [6] shows how to compute this term in logarithmic time by maintaining a data-structure based on a balanced TST of $\mathcal{J}$ - in our case the TST $\mathcal{H}(\mathcal{J})$. Whenever, for some $i \in \{0,1\}$ and $u' \in \mathcal{J}$, the value $\kappa_i(\mathcal{J}, u')$ changes, the data-structure is updated in logarithmic time.

We shall maintain, for each contraction $\mathcal{J}$, a set of $\log(T)$ such data-structures - one for each value of $j$. We define the subroutine EVIDENCE$(\mathcal{J}, u')$ as that which updates all these data-structures after $\kappa_i(\mathcal{J}, u')$ changes. We also make sure that the data-structures are updated whenever REBALANCE$(\mathcal{H}(\mathcal{J}), \cdot)$ is called. We then define the subroutine MARGINAL$(\mathcal{J}, u)$ as that which computes $\Lambda(\mathcal{J}, u, j)$ for each $j \in [\log(T)]$, and then sums the results and divides by $4\log(T)$. Hence, the output of MARGINAL$(\mathcal{A}(v), \tilde{\gamma}(x_t))$ is equal to $\theta_t(v)$.

## 4.5 CBNN

Now that we have defined all our subroutines we give, in Algorithm 5, the algorithm CBNN which is an efficient implementation of CANPROP with initial weighting given in Equation (4).

---

**Algorithm 5** CBNN at trial $t$

---

1: GROW$_t$
2: $u_t \leftarrow \tilde{\gamma}(x_t)$
3: $v_{t,0} \leftarrow r(\mathcal{B})$
4: **for** $j = 0, 1, \cdots, (\log(K) - 1)$ **do**
5:     **for** $v \in \{\lhd(v_{t,j}), \rhd(v_{t,j})\}$ **do**
6:         INSERT$_t(\mathcal{A}(v))$
7:         $\theta_t(v) \leftarrow$ MARGINAL$(\mathcal{A}(v), u_t)$
8:     **end for**
9:     $z_{t,j} \leftarrow \theta_t(\lhd(v_{t,j})) + \theta_t(\rhd(v_{t,j}))$
10:     **for** $v \in \{\lhd(v_{t,j}), \rhd(v_{t,j})\}$ **do**
11:         $\pi_t(v) \leftarrow \theta_t(v)/z_{t,j}$
12:     **end for**
13:     $\zeta_{t,j} \sim [0,1]$
14:     **if** $\zeta_{t,j} \leq \pi_t(\lhd(v_{t,j}))$ **then**
15:         $v_{t,j+1} \leftarrow \lhd(v_{t,j})$
16:     **else**
17:         $v_{t,j+1} \leftarrow \rhd(v_{t,j})$
18:     **end if**
19: **end for**
20: $a_t \leftarrow v_{t,\log(K)}$
21: $\tilde{\pi}_t \leftarrow \prod_{j \in [\log(K)]} \pi_t(v_{t,j})$
22: $\psi_{t,\log(K)} \leftarrow \exp(-\eta \ell_{t,a_t}/\tilde{\pi}_t)$
23: **for** $j = \log(K), (\log(K) - 1), \cdots, 1$ **do**
24:     $\psi_{t,(j-1)} \leftarrow 1 - (1 - \psi_{t,j})\pi_t(v_{t,j})$
25:     **if** $v_{t,j} = \lhd(v_{t,j-1})$ **then**
26:         $\tilde{v}_{t,j} \leftarrow \rhd(v_{t,j-1})$
27:     **else**
28:         $\tilde{v}_{t,j} \leftarrow \lhd(v_{t,j-1})$
29:     **end if**
30:     $\kappa_1(\mathcal{A}(v_{t,j}), u_t) \leftarrow \psi_{t,j}/\psi_{t,j-1}$
31:     $\kappa_1(\mathcal{A}(\tilde{v}_{t,j}), u_t) \leftarrow 1/\psi_{t,j-1}$
32:     EVIDENCE$(\mathcal{A}(v_{t,j}), u_t)$
33:     EVIDENCE$(\mathcal{A}(\tilde{v}_{t,j}), u_t)$
34: **end for**

---

# 5 Conclusion

In this paper we introduced the use of the nearest neighbour methodology for the fully adversarial contextual bandit problem when the contexts are selected from a metric space. We developed an extremely efficient algorithm CBNN. We gave a regret bound for CBNN and, as an example, further analysed it in the case in which the contexts (but not necessarily the losses) are drawn i.i.d. from a distribution on a multi-dimensional hypercube: where CBNN requires no knowledge of parameters.

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

## A  Guide to the Appendices

To give the reader some intuition behind CBNN we present, in Appendix B, our initial idea: an algorithm which obtains the regret bound of CBNN but is exponentially slower. We then give a detailed description of CBNN in appendices C to E. Specifically, in Appendix C we describe our novel algorithmic framework CANPROP. In Appendix D we describe contractions and bayesian networks on them, showing how CANPROP can be implemented with them. Finally, in Appendix E we describe TSTs and how they are used to perform our required operations efficiently. In Appendix F we prove, in order, all of the theorems stated in this paper.

## B  The Initial Idea

Here we describe our initial idea - an algorithm, based on EXP4 [3] and *Belief propagation* [20], which attains the regret bound of CBNN but is exponentially slower - taking a per-trial time of $\tilde{\Theta}(KT)$. Since this section is only to give intuition, and the results are surpassed by CBNN, we do not prove the statements made in this section.

To begin with we assume a-priori knowledge of the set $\mathcal{X} := \{x_t \mid t \in [T]\}$ and function $n$ but the final algorithm will not need this knowledge. Without loss of generality assume that $T$ is an integer power of two.

The algorithm is based on EXP4 [3] which we now describe. On every trial $t$ we maintain a weighting $\hat{w}_t : [K]^{\mathcal{X}} \to [0,1]$. We are free to choose any $\hat{w}_1$ satisfying:

$$\sum_{y \in [K]^{\mathcal{X}}} \hat{w}_1(y) = 1 \, .$$

On each trial $t$ the following happens:

1. $x_t$ is revealed

2. For all $a \in [K]$ set $p_{t,a} \leftarrow \sum_{y \in [K]^{\mathcal{X}}} [\![y(x_t) = a]\!] \hat{w}_t(y)$

3. Set $a_t \leftarrow a$ with probability proportional to $p_{t,a}$

4. Receive $\ell_{t,a_t}$

5. For all $a \in [K]$ set $\hat{\ell}_{t,a} \leftarrow [\![a = a_t]\!] \ell_{t,a_t} \|\boldsymbol{p}_t\|_1 / p_{t,a_t}$

6. For all $y \in [K]^{\mathcal{X}}$ set $\hat{w}_{t+1}(y) \leftarrow \hat{w}_t(y) \exp(-\eta \hat{\ell}_{t,y(x_t)})$

It is a classic result [3] that, for any policy $y : \mathcal{X} \to [K]$, the expected $y$-regret of EXP4 is bounded by:

$$\mathbb{E}[R(y)] \leq \frac{\eta KT}{2} - \frac{\ln(\hat{w}_1(y))}{\eta} \, .$$

If $i \in [\log(T)]$ is such that $2^i \leq \Phi(y) \leq 2^{i+1}$ then:

$$\ln\left(\left(\frac{2^i}{T(K-1)}\right)^{\Phi(y)} \left(1 - \frac{2^i}{T}\right)^{(T-1-\Phi(y))}\right) \in \mathcal{O}\left(\ln\left(\frac{KT}{|\Phi(y)|}\right) \Phi(y)\right)$$

so setting:

$$\eta := \rho \sqrt{\frac{1}{KT}}$$

and:

$$\hat{w}_1(y) := \frac{1}{K \log(T)} \sum_{i \in [\log(T)]} \left(\frac{2^i}{T(K-1)}\right)^{\Phi(y)} \left(1 - \frac{2^i}{T}\right)^{(T-1-\Phi(y))}$$

gives us our desired regret bound.

However, we have two issues - the algorithm takes exponential time and the set $\mathcal{X}$ and function $n$ need to be known a-priori. We will hence discuss how to bring the time complexity down to

379 $\tilde{\Theta}(KT)$ and with no a-priori knowledge. To do this first define, for all $i \in [\log(T)]$, the function
380 $\hat{\tau}_i : [K] \times [K] \to [0,1]$ by:

$$\hat{\tau}_i(a, a') := [\![a \neq a']\!]\frac{2^i}{T(K-1)} + [\![a = a']\!]\left(1 - \frac{2^i}{T}\right)$$

381 Note then that for all $y \in [K]^{\mathcal{X}}$ we have:

$$\hat{w}_1(y) \propto \sum_{i \in [\log(T)]} \left(\prod_{s \in [T]\setminus\{1\}} \hat{\tau}_i(y(x_s), y(n(x_s)))\right)$$

382 so that for all trials $t$:

$$\hat{w}_t(y) \propto \sum_{i \in [\log(T)]} \left(\prod_{s \in [T]\setminus\{1\}} \hat{\tau}_i(y(x_s), y(n(x_s)))\right) \prod_{s \in [T]} \exp(-[\![s < t]\!]\eta\hat{\ell}_{s,y(x_s)})$$

383 and hence for all $a \in [K]$ we have:

$$p_{t,a} \propto \sum_{i \in [\log(T)]} \sum_{y \in [K]^{\mathcal{X}}} [\![y(x_t) = a]\!]\left(\prod_{s \in [T]\setminus\{1\}} \hat{\tau}_i(y(x_s), y(n(x_s)))\right) \prod_{s \in [T]} \exp(-[\![s < t]\!]\eta\hat{\ell}_{s,y(x_s)})$$

384 For all $s \in [t]$ and $a \in [K]$ define:

$$\phi'_t(x_s, a) := [\![s < t]\!]\exp(-\eta\hat{\ell}_{s,a}) + [\![s = t]\!]$$

385 A crucial insight is that:

$$\sum_{y \in [K]^{\mathcal{X}}} [\![y(x_t) = a]\!]\left(\prod_{s \in [T]\setminus\{1\}} \hat{\tau}_i(y(x_s), y(n(x_s)))\right) \prod_{s \in [T]} \exp(-[\![s < t]\!]\eta\hat{\ell}_{s,y(x_s)})$$

386 is equal to:

$$\sum_{y \in [K]^{\mathcal{X}_t}} [\![y(x_t) = a]\!]\phi'_t(x_1, y(x_1)) \prod_{s \in [t]\setminus\{1\}} \hat{\tau}_s(y(x_s), y(n(x_s)))\phi'_t(x_s, y(x_s)) \tag{5}$$

387 which is why the algorithm needs only know $\{x_s \mid s \in [t]\}$ and $\{n(x_s) \mid s \in [t] \setminus \{1\}\}$. On trial $t$ we
388 construct a tree with vertex set $\{x_s \mid s \in [t]\}$ which is rooted at $x_1$ and is such that for all $s \in [t] \setminus \{1\}$
389 we have that $n(x_s)$ is the parent of $x_s$. We note that computing the quantity in Equation (5) for all
390 $a \in [K]$ can be done in a time of $\Theta(Kt)$ by Belief propagation [20] on this tree. Hence we have that
391 $\boldsymbol{p}_t$ can be computed in a time of $\Theta(Kt\log(T))$ without a-priori knowledge of $\mathcal{X}$ and $n$.

## C   Cancellation Propagation

393 We now turn to the description and analysis of our algorithm CBNN, starting with our novel
394 algorithmic framework CANPROP.

### C.1   The General CANPROP Algorithm

396 Let $\mathcal{X} := \{x_t \mid t \in [T]\}$. Note that we do not know $\mathcal{X}$ a-priori but for now let's assume we do. We
397 now introduce a general algorithmic framework CANPROP for handling contextual bandit problems
398 with a per-trial time logarithmic in $K$. Without loss of generality assume that $K$ is an integer power
399 of two. Let $\mathcal{B}$ be a full, balanced, oriented binary tree whose leaves are the set of actions $[K]$. Let
400 $\mathcal{B}' := \mathcal{B} \setminus \{r(\mathcal{B})\}$. CANPROP takes a parameter $\eta \in \mathbb{R}_+$ called the *learning rate*. On each trial $t$
401 CANPROP maintains a function:

$$w_t : \mathcal{B}' \times 2^{\mathcal{X}} \to [0,1]$$

402 The function $w_1$ is free to be defined how one likes, as long as it satisfies the constraint that for all
403 internal vertices $v \in \mathcal{B}^{\dagger}$ we have:

$$\sum_{\mathcal{S} \in 2^{\mathcal{X}}} (w_1(\triangleleft(v), \mathcal{S}) + w_1(\triangleright(v), \mathcal{S})) = 1$$

We now describe how CANPROP acts on trial $t$. For all $v \in \mathcal{B}'$ we define:
$$\theta_t(v) := \sum_{\mathcal{S} \in 2^{\mathcal{X}}} [\![x_t \in \mathcal{S}]\!] w_t(v, \mathcal{S})$$

and for all $v \in \mathcal{B}^\dagger$ we define:
$$\pi_t(\triangleleft(v)) := \frac{\theta_t(\triangleleft(v))}{\theta_t(\triangleleft(v)) + \theta_t(\triangleright(v))} \quad ; \quad \pi_t(\triangleright(v)) := \frac{\theta_t(\triangleright(v))}{\theta_t(\triangleleft(v)) + \theta_t(\triangleright(v))}$$

As we shall see CANPROP needs only compute these values for $\mathcal{O}(\ln(K))$ vertices $v$. CANPROP samples a root-to-leaf path $\{v_{t,j} \mid j \in [\log(K)] \cup \{0\}\}$ as follows. $v_{t,0}$ is defined equal to $r(\mathcal{B})$. For all $j \in [\log(K) - 1] \cup \{0\}$, once $v_{t,j}$ has been sampled we sample $v_{t,(j+1)}$ from the probability distribution defined by:
$$\mathbb{P}[v_{t,(j+1)} = v] := [\![\uparrow(v) = v_{t,j}]\!] \pi_t(v) \quad \forall v \in \mathcal{B}'$$

noting that $v_{t,(j+1)}$ is a child of $v_{t,j}$. We define:
$$\mathcal{P}_t := \{v_{t,j} \mid j \in [\log(K)] \cup \{0\}\}$$

CANPROP then selects:
$$a_t := v_{t,\log(K)}$$

and then receives the loss $\ell_{t,a_t}$. The function $w_t$ is then updated to $w_{t+1}$ as follows. Firstly we define,
$$w_{t+1}(v, \mathcal{S}) := w_t(v, \mathcal{S}) \quad \forall(v, \mathcal{S}) \in \{v' \in \mathcal{B}' \mid \uparrow(v') \notin \mathcal{P}_t\} \times 2^{\mathcal{X}}$$

We then define:
$$\psi_{t,\log(K)} := \exp\left(\frac{-\eta \ell_{t,a_t}}{\prod_{j \in [\log(K)]} \pi_t(v_{t,j})}\right)$$

Once we have defined $\psi_{t,j}$ for some $j \in [\log(K)]$ we then define:
$$\psi_{t,(j-1)} := 1 - (1 - \psi_{t,j}) \pi_t(v_{t,j})$$

$$\beta_t(v) := \frac{[\![v \in \mathcal{P}_t]\!] \psi_{t,j} + [\![v \notin \mathcal{P}_t]\!]}{\psi_{t,(j-1)}} \quad \forall v \in \{\triangleleft(v_{t,(j-1)}), \triangleright(v_{t,(j-1)})\}$$

$$w_{t+1}(v, \mathcal{S}) := ([\![x_t \in \mathcal{S}]\!] \beta_t(v) + [\![x_t \notin \mathcal{S}]\!]) w_t(v, \mathcal{S}) \quad \forall(v, \mathcal{S}) \in \{\triangleleft(v_{t,(j-1)}), \triangleright(v_{t,(j-1)})\} \times 2^{\mathcal{X}}$$

The regret bound of CANPROP is given by the following theorem.

**Theorem C.1.** *Suppose we have a function $y : \mathcal{X} \to [K]$. For all $v \in \mathcal{B}$ define:*
$$\mathcal{Q}(v) := \{x \in \mathcal{X} \mid y(x) \in \Downarrow(v)\}$$

*Then the expected $y$-regret of* CANPROP *is bounded by:*
$$\mathbb{E}[R(y)] \le \frac{\eta K T}{2} - \frac{1}{\eta} \sum_{v \in \mathcal{B}'} [\![\mathcal{Q}(v) \ne \emptyset]\!] \ln(w_1(v, \mathcal{Q}(v)))$$

## C.2 Our Parameter Tuning

We now describe and analyse the initial weighting $w_1$ that we will use. Without loss of generality assume $T$ is an integer power of two. Define $\mathcal{X}' := \mathcal{X} \setminus \{x_1\}$. For all $(x, \mathcal{S}) \in \mathcal{X}' \times 2^{\mathcal{X}}$ define:
$$\sigma(x, \mathcal{S}) := [\![[\![x \in \mathcal{S}]\!] \ne [\![n(x) \in \mathcal{S}]\!]]\!]$$

For all $(v, \mathcal{S}) \in \mathcal{B}' \times 2^{\mathcal{X}}$ we define:
$$w_1(v, \mathcal{S}) := \frac{1}{4 \log(T)} \sum_{i \in [\log(T)]} \prod_{x \in \mathcal{X}'} \left(\sigma(x, \mathcal{S}) \frac{2^i}{T} + (1 - \sigma(x, \mathcal{S}))\left(1 - \frac{2^i}{T}\right)\right)$$

Given our parameter $\rho$ we choose our learning rate as:
$$\eta := \rho \sqrt{\frac{\ln(K) \ln(T)}{KT}}$$

Given this initial weighting and learning rate, Theorem C.1 implies the following regret bound.

**Theorem C.2.** *Given $w_1$ and $\eta$ are defined as above, then for any policy $y : \mathcal{C} \to [K]$ the expected $y$-regret of* CANPROP *is bounded by:*
$$\mathbb{E}[R(y)] \in \mathcal{O}\left(\left(\rho + \frac{\Phi(y)}{\rho}\right) \sqrt{\ln(K) \ln(T) KT}\right)$$

## D Binarisation and Implementation with Contractions

### D.1 A Sequence of Binary Trees

For any trial $t$ we have a natural tree-structure on the set $\{x_{t'} \mid t' \in [t]\}$ formed by making $n(x_{t'})$ the parent of $x_{t'}$ for all $t' \in [t] \setminus \{1\}$. However, in order to utilise the methodology of [19] we need to work with binary trees. Hence, we now inductively define a sequence of binary trees $\{\mathcal{Z}_t \mid t \in [T] \setminus \{1\}\}$ where the vertices of $\mathcal{Z}_t$ are a subset of those of $\mathcal{Z}_{t+1}$. We also define a function $\gamma : \mathcal{Z}_T \to \mathcal{X}$. This function $\gamma$ has the property that for any $t \in [T]$ and for any distinct leaves $u, u' \in \mathcal{Z}_t^\star$ we have that $\gamma(u) \neq \gamma(u')$, and that:
$$\{\gamma(u) \mid u \in \mathcal{Z}_t^\star\} = \{x_{t'} \mid t' \in [t]\}$$
We define $\mathcal{Z}_2$ to contain three vertices $\{r(\mathcal{Z}_2), \lhd(r(\mathcal{Z}_2)), \rhd(r(\mathcal{Z}_2))\}$ where:
$$\gamma(r(\mathcal{Z}_2)) := \gamma(\lhd(r(\mathcal{Z}_2))) := x_1 \quad \text{and} \quad \gamma(\rhd(r(\mathcal{Z}_2))) := x_2$$
Now consider a trial $t \in [T]$. We have that $\mathcal{Z}_{t+1}$ is constructed from $\mathcal{Z}_t$ via the following algorithm $\text{GROW}_{t+1}$:

1. Let $u$ be the unique leaf in $\mathcal{Z}_t^\star$ in which $\gamma(u) = n(x_{t+1})$ and let $u^* := \uparrow(u)$.
2. Create two new vertices $u'$ and $u''$.
3. Set $\gamma(u') \leftarrow n(x_{t+1})$ and $\gamma(u'') \leftarrow x_{t+1}$.
4. If $u = \lhd(u^*)$ then set $\lhd(u^*) \leftarrow u'$. Else set $\rhd(u^*) \leftarrow u'$.
5. Set $\lhd(u') \leftarrow u''$ and $\rhd(u') \leftarrow u$

We also define a function $d : \mathcal{Z}_T \to \mathbb{N} \cup \{0\}$ as follows. Define $d'(x_1) := 0$ and for all $t \in [T] \setminus \{1\}$ inductively define $d'(x_t) := d'(n(x_t)) + 1$. Finally define $d(u) := d'(\gamma(u))$ for all $u \in \mathcal{Z}_T$. Since for all $t \in [T]$ we have that the vertices of $\mathcal{Z}_t$ are a subset of those of $\mathcal{Z}_T$ we have that $d$ also defines a function over $\mathcal{Z}_t$ for all $t \in [T]$.

For all $t \in [T]$ we define $u_t$ to be the unique leaf of $\mathcal{Z}_t$ for which $\gamma(u_t) = x_t$.

### D.2 Contractions

Our efficient implementation of CANPROP will have a data-structure at every vertex $v \in \mathcal{B}'$. However, to achieve polylogarithmic time per trial we can only update a polylogarithmic number of these data-structures per trial. This necessitates the use of *contractions* of our trees $\{\mathcal{Z}_t \mid t \in [T] \setminus \{1\}\}$ which are defined as follows. A *contraction* of a full binary tree $\mathcal{Q}$ is another full binary tree $\mathcal{J}$ which satisfies the following:

- The vertices of $\mathcal{J}$ are a subset of those of $\mathcal{Q}$.
- $r(\mathcal{J}) = r(\mathcal{Q})$
- Given an internal vertex $u \in \mathcal{J}^\dagger$ we have $\lhd_{\mathcal{J}}(u) \in \Downarrow_{\mathcal{Q}}(\lhd_{\mathcal{Q}}(u))$ and $\rhd_{\mathcal{J}}(u) \in \Downarrow_{\mathcal{Q}}(\rhd_{\mathcal{Q}}(u))$
- Any leaf of $\mathcal{J}$ is a leaf of $\mathcal{Q}$.

Note that any contraction of $\mathcal{Z}_t$ is also a contraction of $\mathcal{Z}_{t+1}$ and hence, by induction, a contraction of $\mathcal{Z}_{t'}$ for all $t' \geq t$. Given a trial $t$ and a contraction $\mathcal{J}$ of $\mathcal{Z}_{t-1}$ we now define the operation $\text{INSERT}_t(\mathcal{J})$ which acts on $\mathcal{J}$ by the following algorithm:

1. Let $\hat{u}$ be the unique vertex in $\mathcal{J} \setminus r(\mathcal{J})$ such that $u_t$ lies in the maximal spanning tree of $\mathcal{Z}_t$ with $\uparrow_{\mathcal{J}}(\hat{u})$ and $\hat{u}$ as leaves.
2. Let $u^* := \Gamma_{\mathcal{Z}_t}(u_t, \hat{u})$.
3. Add the vertices $u^*$ and $u_t$ to the tree $\mathcal{J}$.
4. Let $u' := \uparrow_{\mathcal{J}}(\hat{u})$.
5. If $\hat{u} = \lhd_{\mathcal{J}}(u')$ then set $\lhd_{\mathcal{J}}(u') \leftarrow u^*$. Else set $\rhd_{\mathcal{J}}(u') \leftarrow u^*$.
6. If $\hat{u} \in \Downarrow_{\mathcal{Z}_t}(\lhd_{\mathcal{Z}_t}(u^*))$ then set $\lhd_{\mathcal{J}}(u^*) \leftarrow \hat{u}$ and $\rhd_{\mathcal{J}}(u^*) \leftarrow u_t$. Else set $\rhd_{\mathcal{J}}(u^*) \leftarrow \hat{u}$ and $\lhd_{\mathcal{J}}(u^*) \leftarrow u_t$

Later in this paper we will show how this operation can be done in polylogarithmic time. Note that after the operation we have that $\mathcal{J}$ is a contraction of $\mathcal{Z}_t$ and $u_t$ has been added to it's leaves. From now on when we use the term *contraction* we mean any contraction of $\mathcal{Z}_T$.

### D.3 Contraction-Based Bayesian Networks

Here we shall define a bayesian network over any contraction $\mathcal{J}$ and show how it can be utilised to compute certain quantities required by CANPROP. This bayesian network takes a parameter $\epsilon \in [0, 1]$. First define the quantity $\phi_0(\epsilon) := 0$ and for all $j \in \mathbb{N} \cup \{0\}$ inductively define:

$$\phi_{j+1}(\epsilon) := (1 - \epsilon)\phi_j(\epsilon) + \epsilon(1 - \phi_j(\epsilon))$$

The algorithm must compute these quantities for various values of $\epsilon$. However, for all $t \in [T]$ we have that $\phi_t(\epsilon)$ doesn't have to be computed until trial $t$ so computing these quantities is constant time per trial (for each value of $\epsilon$). Given a contraction $\mathcal{J}$, a value $\epsilon \in [0, 1]$, a vertex $u \in \mathcal{J} \setminus r(\mathcal{J})$ and indices $i, i' \in \{0, 1\}$ define:

$$\tilde{\tau}_{i,i'}(\mathcal{J}, u, \epsilon) := [\![i \neq i']\!]\phi_{(d(u)-d(\uparrow_{\mathcal{J}}(u)))}(\epsilon) + [\![i = i']\!](1 - \phi_{(d(u)-d(\uparrow_{\mathcal{J}}(u)))}(\epsilon))$$

which defines the transition matrix from $\uparrow_{\mathcal{J}}(u)$ to $u$ in a bayesian network. We shall now show how belief propagation over such bayesian networks can be used to compute the quantities we need in CANPROP. Suppose we have a contraction $\mathcal{J}$, a value $\epsilon \in [0, 1]$ and a function $\lambda : \mathcal{J}^\star \to \mathbb{R}_+$. This function $\lambda$ induces a function $\lambda' : \mathcal{X} \to \mathbb{R}_+$ defined as follows. Given $x \in \mathcal{X}$, if there exists a leaf $u \in \mathcal{J}^\star$ with $\gamma(u) = x$ then $\lambda'(x) = \lambda(u)$. Otherwise $\lambda'(x) = 1$. We then define a weighting $\tilde{w}(\lambda, \epsilon, \cdot) : 2^{\mathcal{X}} \to \mathbb{R}_+$ such that for all $\mathcal{S} \in 2^{\mathcal{X}}$ we have:

$$\tilde{w}(\lambda, \epsilon, \mathcal{S}) := \left(\prod_{x \in \mathcal{S}} \lambda'(x)\right)\left(\prod_{x \in \mathcal{X}'} (\sigma(x, \mathcal{S})\epsilon + (1 - \sigma(x, \mathcal{S}))(1 - \epsilon))\right)$$

For the CANPROP algorithm we will need to compute

$$\sum_{\mathcal{S} \in 2^{\mathcal{X}}} [\![\gamma(\hat{u}) \in \mathcal{S}]\!]\tilde{w}(\lambda, \epsilon, \mathcal{S}) \tag{6}$$

for some leaf $\hat{u} \in \mathcal{J}^\star$. We shall now show how we can compute this quantity via belief propagation on the bayesian network. In particular we shall construct a quantity $\tilde{\Lambda}(\mathcal{J}, \lambda, \epsilon, u)$ equal to the quantity in Equation (6). To do this first define the function $\lambda^* : \mathcal{J} \to \mathbb{R}_+$ so that for all $u \in \mathcal{J}^\star$ we have $\lambda^*(u) = \lambda(u)$ and for all $u \in \mathcal{J}^\dagger$ we have $\lambda^*(u) = 1$. For all vertices $u \in \mathcal{J}$ and all indices $i \in \{0, 1\}$ define:

$$\tilde{\kappa}_i(\lambda, u) := [\![i = 0]\!] + [\![i = 1]\!]\lambda^*(u)$$

For all $\hat{u} \in \mathcal{J}$ define:

$$\mathcal{F}(\mathcal{J}, \hat{u}) := \{f \in \{0, 1\}^{\mathcal{J}} \mid f(\hat{u}) = 1\}$$

and then define:

$$\tilde{\Lambda}(\mathcal{J}, \lambda, \epsilon, \hat{u}) := \sum_{f \in \mathcal{F}(\mathcal{J}, \hat{u})} \prod_{u \in \mathcal{J} \setminus r(\mathcal{J})} \tilde{\tau}_{f(\uparrow_{\mathcal{J}}(u)), f(u)}(\mathcal{J}, u, \epsilon)\tilde{\kappa}_{f(u)}(\lambda, u)$$

The equality of this quantity and that given in Equation (6) is given by the following theorem.

**Theorem D.1.** *Given a contraction $\mathcal{J}$, a function $\lambda : \mathcal{J}^\star \to \mathbb{R}_+$, some $\epsilon \in [0, 1]$ and some leaf $\hat{u} \in \mathcal{J}^\star$ we have:*

$$\tilde{\Lambda}(\mathcal{J}, \lambda, \epsilon, \hat{u}) = \sum_{\mathcal{S} \in 2^{\mathcal{X}}} [\![\gamma(\hat{u}) \in \mathcal{S}]\!]\tilde{w}(\lambda, \epsilon, \mathcal{S})$$

Note that $\tilde{\Lambda}(\mathcal{J}, \lambda, \epsilon, \hat{u})$ is of the exact form to be solved via belief propagation over $\mathcal{J}$. However, belief propagation is still too slow (taking $\Theta(|\mathcal{J}|)$ time) - we will remedy this later.

### D.4 Cancelation Propogation with Contractions

We now describe how to implement CANPROP with contractions. For each $v \in \mathcal{B}'$ we maintain a contraction $\mathcal{A}(v)$ and a function $\zeta(v, \cdot) : \mathcal{A}(v)^\star \to \mathbb{R}_+$. We initialise with $\mathcal{A}(v)$ identical to $\mathcal{Z}_2$ and $\zeta(v, u) = 1$ for both leaves $u \in \mathcal{Z}_2^\star$. Via induction over $t$ we will have that at the start of each trial $t$ we have, for all sets $\mathcal{S} \in 2^{\mathcal{X}}$, that:

$$w_t(v, \mathcal{S}) = \frac{1}{4\log(T)} \sum_{i \in [\log(T)]} \tilde{w}(\zeta(v, \cdot), 2^i/T, \mathcal{S}) \tag{7}$$

On trial $t$ we do as follows. First we update $\mathcal{Z}_{t-1}$ to $\mathcal{Z}_t$ using the algorithm $\text{GROW}_t$. We will perform the necessary modifications to our contractions as we sample the path $\mathcal{P}_t$. In particular we first set $v_{t,0} \leftarrow r(\mathcal{B})$ and then for each $j \in [\log(K) - 1] \cup \{0\}$ in turn we do as follows. For each $v \in \{\triangleleft(v_{t,j}), \triangleright(v_{t,j})\}$ run $\text{INSERT}_t(\mathcal{A}(v))$ and set $\zeta(v, u_t) \leftarrow 1$. Since $\zeta(v, u_t) = 1$ Equation (7) still holds and hence, by Theorem D.1, we have:

$$\theta_t(v) = \frac{1}{4\log(T)} \sum_{i \in [\log(T)]} \tilde{\Lambda}(\mathcal{A}(v), \zeta(v, \cdot), 2^i/T, u_t)$$

where $\zeta(v, \cdot)$ is the function that maps each $u \in \mathcal{A}(v)$ to $\zeta(v, u)$. After $\theta_t(v)$ has been computed for both $v \in \{\triangleleft(v_{t,j}), \triangleright(v_{t,j})\}$ we can now sample $v_{t,j+1}$.

Once we have selected the action $a_t$ we then update the functions $\{\zeta(v, \cdot) \mid \uparrow_\mathcal{B}(v) \in \mathcal{P}_t\}$ by setting $\zeta(v, u_t) \leftarrow \beta_t(v)$ for all $v \in \mathcal{B}'$ with $\uparrow_\mathcal{B}(v) \in \mathcal{P}_t$. It is clear now that Equation (7) holds inductively.

### D.5 Notational Relationship to the Main Body

We now point out how the notation in this section relates to that of the main body. In particular we have, for all $v \in \mathcal{B}'$, all $u \in \mathcal{A}(v)$, all $j \in [\log(T)]$ and all $i, i' \in \{0, 1\}$, that:

- $\tau_{i,i'}(\mathcal{A}(v), u, j) = \tilde{\tau}_{i,i'}(\mathcal{A}(v), u, 2^j/T)$

- $\kappa_i(\mathcal{A}(v), u) = \tilde{\kappa}_i(\zeta(v, \cdot), u)$

- $\Lambda(\mathcal{A}(v), u, j) = \tilde{\Lambda}(\mathcal{A}(v), \zeta(v, \cdot), 2^j/T, u)$

## E Utilising Ternary Search Trees

There are now only two things left to do in order to achieve polylogarithmic time per trial - to make an efficient online implementation of the $\text{INSERT}_t(\cdot)$ operation and an efficient online algorithm to perform belief propagation over our contractions. In order to do this we will utilise the methodology of [19] which we now describe. However, we do not give the full details of the rebalancing technique and refer the reader to [19] for these details.

### E.1 Ternary Search Trees

In this section we will consider a full binary tree $\mathcal{J}$. A *(full) ternary tree* $\mathcal{D}$ is a rooted tree in which each internal vertex $s \in \mathcal{D}^\dagger$ has three children denoted by $\triangleleft(s), \triangledown(s), \triangleright(s)$ and called the left, centre, and right children respectively. We now define what it means for a ternary tree $\mathcal{D}$ to be a ternary search tree (TST) of $\mathcal{J}$. Firstly, the vertex set of $\mathcal{D}$ is partitioned into two sets $\mathcal{D}^\circ$ and $\mathcal{D}^\bullet$. Every vertex $s \in \mathcal{D}$ is associated with a vertex $\mu(s) \in \mathcal{J}$ and every $s \in \mathcal{D}^\bullet$ is also associated with a vertex $\mu'(s) \in \Downarrow_\mathcal{J}(\mu(s))^\dagger$. The root $r(\mathcal{D})$ of $\mathcal{D}$ is contained in $\mathcal{D}^\circ$ and $\mu(r(\mathcal{D})) := r(\mathcal{J})$. Each internal vertex $s \in \mathcal{D}^\dagger$ is associated with a vertex $\xi(s) \in \mathcal{J}$. If $s \in \mathcal{D}^\circ$ then $\xi(s) \in \Downarrow(\mu(s))^\dagger$ and if $s \in \mathcal{D}^\bullet$ then $\xi(s)$ lies on the path (in $\mathcal{J}$) from $\mu(s)$ to $\uparrow(\mu'(s))$. For all $s \in \mathcal{D}^\dagger$ we have:

- $\triangledown(s) \in \mathcal{D}^\bullet$, $\mu(\triangledown(s)) := \mu(s)$ and $\mu'(\triangledown(s)) := \xi(s)$.
- $\triangleleft(s)$ satisfies:
    - If $s \in \mathcal{D}^\circ$ then $\triangleleft(s) \in \mathcal{D}^\circ$ and $\mu(\triangleleft(s)) := \triangleleft(\xi(s))$.
    - If $s \in \mathcal{D}^\bullet$ and $\mu'(s) \in \Downarrow(\triangleright(\xi(s)))$ then $\triangleleft(s) \in \mathcal{D}^\circ$ and $\mu(\triangleleft(s)) := \triangleleft(\xi(s))$
    - Else $\triangleleft(s) \in \mathcal{D}^\bullet$, $\mu(\triangleleft(s)) := \triangleleft(\xi(s))$ and $\mu'(\triangleleft(s)) := \mu'(s)$
- $\triangleright(s)$ satisfies:
    - If $s \in \mathcal{D}^\circ$ then $\triangleright(s) \in \mathcal{D}^\circ$ and $\mu(\triangleright(s)) := \triangleright(\xi(s))$.
    - If $s \in \mathcal{D}^\bullet$ and $\mu'(s) \in \Downarrow(\triangleleft(\xi(s)))$ then $\triangleright(s) \in \mathcal{D}^\circ$ and $\mu(\triangleright(s)) := \triangleright(\xi(s))$
    - Else $\triangleright(s) \in \mathcal{D}^\bullet$, $\mu(\triangleright(s)) := \triangleright(\xi(s))$ and $\mu'(\triangleright(

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

## F Proofs

### F.1 Theorem 3.1

For brevity we write $\alpha$ instead of $\alpha(y, \mu)$. Choose some $\delta > 0$. Let $\mathcal{E} := \mathcal{C} \setminus \mathcal{M}(y, \delta)$. Let $\mathcal{X}$ be a set of $m$ contexts drawn i.i.d. at random from $\mu$. Now consider some $x$ drawn from $\mu$ and let $\hat{x}$ be a $c$-nearest neighbour of $x$ in $\mathcal{X}$.

Suppose that $x \in \mathcal{E}$. Let $\mathcal{A}$ be the ball of radius $\delta/c$ centred at $x$. We have that:

$$\mu(\mathcal{A}) \geq \epsilon \lambda \left(\frac{\delta}{c}\right)^d$$

where $\lambda$ is a constant dependent on $d$. This means that for any $x'$ drawn from $\mu$ we have that:

$$\mathbb{P}[x' \notin \mathcal{A}] \leq 1 - \lambda \epsilon \left(\frac{\delta}{c}\right)^d \leq \exp\left(-\lambda \epsilon \left(\frac{\delta}{c}\right)^d\right)$$

Suppose that:

$$m \geq \frac{-\ln(\alpha\delta)}{\lambda \epsilon} \left(\frac{c}{\delta}\right)^d$$

Note that if there exists $x'' \in \mathcal{X}$ with $x'' \in \mathcal{A}$ then $\Delta(x, \hat{x}) \leq \delta$ so that $y(x) = y(\hat{x})$. The above equations then give us:

$$\mathbb{P}[y(x) \neq y(\hat{x}) \mid x \in \mathcal{E}] \leq \exp\left(-m\lambda\epsilon \left(\frac{\delta}{c}\right)^d\right) \leq \alpha\delta$$

We then have that:
$$\mathbb{P}[y(x) \neq y(\hat{x})] \leq \alpha\delta + \mu(\mathcal{M}(\delta)) \in \mathcal{O}(\alpha\delta)$$

Since:
$$\delta \in \tilde{\mathcal{O}}(c(\epsilon m)^{-1/d})$$

we now have:
$$\mathbb{P}[y(x) \neq y(\hat{x})] \in \tilde{\mathcal{O}}\left(c\alpha(\epsilon m)^{-1/d}\right)$$

### F.2 Theorem 3.2

This theorem is proved in appendices C to E and the theorems therein.

### F.3 Theorem 3.3

Choose a set $\mathcal{S} \subseteq \mathcal{C}$ in which for all $t \in [T]$ there exists $x \in \mathcal{S}$ with $\Delta(x, x_t) < \gamma(x, y)/3c$. For all trials $t$ let $\mathcal{S}_t$ be the set of all contexts $x \in \mathcal{S}$ in which there exists $s \in [t]$ with $\Delta(x, x_s) < \gamma(x, y)/3c$.

Now consider a trial $t$ in which $y(x_t) \neq y(n(x_t))$ and choose $x \in \mathcal{S}$ with $\Delta(x, x_t) < \gamma(x, y)/3c$.

Assume, for contradiction, that $x \in \mathcal{S}_{t-1}$. Then there exists $s \in [t-1]$ with $\Delta(x, x_s) < \gamma(x, y)/3c$ so that by the triangle inequality we have:

$$\Delta(x_t, x_s) \leq \Delta(x, x_s) + \Delta(x, x_t) < 2\gamma(x, y)/3c$$

which implies that $\Delta(x_t, n(x_t)) < 2\gamma(x, y)/3$. By the triangle inequality we then have that:

$$\Delta(x, n(x_t)) \leq \Delta(x_t, n(x_t)) + \Delta(x, x_t) < 2\gamma(x, y)/3 + \gamma(x, y)/3c \leq 3\gamma(x, y)/3 = \gamma(x, y)$$

Since $\Delta(x, x_t) < \gamma(x, y)$ we have $y(x) = y(x_t)$ and hence that $y(x) \neq y(n(x_t))$. But this contradicts the fact that $\Delta(x, n(x_t)) < \gamma(x, y)$.

We have hence shown that $x \notin \mathcal{S}_{t-1}$. Since $x \in \mathcal{S}_t$ we then have that $|\mathcal{S}_t| \geq |\mathcal{S}_{t-1}|$. This implies that:

$$\Phi(y) = \sum_{t \in [T]} [\![y(x_t) \neq y(n(x_t))]\!] \leq |\mathcal{S}_T| \leq |\mathcal{S}|$$

as required.

 ## F.4 Theorem 3.4

By linearity of expectation we have:

$$\mathbb{E}[\Phi(y)] \leq 1 + \sum_{t \in [T]} g_t(\mu, y, c)$$

and from Theorem 3.1 we have:

$$g_t(\mu, y, c) \in \mathcal{O}\left(c\alpha(y, \mu)(\epsilon t)^{-1/d}\right)$$

so that:

$$\mathbb{E}[\Phi(y)] \in \mathcal{O}\left(c\alpha(y, \mu)\epsilon^{-1/d}T^{(d-1)/d}\ln(T)\right)$$

By setting:

$$\rho := T^{(d-1)/(2d)}c^{1/2}$$

we then have:

$$\rho + \frac{\mathbb{E}[\Phi(y)]}{\rho} \in \mathcal{O}\left(c^{1/2}(1 + \alpha(y, \mu)\epsilon^{-1/d})T^{(d-1)/(2d)}\ln(T)\right)$$

so that by Theorem 3.2 we have:

$$\mathbb{E}[R(y)] \in \tilde{\mathcal{O}}\left(c^{1/2}(1 + \alpha(y, \mu)\epsilon^{-1/d})K^{1/2}T^{(2d-1)/(2d)}\right)$$

## F.5 Theorem C.1

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

(\triangleleft(v)) + \theta_t(\triangleright(v)))(\pi_t(\triangleleft(v)) \beta_t(\triangleleft(v)) + \pi_t(\triangleright(v)) \beta_t(\triangleright(v)))$$

Without loss of generality assume that $\triangleleft(v) \in \mathcal{P}_t$. Then the above equation implies that:

$$V_t(v) = (\theta_t(\triangleleft(v)) + \theta_t(\triangleright(v)))\frac{\pi_t(\triangleleft(v))\psi_{t,j+1} + \pi_t(\triangleright(v))}{\psi_{t,j}}$$

so by definition of $\psi_{t,j}$ we have:

$$V_t(v) = (\theta_t(\triangleleft(v)) + \theta_t(\triangleright(v))) = \sum_{\mathcal{S} \in 2^{\mathcal{X}}} [\![x_t \in \mathcal{S}]\!]w_t(\triangleleft(v), \mathcal{S}) + \sum_{\mathcal{S} \in 2^{\mathcal{X}}} [\![x_t \in \mathcal{S}]\!]w_t(\triangleright(v), \mathcal{S})$$

Note that this equation trivially holds for all $v \in \mathcal{B}^\dagger \setminus \mathcal{P}_t$ and hence holds for all $v \in \mathcal{B}^\dagger$. Since for all such $v$ and all $\mathcal{S}$ with $x_t \notin \mathcal{S}$ we have $w_{t+1}(\triangleleft(v), \mathcal{S}) = w_t(\triangleleft(v), \mathcal{S})$ and $w_{t+1}(\triangleright(v), \mathcal{S}) = w_t(\triangleright(v), \mathcal{S})$ we then have:

$$\sum_{\mathcal{S} \in 2^{\mathcal{X}}} w_{t+1}(\triangleleft(v), \mathcal{S}) + \sum_{\mathcal{S} \in 2^{\mathcal{X}}} w_{t+1}(\triangleright(v), \mathcal{S}) = \sum_{\mathcal{S} \in 2^{\mathcal{X}}} w_t(\triangleleft(v), \mathcal{S}) + \sum_{\mathcal{S} \in 2^{\mathcal{X}}} w_t(\triangleright(v), \mathcal{S})$$

so, by induction on $t$ we have, for all $t \in [T + 1]$, that:

$$\sum_{\mathcal{S} \in 2^{\mathcal{X}}} w_t(\triangleleft(v), \mathcal{S}) + \sum_{\mathcal{S} \in 2^{\mathcal{X}}} w_t(\triangleright(v), \mathcal{S}) = 1$$

Hence, for all $v \in \mathcal{B} \setminus r(\mathcal{B})$ and $\mathcal{S} \in 2^{\mathcal{X}}$, we have $w_t(v, \mathcal{S}) \in [0, 1]$. We have now shown that $\Delta_{T+1} \geq 0$ so that Equation 19 gives us:

$$\mathbb{E}[R(y)] \leq \frac{1}{\eta}\mathbb{E}[\Delta_1] + \frac{\eta K T}{2}$$

which, by definition of $\Delta_1$, gives us the desired result.

## F.6 Theorem C.2

The fact that the weighting $w_t$ is valid is given by the following lemma:

**Lemma F.1.** *For all $v \in \mathcal{B}^\dagger$ we have:*

$$\sum_{\mathcal{S} \in 2^{\mathcal{X}}} (w_1(\triangleleft(v), \mathcal{S}) + w_t(\triangleright(v), \mathcal{S})) = 1$$

*Proof.* We will show that for all $v \in \mathcal{B}'$ we have:

$$\sum_{\mathcal{S} \in 2^{\mathcal{X}}} w_1(v, \mathcal{S}) = \frac{1}{2}$$

which directly implies the result. So take some arbitrary $v \in \mathcal{B}'$. Define, for all $t \in [T]$, the sets:

$$\mathcal{X}'_t := \{x_s \mid s \in [t]\} \setminus \{x_1\} \quad \text{and} \quad \mathcal{F}_t := \{0, 1\}^{\mathcal{X}'_t \cup \{x_1\}}$$

and for all $x \in \mathcal{X}'_t$, $f \in \mathcal{F}_t$ and $i \in [\log(T)]$, define the quantity:

$$\beta_i(x, f) := [\![f(x) \neq f(n(x))]\!]2^i/T + [\![f(x) = f(n(x))]\!](1 - 2^i/T)$$

which is defined since $n(x) \in \mathcal{X}'_t \cup \{x_1\}$. Now fix some $i \in [\log(T)]$. For all $t \in [T - 1]$ we have:

$$\sum_{f \in \mathcal{F}_{t+1}} \prod_{x \in \mathcal{X}'_{t+1}} \beta_i(x, f) = \sum_{f \in \mathcal{F}_t} \left(\prod_{x \in \mathcal{X}'_t} \beta_i(x, f)\right) \sum_{f(x_{t+1}) \in \{0,1\}} \beta_i(x_{t+1}, f)$$

Given any $f \in \mathcal{F}_t$ we have:

$$\sum_{f(x_{t+1}) \in \{0,1\}} \beta_i(x_{t+1}, f) = \left(1 - \frac{2^i}{T}\right) + \frac{2^i}{T} = 1$$

and hence:

$$\sum_{f \in \mathcal{F}_{t+1}} \prod_{x \in \mathcal{X}'_{t+1}} \beta_i(x, f) = \sum_{f \in \mathcal{F}_t} \prod_{x \in \mathcal{X}'_t} \beta_i(x, f)$$

729 Since $\mathcal{X}'_T = \mathcal{X}'$ this implies, by induction, that:

$$\sum_{f \in \mathcal{F}_T} \prod_{x \in \mathcal{X}'} \beta_i(x, f) = \sum_{f \in \mathcal{F}_1} \prod_{x \in \mathcal{X}'_1} \beta_i(x, f) = \sum_{f \in \mathcal{F}_1} \prod_{x \in \emptyset} \beta_i(x, f) = \sum_{f \in \mathcal{F}_1} 1 = |\mathcal{F}_1| = 2 \qquad (20)$$

730 Note that we have a bijection $\mathcal{G} : \mathcal{F}_T \rightarrow 2^{\mathcal{X}}$ defined by:

$$\mathcal{G}(f) := \{x \in \mathcal{X} \mid f(x) = 1\} \quad \forall f \in \mathcal{F}_T$$

731 and that for all $(i, f, x) \in [\log(T)] \times \mathcal{F}_T \times \mathcal{X}'$ we have:

$$\beta_i(x, f) = \sigma(x, \mathcal{G}(f))2^i/T + (1 - \sigma(x, \mathcal{G}(f)))(1 - 2^i/T)$$

732 Hence, Equation (20) shows us that for all $i \in [\log(T)]$ we have:

$$\sum_{\mathcal{S} \in 2^{\mathcal{X}}} \prod_{x \in \mathcal{X}'} \left( \sigma(x, \mathcal{S})\frac{2^i}{T} + (1 - \sigma(x, \mathcal{S})) \left(1 - \frac{2^i}{T}\right) \right) = 2$$

733 This implies that:

$$\sum_{\mathcal{S} \in 2^{\mathcal{X}}} w_1(v, \mathcal{S}) = \frac{1}{2}$$

734 which implies the result. $\qquad\qquad\square$

735 Now that we have shown that the weighting $w_1$ is valid we can utilise Theorem C.1 to prove our
736 regret bound. For any set $\mathcal{S} \in 2^{\mathcal{X}}$ define:

$$\phi(\mathcal{S}) := \sum_{x \in \mathcal{X}'} \sigma(x, \mathcal{S})$$

737 For any $i \in [\log(T)]$ define the function $f_i : [T - 1] \rightarrow \mathbb{R}$ by

$$f_i(c) := \left(1 - \frac{2^i}{T}\right)^{T-1-c} \left(\frac{2^i}{T}\right)^c$$

738 for all $c \in [T - 1]$. Choose any set $\mathcal{S} \in 2^{\mathcal{X}}$ and define:

$$j := \min\{\lceil \log(\phi(\mathcal{S}) + 1) \rceil, \log(T) - 1\}$$

739 If $\phi(\mathcal{S}) \geq T/2$ then $j = \log(T) - 1$ so $2^j/T = 1/2$ and hence:

$$-\ln(f_j(\phi(\mathcal{S}))) = (T - 1)\ln(2) \leq 2\phi(\mathcal{S})\ln(2) \in \mathcal{O}(\phi(\mathcal{S})) \qquad (21)$$

740 Now consider the case in which $\phi(\mathcal{S}) < T/2$. Let $h := 2^j/T$. In this case $2^j/T \leq 1/2$ and hence $f_j$
741 is monotonic decreasing so since $2^j \geq \phi(\mathcal{S})$ we have:

$$\ln(f_j(\phi(\mathcal{S})) \geq \ln(f_j(2^j)) = \ln(f_j(Th)) \geq T((1 - h)\ln(1 - h) + h\ln(h)) \geq -Th\ln(e/h)$$

742 so since $\phi(\mathcal{S}) + 1 \geq 2^j/2 = Th/2$ and $h \geq 1/T$ we have:

$$-\ln(f_j(\phi(\mathcal{S})) \leq 2(\phi(\mathcal{S}) + 1)\ln(eT) \in \mathcal{O}((\phi(\mathcal{S}) + 1)\ln(T)) \qquad (22)$$

743 Equations (21) and (22) show us that for all possible values of $\phi(\mathcal{S})$ we have:

$$-\ln(f_j(\phi(\mathcal{S})) \in \mathcal{O}(\ln(T)(\phi(\mathcal{S}) + 1))$$

744 Noting that for all $v \in \mathcal{B}'$ we have $w_1(v, \mathcal{S}) \geq (1/4 \log(T))f_j(\phi(\mathcal{S}))$ we have now shown that:

$$-\ln(w_1(v, \mathcal{S})) \in \mathcal{O}(\ln(T)(\phi(\mathcal{S}) + 1)) \qquad (23)$$

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

809    Since:

$$\tilde{\tau}_{i,0}(\mathcal{Z}_T, u', \epsilon) + \tilde{\tau}_{i,1}(\mathcal{Z}_T, u', \epsilon) = 1 \tag{36}$$

810    we immediately have the result for the case that the height of $\Downarrow_{\mathcal{Z}_T}(u')$ is zero. Now suppose that the
811    result holds whenever the height of $\Downarrow_{\mathcal{Z}_T}(u')$ is equal to $j$ (for some $j \in \mathbb{N}$). We will now show that it
812    holds whenever the height of $\Downarrow_{\mathcal{Z}_T}(u')$ is equal to $j+1$ which will prove that the result holds always.
813    By the inductive hypothesis we have, for all $i' \in \{0,1\}$

$$\sum_{f \in \{0,1\}^{\mathcal{H}(\lhd(u'))}} [\![f(u') = i']\!] \hat{\zeta}(\lhd(u'), f) = 1$$

814    and

$$\sum_{f \in \{0,1\}^{\mathcal{H}(\rhd(u'))}} [\![f(u') = i']\!] \hat{\zeta}(\rhd(u'), f) = 1$$

815    so:

$$\sum_{f \in \{0,1\}^{\mathcal{H}(u')}} [\![f(\uparrow_{\mathcal{Z}_T}(u')) = i]\!][\![f(u') = i']\!] \hat{\zeta}(\lhd(u'), f) \hat{\zeta}(\rhd(u'), f) = 1$$

816    and hence:

$$\sum_{f \in \{0,1\}^{\mathcal{H}(u')}} [\![f(\uparrow_{\mathcal{Z}_T}(u')) = i]\!][\![f(u') = i']\!] \hat{\zeta}(u', f) = \tilde{\tau}_{i,i'}(\mathcal{Z}_T, u, \epsilon)$$

817    Summing over $i' \in \{0,1\}$ and noting Equation (36) then shows us the result holds for this case and
818    hence, by induction, holds always. $\qquad\square$

819 Given $u', u'' \in \mathscr{Z}_T$ with $u'' \in \Downarrow_{\mathscr{Z}_T}(u')$ we define $\hat{\mathcal{H}}(u', u'')$ to be the maximal subtree of $\mathscr{Z}_T$ which

820 has $u'$ and $u''$ as leaves. Given, in addition, $f : \hat{\mathcal{H}}(u', u'') \to \{0, 1\}$ we define:

$$\tilde{\zeta}(u', u'', f) := \prod_{u \in \hat{\mathcal{H}}(u', u'') \setminus \{u'\}} \tilde{\tau}_{f(\uparrow_{\mathscr{Z}_T}(u)), f(u)}(\mathscr{Z}_T, u, \epsilon)$$

821 and:

$$\delta(u', u'') := d(u'') - d(u')$$

822 We now have the following lemma.

823 **Lemma F.4.** *Given $u', u'' \in \mathscr{Z}_T$ with $u'' \in \Downarrow_{\mathscr{Z}_t}(u') \setminus \{u'\}$ and indices $i', i'' \in \{0, 1\}$ we have that:*

$$\sum_{f \in \{0,1\}^{\hat{\mathcal{H}}(u', u'')}} [\![f(u') = i']\!][\![f(u'') = i'']\!]\tilde{\zeta}(u', u'', f)$$

824 *is equal to*

$$[\![i' \neq i'']\!]\phi_{\delta(u', u'')}(\epsilon) + [\![i' = i'']\!](1 - \phi_{\delta(u', u'')}(\epsilon))$$

825 *Proof.* We prove by induction on the distance from $u'$ to $u''$ in $\mathscr{Z}_T$. If this distance is one then we

826 have $u' = \uparrow_{\mathscr{Z}_T}(u'')$ and $\hat{\mathcal{H}}(u', u'') = \{u', u''\}$ so we have:

$$\sum_{f \in \{0,1\}^{\hat{\mathcal{H}}(u', u'')}} [\![f(u') = i']\!][\![f(u'') = i'']\!]\tilde{\zeta}(u', u'', f) = \tilde{\tau}_{i', i''}(\mathscr{Z}_T, u'', \epsilon)$$

827 which immediately implies that the inductive hypothesis holds in this case. Now suppose that the

828 inductive hypothesis holds whenever the distance from $u'$ to $u''$ is $j$. We now consider the case

829 that the distance from $u'$ to $u''$ is $j + 1$. Let $u^*$ be the child of $u'$ that lies in $\hat{\mathcal{H}}(u', u'')$. Without

830 loss of generality assume that $u''$ is a descendant of $\triangleleft(u^*)$. Now choose any $i^* \in \{0, 1\}$. Given

831 $f : \hat{\mathcal{H}}(u', u'') \to \{0, 1\}$ let:

$$h(i^*, f) = [\![f(u') = i']\!][\![f(u'') = i'']\!][\![f(u^*) = i^*]\!]$$

832 and let $f'$ and $f''$ be the restriction of $f$ onto the sets $\hat{\mathcal{H}}(u^*, u'')$ and $\mathcal{H}(\triangleright(u^*))$ respectively. Note that

$$\tilde{\zeta}(u', u'', f) = \tilde{\tau}_{f(u'), f(u^*)}(\mathscr{Z}_T, u^*, \epsilon)\tilde{\zeta}(u^*, u'', f')\hat{\zeta}(\triangleright(u^*), f'')$$

833 By Lemma F.3 and the inductive hypothesis we then have that the quantity:

$$\sum_{f \in \{0,1\}^{\hat{\mathcal{H}}(u', u'')}} h(i^*, f)\tilde{\zeta}(u', u'', f)$$

834 is equal to the quantity:

$$\tilde{\tau}_{i', i^*}(\mathscr{Z}_T, u^*, \epsilon)([\![i^* \neq i'']\!]\phi_{\delta(u^*, u'')}(\epsilon) + [\![i^* = i'']\!](1 - \phi_{\delta(u^*, u'')}(\epsilon)))$$

835 Summing over $i^* \in \{0, 1\}$ gives us the result. We have hence proved the result in general.  $\square$

836 Suppose we have some $f : \mathcal{J} \to \{0, 1\}$. Let:

$$\hat{h}(f) = \{f' \in \{0, 1\}^{\mathscr{Z}_T} \mid \forall u \in \mathcal{J}, \ f'(u) = f(u)\}$$

837 Given $u \in \mathcal{J}$ we have that:

$$[\![f(\uparrow_{\mathcal{J}}(u)) \neq f(u)]\!]\phi_{\delta(\uparrow_{\mathcal{J}}(u), u)}(\epsilon) + [\![f(\uparrow_{\mathcal{J}}(u)) = f(u)]\!](1 - \phi_{\delta(\uparrow_{\mathcal{J}}(u), u)}(\epsilon))$$

838 is equal to $\tilde{\tau}_{f(\uparrow_{\mathcal{J}}(u)), f(u)}(\mathcal{J}, u, \epsilon)$ and hence Lemma F.4 implies that:

$$\sum_{f' \in \hat{\mathcal{H}}(\uparrow_{\mathcal{J}}(u), u)} [\![f'(\uparrow_{\mathcal{J}}(u)) = f(\uparrow_{\mathcal{J}}(u))]\!][\![f'(u) = f(u)]\!]\tilde{\zeta}(\uparrow_{\mathcal{J}}(u), u, f') = \tilde{\tau}_{f(\uparrow_{\mathcal{J}}(u)), f(u)}(\mathcal{J}, u, \epsilon)$$

839 so since, by the definition of a contraction, the edge sets of the subtrees in $\{\hat{\mathcal{H}}(\uparrow_{\mathcal{J}}(u), u) \mid u \in$

840 $\mathcal{J} \setminus \{r(\mathcal{J})\}\}$ partition the edge set of $\mathscr{Z}_T$ we have, by definition of $\tilde{\zeta}$, that:

$$\sum_{f' \in \hat{h}(f)} \prod_{u \in \mathscr{Z}_T \setminus \{r(\mathscr{Z}_T)\}} \tilde{\tau}_{f'(\uparrow_{\mathscr{Z}_T}(u)), f'(u)}(\mathscr{Z}_T, u, \epsilon) = \