# OpenReview forum: "Nearest Neighbour with Bandit Feedback"
_NeurIPS.cc/2023/Conference — NeurIPS 2023 poster_

### Official Review · Reviewer_WneD · 2023-06-30

**Soundness:** 4 excellent
**Presentation:** 3 good
**Contribution:** 4 excellent
**Rating:** 7
**Confidence:** 2

**Summary:**

This paper considers contextual bandits in a nearest-neighbor paradigm. In this paradigm, the contexts exist in a metric space, such that contexts that are close in the metric space are also likely to admit the same "correct" action.
In other words, the decision boundary of the optimal mapping from context to action is assumed to be small.
Intuitively, in such a setting, a nearest-neighbor type strategy is reasonable, in which one decides on an action for the current context based on past history on nearby contexts.

First, the paper considers contextual bandits in the adversarial setting, with the added property that at any time step t, one of the contexts for a previous time step is "flagged". Intuitively, this flag corresponds to it being similar to the current context, providing some advice to the algorithm. The paper presents an algorithm for this model, and analyzes its regret in terms of the similarity between the current context and the flagged past context (as measured in terms of the optimal policy's assignment for these contexts). Next, the paper hones in on the case in which the flagged context is a (c-approximate) nearest neighbor of the current context, and simplifies the regret bound to be a function of the parameter c as well as the distance of the input from the decision boundary of the optimal policy.


The algorithm presented in the paper seems to be a very intricate construction of search trees over the given action space, to support the given regret bounds as well as efficient calculation.


Comments:

Section 4.2: I found the definition of the ternary tree to be technical and hard to understand. Perhaps give an intuitive explanation of the construction before delving into notation and formal definitions.


**Strengths:**

The assumption that contexts exhibit closeness-based similarity is natural, and allows for meaningful bounds even when the space of possible contexts is infinite.
The algorithm itself is interesting and very nontrivial.
The paper is generally well-written.

**Weaknesses:**

The paper is somewhat pseudocode and notation heavy. I would personally rather have more intuition about the various components and their role in the algorithm, even at the cost of not having a full description of every procedure in the main body of the paper.

**Questions:**

none

**Limitations:**

yes

---

> ### Author Rebuttal · Authors · 2023-08-08
>
> Thank you for your review - we have the following comments and responses (to the phrases in quotation marks).
>
> "In other words, the decision boundary of the optimal mapping from context to action is assumed to be small"
> - We note that our comparator policy $y$ can be anything - it does not need to be the exact optimal mapping (which could have a large decision boundary). Choosing a smoother comparator policy (lower $\Phi(y)$) gives a lower regret but choosing a more complex comparator policy (high $\Phi(y)$) may fit the data better (and so have a lower inherent loss).
>
> "Section 4.2: I found the definition of the ternary tree to be technical and hard to understand."
> - We understand it is very complex and we commit to writing an intuitive explanation.
>
> "The paper is somewhat pseudocode and notation heavy."
> - We wanted the main body to be complete in that all novel pseudocode is given. We do have an extra page if accepted so we can provide an “overview” subsection at the beginning of Section 4 where we can describe how the different components will fit together.

---

> > ### Comment · Reviewer_WneD · 2023-08-11
> >
> > Thank you for your response.

---

### Official Review · Reviewer_nfmq · 2023-07-01

**Soundness:** 3 good
**Presentation:** 3 good
**Contribution:** 3 good
**Rating:** 6
**Confidence:** 2

**Summary:**

The paper under review investigates the novel application of nearest neighbor search in the context of contextual bandit problems, which I find both new and intriguing. The main contribution lies in the derived result that bounds the regret using \Phi(y), a metric that quantifies the likelihood of disparate optimal choices among closely related contexts. I appreciate the generality and applicability of this metric, as it encapsulates contextual dependencies and has the potential for broad practical use. I am not familiar with the techniques used in the analysis part, but they look good to me.

**Strengths:**

This paper studies the application of nearest neighbor search in contextual bandits, which is new and practical. And it gets some interesting theoretical results.
The paper is written very clearly and has a good structure.

**Weaknesses:**

Some notations in the second part of section 2 (those related to trees) are not very intuitive and hard to interpret, but perhaps it is not easy to design symbols for them.

**Questions:**

Just to make sure that there is no hidden assumption on the relationship between context and loss vector? (compared with linear bandits)
Is it the case that if each context has completely different loss vectors, then \phi(y) will be very large and the bound becomes vacuous?

---

> ### Author Rebuttal · Authors · 2023-08-08
>
> Thank you for your review - we have the following comments and responses (to the phrases in quotation marks).
>
> "a metric that quantifies the likelihood of disparate optimal choices among closely related contexts."
> - $y$ can be anything. i.e. $y(x)$ needs not be the optimal choice for $x$ (see below). Typically though we are only interested in policies $y$ where $y(x)$ is often a good choice for $x$.
>
> "Some notations in the second part of section 2 (those related to trees) are not very intuitive and hard to interpret…"
> - we are sorry for the confusing symbols - we have tried to make them fit the definition where possible.
>
> "Just to make sure that there is no hidden assumption on the relationship between context and loss vector?"
> - There is no assumption whatsoever on the relationship between context and loss vector. There is also no restriction whatsoever on our comparator policy $y$ (i.e. we can choose $y$ to be anything - $y(x)$ does not need to be the best action for $x$). Choosing a smoother comparator policy (lower $\Phi(y)$) gives a lower regret but choosing a more complex comparator policy (high $\Phi(y)$) may fit the data better (and so have a lower inherent loss). If the contexts all have completely different loss vectors then yes - the bound will be vacuous (but no algorithm can achieve a non-vacuous bound in this case).

---

> > ### Comment · Reviewer_nfmq · 2023-08-16
> >
> > Thank you for your response. The paper looks good to me.

---

> ### Comment · Area_Chair_2JED · 2023-08-16
> **Please acknowledge rebuttal**
>
> Dear reviewer,
>
> The authors have posted a rebuttal. Please acknowledge that you have read it and indicate whether they have adequately addressed your concerns/comments. The author-reviewer discussion phase ends on Aug 21 so please engage with the authors before that if you need any more clarifications.
> Thanks,
> AC

---

### Official Review · Reviewer_UgjS · 2023-07-05

**Soundness:** 3 good
**Presentation:** 2 fair
**Contribution:** 2 fair
**Rating:** 6
**Confidence:** 2

**Summary:**

This paper studies the contextual multi-armed bandit problem in the adversarial setting. The authors propose an algorithm, CanProp, which utilizes an adaptive approximate nearest neighbor data structure to select the arm to pull for a given context.

**Strengths:**

The authors provide a novel algorithm for the contextual multi-armed bandit problem in the adversarial setting. The algorithm appears to be theoretically sound, and the authors provide regret guarantees, which they specialize to the stochastic case.

**Weaknesses:**

1. Practicality and implementability: The algorithm relies heavily on data structures which do not seem practically feasible. The authors do not provide any empirical results to support their claims that this can be faster than existing methods

2. Notation: the authors use significant notation, some of which may be required, but some of which is unnecessarily difficult to follow. For example, line 75, using $x$ to indicate a random element instead of $X$. This is confusing, as in the condition stated in line 128, the requirement is that the function of $x$, $P[y(X)=a|X=x]$, is Lipschitz in $x$.

**Questions:**

How central is computational efficiency to the novelty claims of this work? If exact nearest neighbors are trivially computed at each time step (resulting in $O(t)$ complexity in round $t$) is the regret guarantee trivial? That is to say, is the result in Appendix B already known and the benefit of this paper is in a computational speed up, or was the regret guarantee itself not previously known?

---

> ### Author Rebuttal · Authors · 2023-08-08
>
> Thank you for your review - we have the following comments and responses (to the phrases in quotation marks).
>
> " The algorithm relies heavily on data-structures that do not seem practically feasible"
> - Could you please elaborate on the phrase "do not appear to be practically feasible"? While these algorithms are complex, they are indeed implementable and exhibit remarkable speed. It's worth mentioning that we have detailed a slower but novel algorithm in Appendix B (referenced in Line 197) that achieves essentially the same level of regret and is straightforward to implement.
>
> "The authors do not provide any empirical results to support their claims that this can be faster than existing methods"
> - The alternative algorithm we present in Appendix B operates with a running time of $\Theta(NK)$ in worst-case scenarios, highlighting the exponential speedup of our main algorithm. Given the mathematical underpinning of this fact, we believe that there exists no imperative necessity to provide empirical validation for this assertion.
>
> "Notation: the authors use significant notation..."
> - We are sorry if the notation is difficult to follow and will attempt to make things clearer. $P[y(x)=a|x]$ is meant to be read as “the probability that $y(x)$ is equal to $a$, given $x$”.
>
> "How central is the computational efficiency to the novelty claims of this work?"
> - As far as we are aware our regret bound itself is completely novel, irrespective of the computational efficiency. Our “initial idea” in Appendix B is indeed novel. We commit to clarifying this. The regret guarantee for exact nearest neighbour is not at all trivial.

---

> > ### Comment · Reviewer_UgjS · 2023-08-10
> > **Response**
> >
> > Thanks for your point by point response. In light of the novelty of the regret guarantee (independent of computational efficiency) I am increasing my score 5->6, despite the lack of practical validation.
> >
> > 1. Navigating nets, to my understanding, are primarily objects of theoretical interest; large constants prohibit them from yielding practical speedups. Could the authors provide references regarding the practicality of these methods?
> >
> > 2. Implementability: the claims of computational efficiency are significantly highlighted in this work ("extreme efficiency" mentioned 3 times in the first page). However, theoretical claims often hide prohibitively large constants and lower order terms, when Big O notation is used. Thus, numerical simulations are a great way to corroborate theoretical claims. A similar point is true of regret; even seeing toy simulations with the slower algorithm (or with an oracle that provides the exact nearest neighbor) would be good to validate that the hidden constants are not too large. Obviously, NeurIPS makes no requirements that simulations be run. However in the absence of a careful analysis exposing constants and lower order terms, numerical simulations are the only way to demonstrate the practical efficiency of a proposed method. Thus, it seems misleading to refer to these untested methods as "extremely efficient".
> >
> > 3. The novelty of the regret bound on its own is interesting! Some additional discussion highlighting this would be helpful.

---

> > > ### Author Response · Authors · 2023-08-17
> > >
> > > Thank you for increasing your score. Here are the responses to your questions:
> > >
> > > 1. Yes - for navigating nets the time is exponential in the doubling dimension of the metric space so it is only efficient for relatively low dimensional spaces. There is a large volume of work on the efficient computation of approximate nearest neighbours - especially in Euclidean space. The reason that we chose to focus on navigating nets is that it applies to any metric space and we are sure that it updates in logarithmic time. We note that cover trees also have these properties and it appears that they are more practical. We will have a deeper look into some of the faster algorithms in Euclidean space to find out which have logarithmic-time updates and include such references. We will also try to find references on the practical use of navigating nets and cover trees.
> > >
> > > 2.  For the computational complexity, the $\mathcal{O}$ hides only a constant factor and we do not think it will be that large - the algorithm will be massive speedup (over our slower algorithm) for realistic values of $K$ and $T$.
> > > Concerning the regret, the $\tilde{O}$ hides a factor of only $\mathcal{O}(\sqrt{\ln(K)\ln(T)})$. The constant under the $\mathcal{O}$ will be very small. We choose to represent our bound under a $\tilde{O}$ to improve readability rather than to hide a large term.
> > >
> > > 3. Yes - we can add such discussion.

---

### Official Review · Reviewer_PH3Q · 2023-07-06

**Soundness:** 3 good
**Presentation:** 2 fair
**Contribution:** 3 good
**Rating:** 7
**Confidence:** 2

**Summary:**

This work studies adversarial contextual bandits. The approach to solving this problem considered in this work is to use the nearest neighbor (NN)search sub-routine algorithm, and the regret bound depends on a term that characterizes the efficiency of the NN oracle. The main advantage of using an NN-based algorithm is that the per-trial computation time can be improved exponentially, compared with previous EXP-4 based algorithms.

**Strengths:**

The proof is technically sound as far as I know.

The improvement of the algorithm running time is significant, if the exponential improvement is provided firstly by this work.

The presentation is clear.

**Weaknesses:**

Some typos. For example:
- line 212, $a_t$ equal to $z_{t, \log K}$-> $v_{t, \log K}$

No experiments. The authors could provide some simulation results to suggest their CBNN approach indeed works in practice. I am particularly interested in the comparison between CBNN and EXP-4, from both the computation time comparison and regret comparison.



**Questions:**

Is it possible to provide some hardness results to show that the per-trial computational time (polylog) is actually optimal?

**Limitations:**

This work is a theoretical work. It does need to address the societal impact issue.

---

> ### Author Rebuttal · Authors · 2023-08-08
>
> Thank you for your review - we have the following comments and responses (to the phrases in quotation marks).
>
> "the per-trial computation time can be improved exponentially, compared with previous EXP-4 based algorithms."
> - By Exp4 do you mean our “initial idea” of combining Exp4 and Belief propagation on a dynamic tree (mentioned in Line 197)? We would like to stress that this algorithm is (as far as we are aware) novel so can be viewed as part of our contribution.
>
> "if the exponential improvement is provided firstly by this work."
> - The slower algorithm is also our invention so the exponential improvement is also novel.
>
> "No experiments."
> - The running time of our Exp4 based algorithm is (in the worst case) strictly $\Theta(TK)$ per trial so CBNN is certainly an exponential improvement (we note that the Exp4 based algorithm is in itself part of our contribution). You are right, however, that the two algorithms are not exactly equivalent so there will, empirically, be a slight difference in the regret. If time allows we will endeavour to do some experiments.
>
> "Is it possible to provide some hardness results…"
> - We will think about this question
>
> "It does need to address the societal impact issue."
> - We will address this, but don't foresee any negative societal impact.

---

> ### Comment · Area_Chair_2JED · 2023-08-16
> **Please acknowledge rebuttal**
>
> Dear reviewer,
>
> The authors have posted a rebuttal. Please acknowledge that you have read it and indicate whether they have adequately addressed your concerns/comments. The author-reviewer discussion phase ends on Aug 21 so please engage with the authors before that if you need any more clarifications.
> Thanks,
> AC

---

### Official Review · Reviewer_j5Qr · 2023-07-26

**Soundness:** 3 good
**Presentation:** 3 good
**Contribution:** 3 good
**Rating:** 6
**Confidence:** 2

**Summary:**

This paper considers non-stochastic contextual bandit problems in which the regret is defined with an arbitrary decision policy that maps contexts to arms.
For this problem,
a framework of algorithms based on the nearest neighbor rule is developed.
The paper provides generic regret bounds for this framework and show regret bounds for semi-stochastic settings in which contexts consist of $d$-dimensional vectors.

**Strengths:**

- A general framework applicable to a wide range of contextual bandits is proposed.
- The proposed algorithm is superior in terms of computational efficiency due to the use of sophisticated data structures.
- The algorithmic procedures and theoretical results are clearly explained.

**Weaknesses:**

- Comparisons with existing studies (both in terms of approach and results) are limited.
-

**Questions:**

- Are there any regret lower bounds that can be compared to the results of this paper?
- This paper uses a general result (Theorem 3.2) to construct a bound for one specific example (Theorem 3.4). Theorem 3.2 appears to be so general that I expect to be able to derive nontrivial results for other specific examples examples as well (e.g., when the context is discrete).
Can you think of any such examples?

**Limitations:**

I have no concerns about the limitations and potential negative societal impact.

---

> ### Author Rebuttal · Authors · 2023-08-08
>
> Thank you for your review - we have the following comments and responses (to the phrases in quotation marks).
>
> "Comparisons with existing studies are limited."
> - As far as we are aware our work is the first to achieve any of the results given in our paper. In the case of Theorem 3.4 (and the new modification given in the general response) there are (worse or incomparable) results in the literature (see [21] and the citations therein), often limited to the fully-stochastic special case, that we can compare against and we commit to doing so. In the case of the more general theorems we know of no other non-trivial results to compare against. As far as we know this is the first time a nearest-neighbour approach has been applied to adversarial bandits and the first time a 1NN approach has been applied to any bandit problem. We commit to expanding our related work section.
>
> "Are there any regret lower bounds that can be compared to the results of this paper?"
> - Yes - we can show that (when the parameter $\rho$ is tuned correctly) our bound is almost optimal. To prove this first note that the non-contextual bandit problem with $S$ trials and $K$ arms has a regret lower bound of $\Theta(\sqrt{KS})$. Now consider any $\Phi$ and let $T=S\Phi$ for some arbitrary $S\in\mathbb{N}$. Let our sequence of contexts $\{x_1, x_2, … x_T\}$ be such that $x_t$ is the nearest neighbour (seen so far) of $x_{t+1}$. Now divide the sequence into $\Phi$ contiguous segments, all of length $S$, and in any particular segment let each context in that segment have the same associated action. With this knowledge we now have $\Phi$ independent problems each with $S$ trials. Each problem has a regret lower bound of order $\sqrt{KS}=\sqrt{KT/\Phi}$. The total regret must hence be lower bounded by $\Theta(\sqrt{KT\Phi})$ which is a logarithmic factor different from our upper bound.
>
> "This paper uses a general result (Theorem 3.2) to construct a bound for one specific example"
> - Theorem 3.4 is just an example. Our main results are theorems 3.2 and 3.3. These bounds can be applied to any metric space (Theorem 3.3) or anything (Theorem 3.2). Note also that with Theorem 3.3 we don’t need to know the metric space itself - we need only be able to compute the distance between any pair of contexts. This allows our results to be applied to many more applications than Theorem 3.4 allows. For example, the contexts could be machines connected to the internet and the distance could be how many links between two machines - or the contexts could be complex user profiles and the distance given by some algorithm which computes how similar two user profiles are. Also, our main results allow for any sequence of contexts whilst in Theorem 3.4 they must be drawn i.i.d. at random (which is unrealistic in many applications).

---

> > ### Comment · Reviewer_j5Qr · 2023-08-16
> >
> > Thank you very much for your kind response. I am satisfied with the responses and have no further questions.

---

### Author Rebuttal · Authors · 2023-08-08

We thank the reviewers for their time spent in reviewing our paper.

We would like to note that for our example problem (that of Theorem 3.4) we can reduce the asymptotic dependance on $T$ and $K$ to $\tilde{O}(T^{d/(d+1)}K^{1/(d+1)})$ by first quantising the contexts (a.k.a. binning) as a pre-processing step. Although it reduces the asymptotic dependence on $T$ and $K$, utilising this pre-processing step can lead to a regret much worse than without using it (due to the other terms in the regret) - hence, we would like to keep Theorem 3.4 whist adding this new result (if we have the reviewers' permission). We will now sketch the proof:

Choose some even natural number $q$ which will be tuned later. Let $D$ be the set of all vectors in $[0,1]^d$ such that each component is equal to $z/q$ for some natural number $z$. For each context $x_t$ first map it to the nearest point in $D$. For simplicity here we assume that the decision boundary is the set of vectors in $[0,1]^d$ in which all components except for the first are $1/2$ and the distribution $\mu$ is the uniform distribution (but this proof can be easily extended to capture any possibility). Now, given our policy y, define a new policy $y’$ on $D$ such that:
(1) if $x$ is not on the decision boundary then $y’(x)=y(x)$
(2) if $x$ is on the decision boundary then $y’(x)=1$.
Note that the expected loss of policy $y’$ is no more than that of $y$ plus $O(T/q)$. By applying Theorem 3.3 to the metric space $D$ with policy $y’$ we see that the expected regret of the algorithm w.r.t policy $y’$ is $\tilde{O}(\sqrt{q^{d-1}KT})$.
Hence, the expected regret of the algorithm w.r.t. policy $y$ is $\tilde{O}(\sqrt{q^{d-1}KT}+T/q)$. Setting $q=(T/K)^{1/(d+1)}$ gives us the result.

Note that by treating each bin independently we would get an expected regret of $\tilde{O}(\sqrt{q^{d}KT}+T/q)$ meaning that our nearest neighbour methodology saves us a whole dimension.

We note that, whilst utilising binning as a pre-processing step can sometimes help, it destroys our more general bounds.

---

### Decision · Program_Chairs · 2023-09-21

**Decision:**

Accept (poster)

**Comment:**

Reviewers were unanimous in their support for this paper, and I concur.